# CE-DYNAM (v1), a spatially explicit, process-based carbon erosion scheme for the use in Earth system models

Victoria Naipal[1,2], Ronny Lauerwald[3], Philippe Ciais[1], Bertrand Guenet[1], Yilong Wang[1]

[1] Laboratoire des Sciences des Sciences du Climat et de l'Environnement, CEA CNRS UVSQ, Gif-sur-Yvette, France

[2] Ludwig-Maximilians University, Munich, Germany

[3] Department of Geoscience, Environment and Society, Université Libre de Bruxelles, Brussels, Belgium

*Correspondence :* Victoria Naipal (vnaipal24@gmail.com)

**Abstract.** Soil erosion by rainfall and runoff is an important process behind the redistribution of soil organic carbon (SOC) over land, hereby impacting the exchange of carbon (C) between land, atmosphere and rivers. However, the net role of soil erosion in the global C cycle is still unclear as it involves small-scale SOC removal, transport and re-deposition processes that can only be addressed over selected small regions with complex models and measurements. This leads to uncertainties in future projections of SOC stocks and complicates the evaluation of strategies to mitigate climate change through increased SOC sequestration.

In this study we present the parsimonious process-based Carbon Erosion DYNAMics model (CE-DYNAM) that links sediment dynamics resulting from water erosion with the C cycle along a cascade of hillslopes, floodplains and rivers. The model simulates horizontal soil and C transfers triggered by erosion across landscapes and the resulting changes in land-atmosphere $CO_2$ fluxes at a resolution of about 8 km at the catchment scale. CE-DYNAM is the result of the coupling of a previously developed coarse-resolution sediment budget model and the ecosystem C cycle and erosion removal model derived from the ORCHIDEE land surface model. CE-DYNAM is driven by spatially explicit historical land use change, climate forcing, and global atmospheric $CO_2$ concentrations affecting ecosystem productivity, erosion rates and residence times of sediment and C in deposition sites. The main features of CE-DYNAM are (1) the spatially explicit simulation of sediment and C fluxes linking hillslopes and floodplains, (2) the relative low number of parameters that allow running the model at large spatial scales and over long-time scales, and (3) its compatibility with global land surface models, hereby, providing opportunities to study the effect of soil erosion under global changes.

We present the model structure, concepts, limitations and evaluation at the scale of the Rhine catchment for the period 1850-2005 AD. Model results are validated against independent estimates of gross and net soil and C erosion rates, and the spatial variability of SOC stocks from high-resolution modeling studies and observational datasets. We show that despite local differences, the resulting soil and C erosion rates, and SOC stocks from CE-DYNAM are comparable to high-resolution estimates and observations at sub-basin level.

We find that soil erosion mobilized around $66 \pm 28$ Tg ($10^{12}$ g) of C under changing climate and land use over the non-Alpine region of the Rhine catchment over the entire period, assuming that the erosion loop of the C cycle was in near steady-state by 1850. This caused a net C sink equal to 2.1 - 2.7% of the Net Primary Productivity of the non-Alpine region over 1850-2005 AD. This sink is a result of the dynamic replacement of C on eroding sites that increases in this period due to rising atmospheric $CO_2$ concentrations enhancing the litter C input to the soil from primary production.

**Keywords.** soil erosion; regional carbon cycle; carbon sink; Rhine catchment; regional modelling

**1 Introduction**

Soils contain more carbon (C) than the atmosphere and living biomass together. Relatively small disturbances (anthropogenic or natural) to soil C pools over large areas could add up to substantial C emissions (Ciais *et al.,* 2013). With the removal of natural vegetation and the introduction of mechanized agriculture, humans have accelerated soil erosion rates. Over the last two to three decades, studies have shown that water erosion (soil erosion by rainfall and runoff) amplified by human activities has substantially impacted the terrestrial C budget (Doetterl *et al.,* 2012; Lal, 2003; Lugato *et al.,* 2018; Van Oost *et al.,* 2007, 2012; Stallard, 1998; Wang *et al.,* 2017; Tan *et al.,* 2020). However, the net effect of water erosion on the C cycle at regional to global scale is still under debate. This leads to uncertainties in the future projections of the soil organic C (SOC) reservoir, and complicates the evaluation of strategies to mitigate climate change by increased SOC sequestration.

The study of Stallard (1998) was one of the first to show that water erosion does not only lead to additional C emissions but can also sequester C due to the photosynthetic replacement of SOC at eroding sites and the stabilization of SOC in deeper layers at burial sites. The study of van Oost *et al.* (2007) was the first to confirm the importance of the sequestration of SOC by agricultural erosion at global scale using isotope tracers. Wang *et al.* (2017) gathered data on SOC profiles from erosion and deposition sites around the world and confirmed that water erosion on agricultural land that started from the early/middle Holocene has caused a large net global land C sink. Other studies, however, argue that soil erosion is a net C source to the atmosphere due to increased SOC decomposition following soil aggregate breakdown during transport and at deposition sites (Lal, 2003; Lugato *et al.,* 2018). Most studies modeling soil erosion and its net effect on SOC dynamics at the global scale, however, did not account for the full range of complex effects of climate change, $CO_2$ -driven increase in productivity and potentially soil C inputs, harvest of biomass, land use change, and changes in cropland management (Borrelli *et al.,* 2018; Doetterl *et al.,* 2012; Chappell *et al.,* 2016; Lugato *et al.,* 2018; Van Oost *et al.,* 2007; Wang *et al.,* 2017). In addition, models used at large spatial scales mainly focus on hillslopes and removal processes and neglect floodplain sediment and SOC dynamics (Borrelli *et al.,* 2018; Chappell *et al.,* 2016; Lugato *et al.,* 2018; Van Oost *et al.,* 2007; Tan *et al.,* 2020). This can lead to substantial biases in the assessment of net effects of SOC erosion at the catchment scale as floodplains can store substantial amounts of sediment and C (Berhe *et al.,* 2007; Hoffmann *et al.,* 2013a,b). Studies

addressing long-term large-scale sediment yield from hillslopes and floodplains, such as Pelletier *et al.* (2012), do not
explicitly account for the redistribution of sediment and SOC over land.

Furthermore, soil erosion is one of the main contributors to particulate organic carbon (POC) fluxes in rivers and C export
to the coastal ocean. The riverine POC fluxes are usually much smaller than the SOC erosion fluxes, due to decomposition
and burial in floodplains and in benthic sediments, while POC losses occur in the river network (Tan *et al.,* 2017; Galy *et*
*al.,* 2015). Therefore, uncertainties in large-scale SOC erosion rates over land will lead to even larger uncertainties in
lateral C fluxes between land and ocean for past and future scenarios estimated by global empirical models on riverine C
export (Ludwig and Probst, 1998; Mayorga *et al.,* 2010).

To address these knowledge gaps, we present a parsimonious process-based Carbon Erosion DYNamics Model
(CE-DYNAM), which integrates sediment dynamics resulting from water erosion with the SOC dynamics at the regional
scale. The SOC dynamics are calculated consistently with drivers of land use change, $CO_2$ and climate change by a
process-based global land surface model (LSM), with a simplified reconstruction of the last century increase of crop
productivity. This modelling approach consists of a global sediment budget model coupled to the SOC removal, input, and
decomposition processes diagnosed from the ORCHIDEE global LSM in an offline setting (Naipal *et al.,* 2018). The main
aim of our study is to quantify the horizontal transport of sediment and C along the continuum of hillslopes and
floodplains, and at the same time analyze its impacts on the land-atmosphere C exchange. We validate the new model with
regional observations and high-resolution modelling results of the Rhine catchment. It should be noted here that the
structure of CE-DYNAM is designed in a way that the model can be adapted easily to other large catchments after
calibrating the model parameters to the specific environmental conditions in those catchments. We also discuss the model
uncertainties and the sensitivity of the model to changes in key model parameters and assumptions made. In the next
sections we give a detailed overview of the CE-DYNAM model structure, the coupling of erosion, deposition and transport
with the coarse-resolution SOC dynamics of ORCHIDEE, model application and validation for the non-Alpine region of
the Rhine catchment, and its potentials and limitations.

**2 Methods**

**2.1 General model description**

CE-DYNAM version 1 (v1) is the result of coupling a large-scale erosion and sediment budget model (Naipal *et al.,* 2016)
with the SOC scheme of the ORCHIDEE LSM (Krinner *et al.,* 2005). The most important features of the model are (1) the
spatially explicit simulation of lateral sediment and C transport fluxes over land linking hillslopes and floodplains, (2)
consistent simulation of vertical C fluxes coupled with horizontal transport, (3) the low number of parameters compared to
other C erosion models that operate at a high spatial resolution (Lugato *et al.,* 2018; Billings *et al.,* 2019), which allows

running the model at large spatial scales and over long time-scales up to several thousands of years, (4) the generic input fields for application to any region or catchment, and (5) compatibility with the modelling structure of LSMs.

In the ORCHIDEE LSM, terrestrial C is represented by eight biomass pools, four litter pools and three SOC pools. Each of the pools varies in space, time and over the twelve Plant Functional Types (PFTs). An extra PFT is used to represent bare soil. Anthropogenic and natural disturbances (as a result of climatic changes) to the C pools include fire, crop harvest, changes to the Gross Primary Productivity (GPP), litterfall, autotrophic and heterotrophic respiration (Krinner *et al.,* 2005; Guimberteau *et al.,* 2018). The C-cycle processes are represented by a C emulator that reproduces for each PFT all C pools and fluxes between the pools exactly as in ORCHIDEE in the absence of erosion. A net land use change scheme is included in the emulator with mass-conservative bookkeeping of SOC and C input when a PFT is changed into another from anthropogenic land use change (Naipal *et al.,* 2018). The sediment budget model has been added in the emulator to simulate large-scale long-term soil and SOC redistribution by water erosion using coarse-resolution precipitation, land-cover and LAI data from Earth System Models (Naipal *et al.* 2015, 2016). The C emulator including erosion removal was developed by Naipal *et al.* (2018) to reproduce the SOC vertical profile, removal of soil and SOC, and compensatory SOC storage from litter input. As soil erosion is assumed not to change soil and hydraulic parameters but only the SOC dynamics, the emulator allows substituting for the ORCHIDEE model and performing simulations on time scales of millennia with a daily time step and a spatial resolution of 5 arcminute (~ 8x8 km), which would be a very computationally expensive or nearly impossible with the full LSM. The concept and all equations of the emulator are described in Naipal *et al.* (2018). The following subsections describe the different components of the CE-DYNAM that couples the C and soil removal scheme (Naipal *et al.,* 2018) with the horizontal transport and burial of eroded soil and C (Naipal *et al.,* 2016).

**2.2 The soil erosion scheme**

The potential gross soil erosion rates are calculated by the Adjusted Revised Universal Soil Loss Equation (Adj. RUSLE) model (Naipal *et al.,* 2015), which is based on the Revised Universal Soil Loss Equation (RUSLE) (Renard *et al.,* 1997) and is part of the sediment budget model (Naipal *et al.,* 2016) (Fig 1). In the Adj. RUSLE the yearly average soil erosion rate is a product of rainfall erosivity ( $R$ ), slope steepness ( $S$ ), land cover and management ( $Cm$ ) and soil erodibility ( $K$ ):

$$E = S \times R \times K \times Cm \qquad (1)$$

Note that the original RUSLE model further includes a slope-length factor ( $L$ ), which gives the length of a field in the direction of steepest descent, and a support practice factor ( $P$ ), which accounts for management practices to mitigate soil erosion. These two factors have been excluded here, because their quantification still includes many uncertainties and is not practical for applications at regional to global scales. These factors are largely affected by local man-made structures

(such as field size) and management practices, which are difficult to assess for present day and whose changes over the past are even more uncertain. In addition, we focus in this study on the potential effect of soil erosion on the C budget without erosion-control (EC) practices.

Naipal *et al.* (2015) have developed a methodology to derive the *S* and *R* factors from 5 arcmin resolution (5 x 5 arcminute raster) data on elevation and precipitation, hereby preserving the high-resolution spatial variability in slope and temporal variability in erosivity. In the rest of the manuscript we will refer to X by X km/arcminute raster cells alway with X km/arcmin resolution. Despite the comparatively coarse resolution of the erosion model, the so derived *R* factor was shown to compare well with the corresponding high-resolution product published by Panagos *et al.* (2017). In the study of Naipal *et al.* (2016), where the soil erosion model was applied for the last millennium, the change in climate was taken into account in the calculation of the *R* factor. For this study, we assume that the climate zones as defined by the Koeppen-Geiger climate classification have not changed drastically since 1850 AD.

**2.3 The sediment deposition and transport scheme**

The sediment deposition and transport scheme is adapted from the sediment budget model described by Naipal *et al.* (2016), which was calibrated and validated for the Rhine catchment (Figs 1&2). In the sediment budget model rivers and streams are not explicitly simulated. Instead each grid cell contains a floodplain fraction to ensure sediment transport between the grid cells (transport from one grid cell to another can only follow the connectivity of floodplains). It should be noted that global soil databases do not identify floodplain soil as a separate soil class, although national soil databases might. Because we aim to present a carbon erosion model that should be also applicable for other similar catchments, we followed a two-step methodology to derive floodplains in the Rhine catchment. For this purpose we used hydrological parameters and existing data on hillslopes and valleys. First, grid cells were identified that consisted entirely out of floodplains. For this we used the gridded global data set of soil at 5 arcminute resolution, with intact regolith, and sedimentary deposit thicknesses of Pelletier *et al.* (2016) (Table 1), and identified lowlands and hillslopes based on soil thickness and depth to bedrock. The lowlands were classified as grid cells that contain only floodplains and no hillslopes. Second, we calculated the floodplain fraction ($A_{fl}$) of a grid cell *i* that has both hillslopes and floodplains as a function of stream length and width based on the methodology developed by Hoffmann *et al.* (2007) for the Rhine:

$$A_{fl}(i) = L_{stream}(i) \times W_{stream}(i) \tag{2}$$

Where, $L_{stream}$ is the stream length derived from the HydroSHEDS database (Lehner and Grill, 2013) (Table 1).

$$W_{stream}(i) = a \times A_{upstream}^{b}(i) \tag{3}$$

Where, $A_{upstream}$ is the upstream catchment area, and $a$ is equal to 60.8, and $b$ is equal to 0.3.

The parameters $a$ and $b$ have been derived using the scaling behavior of floodplain width as estimated from measurements
on the Rhine (Hoffmann *et al.,* 2007). The sediment deposition on hillslopes ($D_{hs}$) and in floodplains ($D_{fl}$) is calculated as a
function of the gross soil removal rates ($E$) according to Naipal *et al.* (2016) with the following equations:

$$D_{fl}(i) = f(i) \times E(i) \tag{4a}$$

$$D_{hs}(i) = (1 - f(i)) \times E(i) \tag{4b}$$

$$f(i) = a_f \times e^{\left(\frac{b_f \times \theta(i)}{\theta_{max}}\right)} \tag{5}$$

Where, $f$ is the floodplain deposition factor at 5 arcminute resolution that determines the fraction of eroded material
transported and deposited in the floodplain fraction of a grid cell. $a_f$ and $b_f$ are constants that relate $f$ to the average
topographical slope ($\theta$) of a grid cell depending on the type of land cover . $\theta_{max}$ us the maximum topographical slope of the
entire Rhine catchment.

The parameters $a_f$ and $b_f$ are chosen in such a way that $f$ varies between 0.2 and 0.5 for cropland, reflecting the decreased
sediment connectivity between hillslopes and floodplains created by man made structures such as ditches and hedges. For
natural vegetation such as forests and natural grassland, $a_f$ and $b_f$ are chosen in a way that $f$ varies between 0.5 and 0.8
assuming that in these landscapes hillslopes and floodplains are well-connected. This assumption on the reduced sediment
connectivity for agricultural landscapes is supported by several previous studies on the effect of erosion on sediment yield
(Hoffmann *et al.,* 2013a; de Moor and Verstraeten, 2008; Gumiere *et al.,* 2011; Wang *et al.,* 2015). These studies showed
that man-made activities on agricultural landscapes result in a trapping of eroded soil in colluvial deposition sites, reducing
the sediment transport from hillslopes to floodplains. The model parameter $f$ has been calibrated for the Rhine catchment
by Naipal *et al.* (2016), where the ranges mentioned above are found to produce a ratio between hillslope and floodplain
sediment storage that was comparable to observations. The studies of Wang *et al.* (2010; 2015) identified a range for the
hillslope sediment delivery to be between 50 and 80 %, which is similar to the range in the (1-f) factor in our model. In
each case and within the defined boundaries, the slope gradient determines the final value of $f$. Eroded material that has not
been deposited in the floodplains is assumed to be deposited at the foot of the hillslopes as colluvial sediment.

The floodplain fractions of the grid cells are connected through a 5 arcminute resolution flow routing network (Naipal *et*
*al.,* 2016), where the rivers and streams are indirectly included in the floodplain area but not explicitly simulated. By
routing the sediment and C through the floodplain fractions of grid cells we lump together the slow process of riverbank
erosion by river dynamics (time scale ≈ a few years to thousands of years), and the rather fast process of transport of

eroded material by the rivers (time scale ≈ days). The rate by which sediment and SOC leave the floodplain of a grid cell to go to the floodplain of an adjacent grid cell is determined by the sediment residence time. The sediment residence time (τ) is a function of the upstream contributing area (*Flowacc*):

$$\tau(i) = e^{\frac{Flowacc(i)-a_\tau}{b_\tau}} \qquad (6)$$

The study of Hoffmann *et al.* (2008) showed that the majority of floodplain sediments have a residence time that ranges between 0 and 2000 years, with a median of 50 years. The constants $a_t$ and $b_t$ are chosen in such a way that basin τ varies between the 5th and 95th percentile of those observations, with a median for the whole catchment of 50 years. These constants are uniform for the whole basin. These constants need to be calibrated based on local data of sediment ages before CE-DYNAM can be applied to other catchments.

Floodplain SOC storage follows the same residence time as sediment on top of the actual decomposition rate of C in a grid cell of ORCHIDEE. The routing of sediment and C between the grid cells follows a multiple-flow routing scheme. In this scheme the flow coming from a certain grid cell is distributed across all lower-lying neighbors based on a weight (*W*, dimensionless) that is calculated as a function of the contour length (*c*):

$$W_{(i+k,j+l)} = \frac{\theta_{(i+k,j+l)} \times c_{(i+k,j+l)}}{\sum\limits_{k,l=-1}^{k,l=1} \left[ \theta_{(i+k,j+l)} \times c_{(i+k,j+l)} \right]} \qquad (7)$$

Where *c* is 0.5 x grid size (m) in the cardinal direction and 0.354 x grid size (m) in the diagonal direction. (*i, j*) is the grid cell in consideration where *i* counts grid cells in the latitude direction and *j* in the longitude direction. *i+k* and *j+l* specify the neighboring grid cell where k and l can be either -1, 0 or 1. *θ* is calculated as the division between the difference in elevation (*h*) given in meters and the grid cell size (*d*), also in meters:

$$\theta_{(i+k,j+l)} = \frac{h_{(i,j)} - h_{(i+k,j+l)}}{d} \qquad (8)$$

The sediment and C routing is done continuously at a daily time-step to preserve the numerical stability of the model. More detailed explanation of the methods presented in this section can be found in the study of Naipal *et al.* (2016).

**2.4 Litter dynamics**

The four litter pools in the emulator are an below- and an above- ground litter pool, each split into a metabolic and structural pool with different turnover rates as implemented in ORCHIDEE (Krinner *et al.*, 2005). The belowground litter pools consist mostly out of root residues. Both the biomass and litter pools have a loss flux due to fire as incorporated in

ORCHIDEE by the Spitfire model of Thonicke *et al.* (2010). The litter that is not respired or burnt is transferred to the SOC pools based on the Century model (Parton *et al.,* 1987), which was modified by Naipal *et al.* (2018) to include a vertical discretization scheme for SOC.

The vertical discretization scheme was introduced in the emulator to account for a declining C input and SOC respiration with depth, and consists of 20 soil layers with each 10 cm thickness. The litter to soil fluxes from aboveground litter pools are all attributed to the top 10 cm of the soil profile. The litter to soil fluxes from belowground litter pools are distributed exponentially over the whole soil profile according to:

$$I_{be}(z) = I_{0be} \times e^{-r \times z} \qquad\qquad (9)$$

Where $I_{0be}$ is the below-ground litter input to the surface soil layer and *r* is the PFT-specific vertical root-density attenuation coefficient as used in ORCHIDEE. The sum of all layer-dependent litter to soil fractions is equal to the total litter to soil flux as calculated by ORCHIDEE. The vertical SOC profile is modified by erosion and the resulting deposition rates, which is discussed in detail in the following sections.

**2.5 Crop harvest and yield**

We adjusted the representation of crop harvest from ORCHIDEE by assuming a variable harvest index for C3 plants that increases during the historical period as shown in the study of Hay (1995) for wheat and barley, which are also the main C3 crops in the Rhine catchment. The harvest index is defined by the ratio of harvested grain biomass to above-ground dry matter production (Krinner *et al.,* 2005). In this study the harvest index increases linearly between 0.26 and 0.46 (Naipal *et al.,* 2018) consistent with the average values of Hay (1995).

Furthermore, we found that in certain cases the cropland Net Primary Productivity (NPP) was too high during the entire period of 1850 - 2005, especially in the early part of the 20[th] Century. This is because the cropland photosynthetic rates were adjusted in ORCHIDEE to give a cropland NPP representative of present day values that are higher than for the low input agriculture of the early 20[th] Century. To derive a more realistic NPP for wheat and barley in the Rhine catchment we used the long-term crop yield data obtained from a dataset on 120 000 yield observations over the 20[th] century in Northeast French Départements (NUTS3 administrative division) (Schauberger *et al.,* 2018). According to the yield data assembled by Schauberger *et al.* (2018), yields in Northeast France (covers part of the Rhine catchment) for these crops increased fourfold during the last century. Note that crop residues like straw constituted a larger fraction of the total biomass in 1850 than in 2005, but those residues were likely collected and used for animal feed, housing fuel. We did not account for this harvest of residue in the simulation of SOC.

**2.6 SOC dynamics without erosion**

The change in the C content of the PFT-specific SOC pools in the emulator without soil erosion was described by Naipal et al. (2018) (Fig 1) as following:

$$\frac{dSOC_a(t)}{dt} = lit_a(t) + k_{pa} \times SOC_p(t) + k_{sa} \times SOC_s(t) - (k_{ap} + k_{as} + k0_a) \times SOC_a(t) \tag{10}$$

$$\frac{dSOC_s(t)}{dt} = lit_s(t) + k_{as} \times SOC_a(t) - (k_{sa} + k_{sp} + k0_s) \times SOC_a(t) \tag{11}$$

$$\frac{dSOC_p(t)}{dt} = k_{ap} \times SOC_a(t) + k_{sp} \times SOC_s(t) - (k_{pa} + k0_p) \times SOC_p(t) \tag{12}$$

Where, $SOC_a$, $SOC_s$, and $SOC_p$ (g C m$^{-2}$) are the active, slow and passive SOC, respectively. The distinction of these SOC pools, defined by their residence times, are based on the study of Parton *et al.* (1987). The active SOC pool has the lowest residence time (1 - 5 years) and the passive the highest (200 - 1500 years). $lit_a$ and $lit_s$ (g C m$^{-2}$ day$^{-1}$) are the daily litter input rates to the active and slow SOC pools, respectively; $k0_a$, $k0_s$ and $k0_p$ (day$^{-1}$) are the respiration rates of the active, slow and passive pools, respectively; $k_{as}$, $k_{ap}$, $k_{pa}$, $k_{sa}$, $k_{sp}$ are the coefficients determining the flux from the active to the slow pool, from the active to the passive pool, from the passive to the active pool, from the slow to the active pool and from the slow to the passive pool, respectively.

The vertical C discretization scheme in the emulator assumes that the SOC respiration rates decrease exponentially with depth:

$$k_i(z) = k_{0i}(z) \times e^{-re*z} \tag{13}$$

Where $k_i$ is the respiration rate at a soil depth $z$ and $re$ (m$^{-1}$) is a coefficient representing the impact of external factors, such as decreasing oxygen availability with depth. $k_0$ is the respiration rate of the surface soil layer for a certain SOC pool $i$. The variable $re$ is determined in such a way that the total soil respiration of a certain pool over the entire soil profile without erosion is similar to the output of the full ORCHIDEE model. Detailed description of how this is done can be found in the study of Naipal *et al.* (2018).

**2.7 Net C erosion on hillslopes**

In the model we assume that soil erosion takes place on hillslopes, and not in the floodplains due to the usually low topographical slope of floodplains. The factor *(1-f)* determines the fraction of the eroded soil that is deposited in the colluvial reservoirs (Fig 1). Soil erosion always removes a fraction of the SOC stock in the upper soil layer depending on

the erosion rate and bulk density of the soil. The next soil layer contains less C and therefore at the following time-step less C will be eroded under the same erosion rate. In the model, the SOC profile evolution is dynamically tracked and updated at a daily time step, conform with the method of Wang *et al.* (2015). First, a fraction of the C from each soil pool in proportion to the erosion rate is removed from the surface layer. Then, at the same erosion rate, SOC from the subsoil layer becomes the surface layer, maintaining the soil layer thickness in the vertical discretization scheme. Similarly, the SOC from the subsoil later also moves upward one layer. The removal of C by erosion triggers a compensatory C sink due to the reduction in SOC respiration on eroding land. This compensatory C sink and reduced C erosion over time will ultimately lead to an equilibrium state. The change in C content due to net erosion (the eroded sediment/C that leaves the hillslopes after deposition) of the PFT-specific pools for hillslopes can be represented by the following equations:

$$\frac{dSOC_{HSi}(z,t)}{dt} = k_E \times SOC_{HSi}(z+1,t) - k_E \times SOC_{HSi}(z,t) \qquad (14)$$

Where $dSOC_{HSi}(z,t)$ is the change in hillslope SOC of a component pool $i$ at a depth $z$ and at time step $t$. The daily net erosion fraction $k_E$ (dimensionless) is calculated as following:

$$k_E = \frac{f \times \left(\frac{E}{365}\right)}{BD \times dz} \times EF \qquad (15)$$

Where, $E$ is the gross soil erosion rate (t ha$^2$ year $^{-1}$), $f$ is the floodplain deposition factor, $BD$ is the average bulk density of the soil profile (g cm$^{-3}$), $dz$ is the soil thickness (= 0.1 m), and $EF$ is the C enrichment factor that is set to 1 by default. A model sensitivity analysis will be performed (see section 4.3) with $EF > 1$ to represent a higher C concentration in eroded soil compared to the original soil as a result of the selectivity of erosion.

Hillslope erosion without the deposition term has already been tested and applied at the global scale as part of the C removal model presented by Naipal *et al.* (2018).

**2.8 C deposition and transport in floodplains**

The SOC-profile dynamics of floodplains are controlled by: (1) C input from the hillslopes, (2) C import by lateral transport from the floodplain fractions of upstream grid cells, and (3) C export to the floodplain fractions of downstream grid cells (Fig 1). First, the net erosion flux from the surface layer of the hillslope fraction of the grid cell ($k_E \times SOC_{HS}$ at z = 0) is incorporated in the surface layer of the floodplain. At the same deposition rate, the SOC of the surface layer of the floodplain is incorporated in the subsoil layer. Similarly, a fraction of the SOC of the subsoil layer is moved downward one layer. We will refer to this process as the 'downward' moving of C in the soil layer profile. It should be noted that C selectivity during transport and deposition is not taken into account here, meaning that the C pools of the deposited material are the same as the eroded material from the topsoil of eroding areas. At the same time as deposition takes place a

fraction of the C of the surface layer proportional to the sediment residence time ($\tau$) is exported out of the catchment
following the sediment routing scheme, resulting in the 'upward' moving of the C from the subsoil layers. This process
represents the river bank erosion and resulting POC export by the water network, although rivers and streams are not
explicitly represented in the model. As we do not have information on the sub-grid spatial distribution of land cover
fractions we first sum the exported C flux over all PFTs before assigning the flux proportionally to the land cover fractions
of the receiving downstream-located grid cells. The C that is imported from the neighboring grid cells follows the same
procedure as the deposition of eroded material, and results in a 'downward' moving of the C in the soil profile. The change
in C content due to deposition and routing of the PFT-specific SOC pools for floodplains can be represented by the
following equations:

$$\frac{dSOC_{FLi}(z,t)}{dt} = \left(\left(k_D + k_{i_{out}}\right) \times SOC_{FLi}(z-1,t)\right) + \left(\frac{1}{(\tau \times 365)} \times SOC_{FLi}(z+1,t)\right) - \left(\left(k_D + \frac{1}{(\tau \times 365)} + k_{i_{out}}\right) \times SOC_{FLi}(z,t)\right),$$
for z > 0                                                                                                     (16)

$$\frac{dSOC_{FLi}(0,t)}{dt} = \sum_{n=1}^{n=9} \left(k_{i_{out}}(n) \times SOC_{FLi}(0,t)(n)\right) + \left(k_E \times SOC_{HSi}(0,t)\right) + \left(\frac{1}{(\tau \times 365)} \times SOC_{FLi}(1,t)\right) - \left(\left(k_D + \frac{1}{(\tau \times 365)} + k_{i_{out}}\right) \times SOC_{FLi}(0,t)\right)$$
, for z = 0                                                                                                    (17)

Where $n$ is the neighboring grid cell that flows into the current grid cell, $dSOC_{FLi}(z,t)$ is the change in floodplain SOC of a
component pool $i$ at a depth $z$ and at time step $t$, and $SOC_{HS}$ is the hillslope SOC stock. $k_D$ is the deposition rate and equal
to:

$$k_D = \frac{k_E \times AREA_{HS}}{AREA_{FL}}$$                                                                (18)

Where $AREA_{HS}$ is the hillslope area and $AREA_{FL}$ is the floodplain area (m$^2$) of a grid cell. $k_{i_{out}}$ is the import rate per C pool $i$
from neighboring grid cells (dimensionless) and can be calculated as:

$$k_{i_{out}} = \frac{\sum_{n=1}^{n=9}\left(W \times \frac{1}{\tau \times 365} \times AREA_{FL}\right)(n)}{AREA_{FL}}$$  (19)

Where, $W$ is the weight index of equation 7.

The first term of equation 16 represents the 'downward' moving of the incoming C related to the C deposition flux from
the hillslope fraction of the grid cell and the lateral C import flux from the floodplain fractions of upstream neighboring
grid cells. The second term represents the 'upward' moving of SOC related to the lateral C transfer to downstream
neighboring grid cells. The third term of equation 16 represents the total C loss flux from the current soil layer $z$, which is a

result of either the 'upward' or 'downward' moving of the C in the soil profile. The first term of equation 17 represents the incoming lateral C flux from the floodplains of the upstream neighboring grid cells. The second term represents the C deposition flux coming from the hillslope fraction of the grid cell. The third term represents the 'upward' moving of the SOC from the subsoil layer to the topsoil layer as a result of sediment/C routing. The last term of equation 17 represents the total loss of C from the topsoil layer, of which part is distributed across the neighboring grid cells downstream ( $\frac{1}{(\tau \times 365)}$ ), and part is moved 'downwards' in the soil profile as a result of C deposition ( $k_D$ ) and the incoming lateral C from upstream grid cells ( $k_{i_{out}}$ ).

## 2.9 The land use change bookkeeping model

The land use change bookkeeping scheme includes the yearly changes in forest, grassland and cropland areas in each grid cell as reconstructed by Peng *et al.* (2017) (Table 1). Peng *et al.* (2017) derived historical changes in PFT fractions based on the LUHv2 land use dataset (Hurtt *et al.*, 2011), historical forest area data from Houghton, and present day forest area from ESA CCI satellite land cover (European Space Agency, ESA, 2014). By using different transition rules and independent forest data to constrain the changes in crop and urban PFTs they derived the most suitable historical PFT maps.

When land use change takes place, the litter and SOC pools of all shrinking PFTs are summed and allocated proportionally to the expanding PFTs, maintaining the mass-balance. In this way the litter pools and SOC stocks get impacted by different input and respiration rates for each soil layer. When forest is reduced, three wood products with decay rates of 1, 10 and 100 years are formed and harvested. The biomass pools of other shrinking land cover types are transformed to litter and allocated to the expanding PFTs. More details on the land use scheme are described in the study of Naipal *et al.* (2018).

## 2.10 Study-Area

The model is tested for the Rhine catchment (Fig 2), which has a total basin area of about 185,000 km$^2$ covering five different countries in Central Europe. Its large size is beneficial for the application of a coarse-resolution model such as CE-DYNAM to study large-scale regional dynamics in the C cycle due to soil erosion. The Rhine catchment has a contrasting topography, with steep slopes larger than 20 % upstream in the Alps, and large, wide and flat floodplains at the foot of the Alps, the upper Rhine and the lower Rhine. The floodplains store large amounts of sediment and C that originate from eroding hillslopes upstream. These sediment storages provide the possibility to study the long-term effect of erosion on hillslope and floodplain dynamics. Furthermore, the Rhine catchment has been experiencing different stages of land use change over the Holocene, with land degradation dating back to more than 5500 years ago (Dotterweich, 2013). In contrast, during the last two decades there has been a general afforestation and soil erosion has been decreasing. These land use

changes and changes in erosion make an interesting and important case to study the effect of anthropogenic activities on the C cycle in Europe.

In addition, the Rhine catchment has been the focus of many erosion studies providing observations on erosion and sediment dynamics that can be used for model validation (Asselman, 1999; Asselman *et al.,* 2003; Erkens, 2009; Hoffmann *et al.,* 2007, 2008, 2013a, 2013b; Naipal *et al.,* 2016). The global sediment budget model that forms the basis for the sediment dynamics scheme of CE-DYNAM has been validated and calibrated for the Rhine catchment with observations on sediment storage from Hoffmann *et al.* (2013a) and scaling relationships between sediment storage and basin area (Naipal *et al.,* 2016). Hoffmann *et al.* (2008, 2013a) did an inventory of 41 hillslope and 36 floodplain sediment and SOC deposits related to soil erosion over the last 7500 years. The floodplain sediment observations consist mostly out of organic material (gyttja, peat) and fine sediments (fine sand, loam, silt) in overbank deposits (Hoffmann *et al.,* 2008). These fine sediments are a result of long-term soil erosion on the hillslopes. Hoffmann *et al.* (2013a) found that the sediment and SOC deposits were quantitatively related to the basin size according to certain scaling functions, where floodplain deposits increased in a non-linear way with basin size while the hillslope deposits showed a linear increase with basin size. We use these relationships to validate the spatial variability in SOC storage of floodplains and hillslopes simulated by CE-DYNAM. The scaling relationships have the form of a simple power law:

$$M = a \times \left(\frac{A}{A_{ref}}\right)^b \tag{20}$$

Where $M$ is the sediment storage or the SOC storage, $a$ is the storage (Mt) related to an arbitrary chosen area $A_{ref}$, while $b$ is the scaling exponent.

**2.11 Input data and model simulations**

To create the C emulator that forms the underlying C cycle of CE-DYNAM, we first ran the full ORCHIDEE model for the period 1850-2005 at a coarse resolution of 2.5° degrees latitude and 3.75° degrees longitude, and output all C pools and fluxes. The pools and fluxes were then archived together and used to derive the turnover rates to build the emulator. The SOC scheme of the emulator that has been modified to account for soil erosion processes has been made to run at a spatial resolution of 5 arcminutes, similar to the original global sediment budget model. Then, we performed three main simulations with CE-DYNAM for the Rhine catchment. Simulation S0: The baseline simulation or no-erosion simulation, where SOC dynamics are similar to the full ORCHIDEE model. Simulation S1: The erosion-only simulation, where the hillslopes erode and all eroded C is respired to the atmosphere without reaching the colluvial and alluvial deposition sites. Simulation S2: The simulation with full sediment dynamics where hillslopes and floodplains are connected and can store or lose C. We ran the emulator for 3000 years at a daily time step with the initial climate and land cover of the period 1850 - 1860. To speed up the spin-up simulations we calculated the temporary equilibrium state of the floodplain SOC pools every

10 years analytically. At the end of the spin-up period the floodplain SOC pools were close to equilibrium, with a yearly change of less than 0.001 % of the total floodplain SOC stock. Afterwards, we performed the transient simulations for the period 1851 - 2005 at a daily time step with changing climate and land cover conditions, using the equilibrium SOC stocks as baseline. To ensure a faster performance of CE-DYNAM we delineated the Rhine catchment in seven large sub-basins and ran the model in parallel for each of the sub-basins at a daily timestep. After each year the sub-basins exchanged the lateral C fluxes with each other.

We also performed seven additional sensitivity simulations and four additional uncertainty simulations. Simulation S1_EF and S2_EF are performed to test the model assumption of C enrichment during erosion. Here, we changed the enrichment factor *EF* to two, based on the study of Lugato *et al.* (2018). Simulations S2_Tmin and S2_Tmax are performed to test the rate of C transport between floodplains. Here we modified the mean sediment residence time for the Rhine catchment to a minimum of 60 years (50 % lower than the current value), and to a maximum of 128 years (50 % higher than the current value), respectively. However, we kept the maximum sediment residence time at 1500 years. Simulations S0_RM, S1_RM and S2_RM are performed to test the model assumption on crop residue management, where we assumed that all above-ground crop litter is harvested.

For the uncertainty analysis we performed simulations S1_min and S2_min based on a minimum soil erosion scenario, and S1_max and S2_max based on a maximum soil erosion scenario. These soil erosion scenarios are derived from the uncertainty ranges in the rainfall erosivity and land cover factors of the erosion model. All the model simulations are summarized in table 2.

**2.12 Validation methods and data**

We performed a detailed model validation of the sediment and the C parts of the model according to the following steps: (1) validation of soil erosion rates using observational and high-resolution model estimates for Germany and Europe, (2) validation of C erosion rates using high-resolution model estimates for Europe from Lugato *et al.* (2018), (3) validation of the spatial variability of hillslope and floodplain C storage using observational results from Hoffmann *et al.* (2013a), (4) validation of SOC stocks using observational data from a global soil database and a European land use survey.

The validation of the soil erosion module has been done before in the studies of Naipal *et al.* (2015, 2016). However, we do it again in this study due to different input datasets. In addition, the validation includes soil erosion data from new global soil erosion studies such as Borrelli *et al.* (2018) and Panagos *et al.* (2015). For the validation of gross soil erosion rates we used the high-resolution model estimates of Panagos *et al.* (2015), who applied the RUSLE2015 model at a 100 m resolution at European scale for the year 2010. Similarly to the Adj.RUSLE, RUSLE2015 is also derived from the original RUSLE model. However, in contrast to our model, RUSLE2015 does include the erosion factors *L* and *P*. Furthermore, our model uses more coarsely resolved input datasets (Table 1), for which the equations for the *R* and *S* factors have been

modified. Thus, even though both Adj.RUSLE and RUSLE2015 are derived from the same erosion model, the differences between the models are large, which justifies our model comparison. The extensive validation of the Adj.RUSLE model in this study and previous studies (Naipal *et al.*, 2015, 2016, 2018), shows that despite its coarse resolution, it is applicable at large spatial scales.

Furthermore, we used independent high-resolution erosion estimates from the study of Cerdan *et al.* (2010), available at a 1 km resolution at European scale, which were based on an extensive database of measured erosion rates under natural rainfall in Europe. For the comparison we aggregated the high-resolution model results of both datasets to the resolution of CE-DYNAM. We also used the potential soil erosion map of the Federal Institute for Geosciences and Natural Resources of Germany (Bug *et al.*, 2014) for comparison. This map presents the yearly average soil erosion rates at a 250 m resolution on agricultural land derived from a USLE-based approach, with some modifications to the erosion factors and input data. Before validating our model results we aggregated these high-resolution erosion rates also to the coarser resolution of our model.

Validation of our net soil erosion rates is done based on the 100 m resolution net soil erosion rates derived with the WATEM-SEDEM model (Borrelli *et al.,* 2018). WATEM-SEDEM simulates soil removal by water erosion based on the USLE approach, sediment transport and deposition based on the transport capacity. The model has been extensively employed to estimate net fluxes of sediments across hillslopes at catchment- and regional-scales.

For the validation of C erosion rates, we used the high-resolution model results from Lugato *et al.* (2018), where they coupled the RUSLE2015 erosion model to the Century biogeochemistry model. These model results were available at a resolution of 1 km, where each grid cell was composed of an erosion and deposition fraction. The C erosion rates provided by Lugato *et al.* (2018) were multiplied with the erosion fraction of a 1 km grid cell. Then, the C erosion rates were aggregated to the resolution of CE-DYNAM. Lugato *et al.* (2018) provided an enhanced and a reduced erosion-induced C sink uncertainty scenario, based on different assumptions for C enrichment, burial and C mineralization during transport. In CE-DYNAM the C erosion rates from simulation S1 are multiplied with the hillslope area to get the total C erosion flux of a grid cell. As the study of Lugato *et al.* (2018) considers only agricultural areas, we considered only the crop fraction of a grid cell during the comparison. It should be noted that the SOC dynamics scheme of CE-DYNAM, which is derived from ORCHIDEE LSM, is also based on the Century model. However, there are large differences between the Century model used by Lugato *et al.* (2018) and the C dynamics scheme of ORCHIDEE used in this study. For example, in the Century model the crop productivity is mediated by nitrogen availability, which is not the case in the ORCHIDEE version used for this study. The Century model also includes some management practices such as crop rotations, which are not represented in ORCHIDEE. The Century model runs at a much higher resolution and is calibrated for agricultural land, while ORCHIDEE also simulates forest, grasslands and bare soil. In this way, the final SOC stocks derived with CE-DYNAM are also a result of erosion from other land cover types and land use changes. This is an important feature for land use change, which is not included in the Century model. Furthermore, the ORCHIDEE LSM has been used in many global

intercomparisons and extensively evaluated for C budgets (Müller *et al.,* 2019; Todd-Brown *et al.,* 2013). Finally, ORCHIDEE also includes the last century change in crop production calibrated against data (Guenet *et al.,* 2018).

For the validation of the spatial variability of the SOC stocks of hillslopes and floodplains we used the scaling relationships between basin area and SOC storage derived by Hoffmann *et al. (*2013a). The study by Naipal *et al.* (2016) found that the global sediment budget model is able to reproduce the scaling behaviour of sediment storage. After analyzing the dependence of this scaling behavior, they argue it is an emergent feature of the model and mainly dependent on the underlying topography. This indicates that the scaling features of floodplain and hillslope sediment and C storage should also be applicable to a more recent time period. In order to evaluate the ability of CE-DYNAM to reproduce this scaling behavior for SOC, we selected the grid cells that contained the points of observation of the study of Hoffmann *et al.* (2013a) and performed a regression of the basin area (defined as the upstream contributing area) and the SOC storage for floodplains and hillslopes separately. Comparing the absolute values of the sediment and SOC storages of each grid cell from Hoffmann *et al.* (2013a) was not possible due to the difference in the time-period of the studies, where Hoffmann *et al.* (2013a) focussed on the entire Holocene, while our study focussed only on the period starting from 1850 AD.

For the validation of the total SOC stocks we used the Global Dataset for Earth System Modeling (GSDE) (Shangguan *et al.,* 2014) available at a spatial resolution of 1 km and the Land Use/Land Cover Area Frame Survey (LUCAS) (Palmieri *et al.,* 2011). The LUCAS topsoil SOC stocks, available at a high spatial resolution of 500 m, were calculated using the LUCAS SOC content for Europe (de Brogniez *et al.,* 2015) and soil bulk density derived from soil texture datasets (Ballabio *et al.*, 2016).

**3 Results**

Due to large uncertainties in the model and validation data for the Alpine region we only present and discuss the model and validation results for the non-Alpine part of the Rhine catchment.

**3.1 Model validation**

In this section we present the model validation results using the methods and data described in detail in the previous section.

We find that the quantile distribution of the simulated gross soil erosion rates compares well to the distributions of other observational and high-resolution modelling studies (Cerdan *et al.,* 2010, Panagos *et al.,* 2015, Bug *et al.*, 2014), although CE-DYNAM usually underestimates the very large soil erosion rates such as is found by Cerdan *et al.* (2010) (Fig 3A, B, C). This is due to the coarse spatial and temporal resolution of CE-DYNAM, and the lack of the slope-length factor ( $L$ ) (Cerdan *et al.* (2010) assumed a constant slope length of a 100 m). It should be noted that our study, Cerdan *et al.* (2010) and Bug *et al.* (2014) simulated potential soil erosion rates, not accounting for EC practices represented by the *P* factor.


We also find that the quantile distribution of the simulated net soil erosion from hillslopes compares well with the
distribution from the high-resolution modelling study of Borrelli *et al.* (2018) (Fig 3D). In addition we performed a spatial
comparison of our simulated gross and net erosion rates to those of the studies mentioned above. For this purpose we
delineated 13 sub-basins in the Rhine catchment (Fig S3). Table 3 summarizes the resulting goodness-of-fit statistics of this
comparison and shows that for gross soil erosion our erosion model is generally in good agreement with the other studies at
sub-basin level. However, for net soil erosion, our model results are different to those of the study of Borrelli *et al.* (2018)
due to the different approaches in calculating the sediment deposition. For example, in our study the deposition of sediment
in hillslopes is explicitly calculated as a function of the slope, and vegetation type/cover. Borrelli *et al.* (2018) used the
transport capacity concept (Van Rompaey *et al.,* 2001). Both methods have their uncertainties when applied at large spatial
scales. The method in our study has been designed and calibrated to be used at a large spatial scale, and at coarse
resolution, while the method of Borrelli *et al.* (2018) was originally designed to be applied at spatial scales < 100 m.

We find that the quantile distributions of our simulated agricultural C erosion and deposition rates are similar to those of
the high-resolution modelling study of Lugato *et al.* (2018) (Figs 4A-D). Also the spatial variability of the C erosion rates
at sub-basin level is in good comparison to the validation data (Table 4). However, the linear regression between soil
erosion and C erosion rates of our study lies at the lower end of the relationships derived from the enhanced and reduced
erosion scenarios of Lugato *et al.* (2018) (Fig 5). On the one hand, our study does not include EC practices, leading to
substantially larger simulated soil erosion rates in regions with EC. Figure 5 shows that our simulated erosion rates are in
general larger than the erosion rates from Lugato *et al.* (2018), which may be explained by this mechanism. On the other
hand, the C erosion rates of our study are lower than those of Lugato *et al.* (2018), due to the coarse spatial resolution of
our underlying C-scheme derived from the ORCHIDEE LSM. The decreased spread in our simulated values is also a result
of the coarse resolution of our model.

Accounting for erosion, deposition and transport of SOC leads to a better representation of the simulated topsoil C stocks
per land cover type when compared to SOC stocks of the LUCAS database (Fig 6). The simulated SOC stocks of the top
20 cm of the soil profile fall within the quantile range of the LUCAS SOC stocks for cropland and forest (Fig 6). Although
the topsoil SOC stocks for grassland improved, a large uncertainty range remains. Furthermore, we find that in both the
erosion and no-erosion simulations the SOC stocks for grassland are higher than for forest. This is also observed in the
study of Wiesmeier *et al.* (2012), where they found considerable higher SOC stocks for grassland with a median of 11.8 kg
C m$^{-2}$ compared to forest based on the analysis of 1460 soil profiles in South-Germany. Furthermore, the comparison of the
simulated total SOC stocks to those of the LUCAS and GSDE databases at sub-basin level shows a good model
performance with respect to the spatial variability in topsoil SOC stocks (Table 5).

To validate the spatial variability of floodplain and hillslope SOC stocks separately, we used the scaling relationships found
by Hoffmann *et al.* (2013a) (section 2.12). We find a significantly larger exponent for the scaling relationship between the

simulated floodplain SOC storage and basin area compared to the simulated hillslope SOC storage, when using the grid cells that contain the points of observation corresponding to the study of Hoffmann *et al.* (2013a). This result is in line with what Hoffmann et al. (2013a) found and shows that CE-DYNAM can realistically reproduce the spatial variability in SOC stocks between hillslopes and floodplains (Table 6). However, when deriving the scaling relationships at sub-basin level instead of using individual grid cells we do not find a significant difference in the scaling between floodplains and hillslopes (Table 6).

**3.2 Model application**

We find an average annual soil erosion rate of $1.44 \pm 0.82$ t $ha^{-1}$ $year^{-1}$ over the period 1850 - 2005, which is about half of the  average erosion rate simulated for the last millennium (Naipal *et al.*, 2016) and falls within the range of the average erosion rates of the Holocene (Hoffmann *et al.*, 2013a). This soil erosion flux mobilized around $66 \pm 28$ Tg of C over the same time period, of which on average 57 % is deposited in colluvial reservoirs, 43 % is deposited in alluvial reservoirs, and 0.2 % is exported out of the catchment.

The lower average annual soil erosion rate over the study period compared to the last millennium is a result of the general afforestation in the non-Alpine part of the Rhine catchment that started around 1910 AD according to the data on land cover and land use (Peng *et al.*, 2017; Fig 7B). This land cover data also shows that forest increased by 24 % over the period 1910 - 2005, mostly as a result of grassland to forest conversion. Cropland decreased by 6 % over the period 1920 and 1970, and is relatively stable afterwards. This afforestation leads to a long-term decreasing trend in gross soil and SOC erosion rates on hillslopes (Fig 7C). The temporal variability in the soil and C erosion rates is a result of direct changes in precipitation, as is shown by the temporary increase in erosion rates over the period 1940 - 1960 (Fig 7A). Furthermore, we find that the temporal variability in C erosion rates follows the soil erosion rates closely, indicating that soil erosion dominates the variations in C erosion over this time-period, while increased SOC stocks due to $CO_2$ fertilization and afforestation play a secondary role as a slowly varying trend. It should be noted that the correlation between soil and C erosion might be affected by processes not properly captured by the model such as the selectivity of erosion including the enrichment of C in eroded material.

The cumulative C erosion removal flux of $66 \pm 28$ Tg of C leads to a cumulative net C sink for the whole Rhine region of $216 \pm 23$ Tg C (Fig 7D). This is about 2.1 – 2.7 % of the cumulative NPP and of the same magnitude as the cumulative land C sink of the Rhine without erosion. It should be noted that these are potential fluxes, assuming that the photosynthetic replacement of C is not affected by the degradation of soil due to the removal of nutrients, declining water-holding capacity and other negative changes to the soil structure and texture (processes not covered by our model). The breaking point in figure 7D around 1910 AD is a result of the climate data used as input.

To better understand the erosion-induced net C flux, we analyze the erosion-induced C exchange with the atmosphere by creating C budgets for the entire Rhine catchment for the period 1850 - 1860 and for the period 1950 - 2005 (Figs 8A&B). These C budgets also shed light on changes in the linkage between lateral and vertical C fluxes over time. As we do not explicitly track the movement of eroded C through all reservoirs (for example between eroding hillslopes and colluvial reservoirs), we make use of the changes in SOC stocks and Net Ecosystem Productivity (NEP), which is the difference between NPP and heterotrophic respiration, of the three main simulations (S0, S1, S2) to derive the erosion-induced vertical C fluxes. By subtracting the NEP of hillslopes ($NEP_{HS}$) of the no-erosion simulation (S0) from the erosion-only simulation (S1), we derive the additional photosynthetic replacement of SOC on eroding sites (Eq. 21):

$$E_{rep} = NEP_{HS}(S1) - NEP_{HS}(S0) \qquad (21)$$

Where, $E_{rep}$ is the potential dynamic Photosynthetic replacement of C on eroding sites (assuming no feedback of erosion on NPP). Part of the eroded C that is transported to and deposited in colluvial reservoirs can be respired or buried (Eq. 22). The difference between NEP of simulation S2 and S1 is the NEP caused by the deposition of eroded C in colluvial areas and equal to the difference between the burial and respiration of C in colluvial sites. As we do not explicitly track the respiration of deposited material in the model, we can only derive the net respiration or net burial of C in colluvial deposits ($Rc_{net}$) with the following equation:

$$Rc_{net} = NEP_{HS}(S2) - NEP_{HS}(S1) \qquad (22)$$

The same concept can be applied for the net respiration/burial of floodplains:

$$Ra_{net} = NEP_{FL}(S2) - NEP_{FL}(S0) \qquad (23)$$

Where, $NEP_{FL}$ is the floodplain NEP, and $Ra_{net}$ is the net respiration or net burial of alluvial deposits. Positive values for $Ra_{net}$ or $Rc_{net}$ indicate a net burial (respiration S2 < respiration S0/S1) of the deposited material.

We find that the dynamic replacement of C on eroding sites increased by 17 - 33 % at the end of the period despite decreasing soil erosion rates (Figs 8A & B). This increase in the photosynthetic replacement of C is due to the globally increasing $CO_2$ concentrations that lead to the $CO_2$ fertilization effect, amplified by the afforestation trend in the Rhine over this period. Without this fertilization effect, soil erosion and deposition would be likely a weaker C sink or even a C source over the period 1850 - 2005 (Figs S4A & B). This $CO_2$ fertilization effect promotes a 100% replacement of the eroded C on hillslopes and even leads to a C sink on hillslopes at the end of the study period (Fig 8B). Furthermore, we find that the yearly average gross C erosion flux from eroding sites decreases by 10 - 34 %, while the yearly deposition fluxes in colluvial and alluvial sites decreases by 20 % and 19 - 47 %, respectively. The decrease in the deposition flux to

floodplains is compensated by a better sediment connectivity between hillslopes and floodplains due to afforestation.
Forests have less man-made structures that can prevent the erosion fluxes from reaching the floodplains, which is
represented by a higher floodplain deposition '$f$' factor in the model. The decrease in the erosion flux also leads to a
decreased POC export of the catchment at the end of the study period.

We also find that both the colluvial and alluvial reservoirs show a net respiration flux throughout the time period (Figs 8A
& B). This is consistent with previous studies who found that deposition sites can be areas of increased $CO_2$ emissions
(Billings *et al.,* 2019; Van Oost *et al.,* 2012). However, there is a slight difference in the respiration of deposited C between
the start and end of the transient period. The respiration of deposited SOC in colluvial sites increases with time while the
respiration of deposited SOC in alluvial sites shows rather a decreasing trend. These changes in SOC respiration of
deposited material depends on (1) the amount of deposited material, (2) increasing temperatures over 1850 - 2005 for the
entire catchment, and (3) the constant removal of C-rich topsoil and its deposition in alluvial and colluvial reservoirs,
which makes the deposited sediments generally richer in C than soils on erosion-neutral sites, providing more substrate for
respiration. The largest increase in total respiration of alluvial and colluvial deposits over time takes place in hilly regions
due to the initial increase in erosion rates resulting in large deposits of C. Overall, we find that the increased respiration of
deposited material slightly offsets the increased dynamic C replacement, however, the dynamic C replacement on eroding
sites still dominates the erosion-induced C sink.

**4 Discussion**

In this section we discuss some of the most important model limitations, uncertainties and assumptions.

**4.1 Initial conditions and past global changes**

Initial climate and land cover/use conditions, and the length of the transient period are essential parameters that determine
the resulting spatial distribution of soil and C. Landscapes are in a constant transient state due to global changes, such as
climate change, land use change, accelerated soil erosion. However, we assumed an equilibrium state so that we can
quantify the changes during the transient period. The longer the transient period that covers the essential historical
environmental changes, the more accurate are the present-day distribution of SOC stocks, sediment storages, and related
fluxes. This is especially true when analyzing the redistribution of soil and C as a result of erosion, deposition and
transport, as these soil processes can be very slow. For example, the study of Naipal *et al.* (2016) showed that by
simulating the soil erosion processes for the last millennium a spatial distribution of sediment storages that is similar to
observations can be found. In this study we simulated the steady state based on the initial conditions of the period 1850 -
1860 due to constraints in data availability on precipitation and temperature. By focusing only on the period 1850 - 2005
we miss the effects of significant land use changes in the past that coincided with times of strong precipitation such as in
the 14th and 18th century (Bork *et al.*, 2003). These major anthropogenic changes in the last Holocene substantially affected
the present-day spatial distribution and size of sediment storage and SOC stocks.

The absolute value of the SOC storage from the S2 simulations of the non-Alpine region of the Rhine catchment for the
year 2005 ranges between 2.74 - 2.99 Pg of C, which is larger than the 1.7 ± 0.6 Pg of C that Hoffmann *et al.* (2013a)
measured. It should be noted that the ORCHIDEE model (S0 simulation) already overestimates the total SOC stock of the
non-Alpine region of the Rhine (2.43 Pg of C), when the initial conditions of the period 1850 - 1860 are used. Due to the
fact that we miss the climate and land use changes before the year 1850, we find that floodplains store less SOC than
hillslopes. Although this is in contrast to the findings of Hoffmann *et al.* (2013a), the difference in SOC stocks between
floodplains and hillslopes from the S2 simulations is better than the difference derived from the S0 simulation. We find that
floodplains store between 1.28 - 1.72, and hillslopes store between 1.7 - 2 Pg of C when erosion and deposition processes
are taken into account, compared to 0.69 Pg of C for floodplains and 2.29 Pg of C for hillslopes when these processes are
lacking.

We also find that floodplains have an overall higher C concentration (12 kg m$^{-2}$for a 2 m soil profile) compared to
hillslopes (9 kg m$^{-2}$ for a 2 m soil profile) at the end of the transient period (Fig 9A), which is in line with the findings of
Hoffmann *et al.* (2013a) and what can be derived from global soil databases. This is a result of higher SOC concentrations
in deeper soil layers of floodplains compared to hillslopes (Figs 9 A&B), as is also shown in the study of Hoffman *et al.*
(2013). To be closer to the observational difference between floodplains and hillslopes we would need to consider the
period before the year 1850, extreme climate events, and a higher plant productivity in floodplains resulting from favorable
soil nutrient and hydrological conditions.

**4.2 Model advantages and limitations**

Although we parameterized and applied CE-DYNAM for the Rhine catchment, it is intended to be made applicable to
other large catchments. CE-DYNAM combines soil erosion processes, for which small scale differences in topography are
of utter importance, with a state-of-the-art representation of large-scale SOC dynamics driven by land use and
environmental factors (climate, atmospheric $CO_2$) as simulated by the ORCHIDEE LSM. The flexible structure of
CE-DYNAM makes the model adaptable to the SOC dynamics of other LSMs. In this way it is possible to study the main
processes behind the linkages between soil erosion and the global C cycle.

CE-DYNAM explicitly accounts for hillslope and floodplains re-deposition, which is to our knowledge unique for a
large-scale C erosion model and highly novel. However, it still lacks important processes affecting the C dynamics in
floodplains. The model does not account for a slower respiration rate due to low-oxygen conditions, physical and chemical

stabilization (Berhe *et al.,* 2008; Martínez-mena *et al.,* 2019). The oxidation and preservation of C in deposition environments, especially in alluvial reservoirs remain highly uncertain (Billings *et al.,* 2019).

Due to its simplistic nature and coarse-resolution, CE-DYNAM does not resolve rivers and streams explicitly but assumes that they are included in the floodplain part of the grid cells. As a result, CE-DYNAM does not differentiate between eroded hillslope soil that reaches the water network directly (where the residence time of suspended sediment is in the order of days), and the sediment that is first retained in the floodplains before it reaches the water network due to fluvial erosion (sediment residence time is in the order of a few to thousands of years). CE-DYNAM has been developed and calibrated to simulate long-term changes in sediment and C storage on land and not the short-term variations in sediment and POC fluxes carried by rivers. This limits the application of CE-DYNAM in its current form to accurately quantify sediment and POC fluxes of rivers and streams, and to compare them to observations.

As a result of the above-mentioned model limitation, CE-DYNAM produces a sediment export flux at the end of the year 2005 of about 6472 tonnes per year, which is about two orders of magnitude lower than the estimated suspended sediment flux of about $3.15 \times 10^6$ tons year$^{-1}$ from Asselman *et al.* (2003) or the $0.75 \times 10^6$ tons year$^{-1}$ simulated by Li *et al.* (2020). This sediment export rate leads to a yearly sediment bound POC export of about $2 \times 10^8$ g C year$^{-1}$ 2005. This POC flux is also two orders of magnitude lower than the $2.6 \times 10^{10}$ g C year$^{-1}$ given by the GlobalNEWS2 model (Mayorga *et al.,* 2010) or the $1.5 \times 10^{11}$ g C year$^{-1}$ found by Beusen *et al.* (2005), which is mainly a result of the underestimated simulated sediment export rate.

Furthermore, CE-DYNAM does not simulate fluvial erosion as a complex function of the channel geometry, riverbank erodibility and shear stress (Dröge *et al.,* 1992), due to the lack of data on these parameters at the regional scale, and to keep a balance between model complexity and its computational ability. Also, our model does not resolve erosion of the deposited river sediment by flooding events. This simplified model concept for fluvial erosion contributes to the underestimation of sediment and C export in floodplains. Finally, with the current model setup we do not account for large soil erosion events before 1850 AD or extreme precipitation events that may have a long-term effect on the sediment export rate of the Rhine.

Although we underestimate the riverine sediment and POC fluxes, we find that the spatial variability in sediment storage and SOC stocks of the sub-basins are within or close to observational uncertainty ranges (Table 5, 6; Naipal *et al.,* 2016). We also find that the C density in the topsoil layers of floodplain soils located downstream of the Rhine and the C concentration of the POC flux are realistic. We find a C concentration of ~3.3 % in the exported fine sediments downstream of the Rhine. Abril *et al.* (2005) found a 5.5 % POC mass fraction in suspended sediments for the Rhine. The C density of the topsoil layer of the floodplains in the downstream grid cells in the S2 simulations (S2, S2_min, S2_max) is on average 4.47 kg C m$^{-2}$, which falls within the range of the average C density of 5.13 ± 1.3 kg C m$^{-2}$ measured by Hoffmann *et al.* (2013a) for floodplain overbank deposits. By comparison, the average C density of the topsoil layers of

downstream grid cells in the S0 simulation is 12.78 kg C m$^{-2}$, which is an overestimation. Other model uncertainties that
may affect the SOC stocks and POC fluxes include: (1) The absence of increased plant productivity of floodplains, and
transformations between POC, DOC and $CO_2$, and their fate in rivers and streams. Increased plant productivity of
floodplains is shown to contribute significantly to the higher SOC stocks of floodplains compared to hillslopes, and to the
export of DOC and POC to rivers (Van Oost *et al.,* 2012; Hoffmann *et al.,* 2013a).

In a future study we aim to improve the sediment and POC export, and account for a higher floodplain plant productivity
by using a nutrient-enabled version of the ORCHIDEE LSM (Goll *et al.,* 2017).

Furthermore, the model does not take into account the full effects of the selectivity of erosion, often expressed as the
enrichment ratio, where the C content of eroding soil or the deposited sediment can be different from that of the original
soil. The enrichment varies substantially across landscapes, while the importance of erosion selectivity for C is still under
debate (Nadeu *et al.,* 2015; Wang *et al.,* 2010). However, we did a simple sensitivity test to study the effect of C
enrichment by erosion (section 4.3).

CE-DYNAM does not account for different ratios between the SOC pools (active, slow, passive) with depth due to the
limitation in information to constrain these fractions for floodplains and hillslopes. However, this can be potentially
important for respiration of C in depositional sites and during transport. Studies show that the labile C is decomposed first
during sediment transport and directly after deposition, leaving behind the more recalcitrant C in deposition sites (Berhe *et*
*al.,* 2007; Billings *et al.,* 2019). Due to the simplistic nature of our coarse-resolution model and the lack of data on
oxidation of eroded C during transport we did not include C respiration during transport in the model.

The current SOC scheme of CE-DYNAM does also not account for different residence times of SOC as a function of
landscape position along a hillslope. The SOC decomposition rates can vary significantly along a hillslope due to changes
in soil moisture, temperature, aggregation, and the transport of minerals and nutrients (Doetterl *et al.,* 2016). Currently,
these processes are not resolved in coarse resolution LSMs, contributing to the uncertainty in the large-scale linkage
between soil erosion and SOC dynamics.

Furthermore, there is no feedback between soil erosion and plant productivity in the model. To account for this feedback,
soil erosion processes would need to be explicitly included in a LSM, such as ORCHIDEE, which would increase the
computational complexity of the simulations substantially. The lack of this feedback results in an unlimited dynamic
replacement of C on eroding sites.

Currently, the erosion scheme of CE-DYNAM does not include the *L* (slope-length) and *P* (support-practice) factors. This
might induce some bias in the results, especially for agricultural land. In a future study we aim to make CE-DYNAM better
applicable for agricultural land, where these factors play an important role. For this purpose we will focus on the

development of new methods that can quantify the *L* and *P* factors reliably at the global scale, and will need to re-calibrate the Adj.RUSLE model. Our decision of leaving out the *L* and *P* factors from the erosion equation in this study is based on the global study of Doetterl *et al.* (2012), which showed that the *S, R, Cm* and *K* factors explain approximately 78 % of the total erosion rates on cropland in the USA. This indicates that on cropland the *L* and *P* factors, which are related to agriculture and land management, contribute only 22 % to the overall erosion rates. This percentage is comparable to the uncertainty range in the estimation of the *S, R, Cm* and *K* factors at the regional scale from coarse resolution data. Renard and Ferreira (1993) also mention that the soil loss estimates are less sensitive to slope-length than to most other factors. Furthermore, various studies argue that the estimation of the *L* factor for large areas is complicated and thus can induce significant uncertainty in soil erosion rates calculated based on coarse resolution data (Foster *et al.,* 2002; Kinnell, 2007). Especially, for natural landscapes, such as forests, the estimation of the *L* factor is not straightforward as these natural landscapes usually include steep slopes (Elliot, 2004). In order to stay consistent with the estimation of potential soil erosion for all land cover types, we removed the *L* factor from the equation. The Adj.RUSLE has been already successfully validated at the regional scale, without the *L* and *P* factors, where the spatial variability of soil erosion rates compared well to other high-resolution modeling studies and observational data, and where the absolute values fell within the uncertainty ranges of those validation data (Naipal *et al.,* 2015; Naipal *et al.,* 2016; Naipal *et al.,* 2018; and this study). Finally, the aim of this study was to develop and validate a C erosion scheme for applications at the global scale, where the estimation of the *L* and *P* factors is limited. By showing that the erosion rates from the Adj.RUSLE and CE-DYNAM are within the uncertainty of other data and modelling studies, we assume that it will be applicable for other large catchments in the temperate region.

Finally, CE-DYNAM considers only the rather 'slow' rill and interrill soil erosion processes, and does not take into account severe erosion processes such as gully erosion and landslides, which are bound to extreme precipitation events. The daily timestep of CE-DYNAM and the current setup of the sediment budget module allows only for long-term yearly average changes in erosion and deposition rates and cannot be applied to estimate episodic erosion and deposition events.

**4.3 Sensitivity analysis**

We analyzed the effects of the following model assumptions: (1) C enrichment during erosion, (2) the floodplain sediment residence time, and (3) crop residue management.

To test the C enrichment we increased the *EF* parameter (Eq. 15) from 1 to 2, assuming a strong enrichment of C during erosion (section 2.11). We find that this enrichment results in a gross C erosion flux that is 1.61 times larger than the flux without enrichment (Table 7). This leads also to a larger dynamic replacement of C on eroding sites in combination with a larger burial in depositional sites, which is in accordance with the study of Lugato *et al.* (2018). The resulting C sink from the enrichment simulation is 1.25 times larger than the sink under default conditions (Table 7).

To test the potential effects of a different sediment residence time on the SOC dynamics, we performed a sensitivity study where we changed the basin average sediment residence time to be 50 % higher or 50 % lower but kept the maximum sediment residence time at 1500 years (section 2.11). By changing the average sediment residence time and keeping the maximum fixed, the grid cells with the lowest residence times underwent the largest changes in the residence time and consequently in the floodplain SOC storage and export. The higher the residence time, the longer the deposited soil C will reside in the floodplains, where it can either be respired or buried in deeper soil layers. Therefore, we find that the effects of the sediment residence time on the SOC dynamics are non-linear. Under default conditions we find the highest SOC storage. A 50 % higher average sediment residence time leads to the lowest total SOC storage, with a decrease of 30 % compared to default conditions, while the erosional C sink is reduced by 20 % (Table 7). This could be explained by a higher C decomposition flux for floodplains due to the long residence time of C in deposition areas. Especially in mountainous regions where the soil erosion flux is large and removes a large part of the labile C, a higher sediment residence time will lead to higher C emissions due to decomposition in floodplains. The turnover seems to dominate over the C burial in deeper layers and export. A 50 % lower average sediment residence time also leads to a decrease (of 8 %) in the total SOC storage and a decrease of 6 % in the erosional C sink compared to default conditions (Table 7). Also here, the largest changes are found in mountainous regions where a low sediment residence time leads to a large export of C, which is then deposited in lower lying, more extensive floodplains. Thus, increasing or decreasing the residence time leads to a smaller total SOC storage, resulting from different spatial distributions of this SOC storage. The POC flux under the high sediment residence time scenario is substantially higher than under default conditions (Table 7).

To test the effects of crop residue management we harvested all above-ground crop residues (section 2.11). We find that the total litter C stock is about 15 % smaller than the default case by the end of the year 2005. This leads to a total change in the transient SOC stocks that is 20 % smaller under no erosion (S0), and 26 % smaller under erosion (S2) (Table 7). Our findings confirm that soil management practices such as residue management have a substantial effect on the SOC dynamics.

**5 Conclusions**

We presented a novel spatially-explicit and process-based C erosion dynamics model, CE-DYNAM, which simulates the redistribution of soil and C over land as a result of water erosion and estimates the implications for C budgets at catchment scale. We demonstrated that CE-DYNAM captures the spatial variability in soil erosion, C erosion and SOC stocks of the non-Alpine region of the Rhine catchment when compared to high-resolution estimates and observations. We also showed that the quantile ranges of erosion and deposition rates and C stocks fall within the uncertainty ranges of previous estimates at basin or sub-basin level. Furthermore, we demonstrated the model ability to disentangle vertical C fluxes resulting from the redistribution of C over land and develop C budgets that shed light on the role of erosion in the C cycle. The simple structure of CE-DYNAM and the relatively low amount of parameters make it possible to run several simulations to

investigate the role of individual processes on the C cycle such as the removal by erosion only, or the role of sediment deposition and transport. Its compatibility with land surface models makes it possible to investigate the long-term and large-scale effect of erosion processes under various global changes such as increasing atmospheric $CO_2$ concentrations, changes to precipitation and temperature, and land use change.

The application of CE-DYNAM for the Rhine catchment for the period 1850 - 2005 AD reveals three key findings:

- Soil erosion leads to a cumulative net C sink of 216 ± 23 Tg of C by the end of the period, which is in the same order of magnitude as the cumulative land C sink of the Rhine without erosion. This C sink is a result of an increasing dynamic replacement of C on eroding sites due to the $CO_2$ fertilization effect, despite decreasing soil and C erosion rates over the largest part of the catchment. We conclude that it is important to take into account global changes such as climate change in order to better quantify the net effect of erosion on the C cycle.
- After performing a sensitivity analysis on key model parameters we find that the C enrichment by erosion, crop residue management and the residence time of floodplain sediment can substantially change the overall values of C fluxes and SOC storages. However, the main findings, such as soil erosion being a net C sink for the Rhine catchment, remain.
- Initial climate and land cover conditions and the transient period over which erosion under global changes takes place are essential for determining if soil erosion is a net C sink or source and to what extent.

Altogether, these results indicate that despite model uncertainties related to the relative coarse spatial resolution, missing or simplified processes, CE-DYNAM represents an important step forwards into integrating soil erosion processes and sediment dynamics in Earth system models. The next step would be to improve CE-DYNAM with respect to riverine sediment and POC export fluxes and management practices.

**Code and data availability**

The source code of CE-DYNAM is included as a supplement to this paper. Model data can be accessed from the Zenodo repository under the doi:10.5281/zenodo.2642452 . For the other data sets that are listed in table 1, it is encouraged to contact the first authors of the original references.

**Author contributions**

VN built and implemented the model. YW provided the basic structure for the model and performed simulations with the original ORCHIDEE LSM. All authors contributed in the interpretation of the results and wrote the paper.

**Competing interests**

*The authors declare that they have no conflict of interest.*

**Acknowledgements**

Funding was provided by the Laboratory for Sciences of Climate and Environment (LSCE), CEA, CNRS, and UVSQ. Victoria Naipal, Ronny Lauerwald and Philippe Ciais acknowledges support from the VERIFY project that received funding from the European Union's Horizon 2020 research and innovation program under grant agreement No 776810. Philippe Ciais also acknowledges the support from the European Research Council Synergy project SyG-2013-610028 IMBALANCE-P and the ANR CLAND Convergence Institute.Bertrand Guenet acknowledges support from the project ERANETMED2-72-209 ASSESS. We thank Dr. S. Peng for sharing the PFT maps. We also acknowledge the anonymous reviewers for their useful and constructive comments that helped to clarify this manuscript.

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

**Table 1:** Model input datasets

| Dataset | Spatial resolution | Temporal resolution | Period | Source |
|---|---|---|---|---|
| Historical land cover and land use change | 0.25 degrees | annual | 1850 - 2005 | Peng *et al.* (2017) |
| Climate data (precipitation & temperature) for ORCHIDEE | 0.5 degrees | 6 hourly | 1900 - 2012 | CRU-NCEP version 5.3.2; https://crudata.uea.ac.uk/ cru/data/ncep/; last access: 12 February 2020 |
| precipitation for the Adj. RUSLE | 0.5 degrees | monthly | 1850 - 2005 | ISIMIP2b (Frieler *et al.*, 2017) |
| Soil | 1 km | - | - | Global Soil Dataset for Earth System Modeling, GSDE (Shangguan *et al.*, 2014) |
| Topography | 30 arcseconds | - | - | GTOPO30; U.S. Geological Survey, EROS Data Center Distributed Active Archive Center 2004; https://www.ngdc.noaa.gov/mgg/topo/gltiles.html; last access: 12 February 2020 |
| Flow accumulation | 30 arcseconds | - | - | HydroSHEDS (Lehner *et al.*, 2013); https://www.hydrosheds.org/; last access: 12 February 2020 |
| Hillslopes/Floodplain area | 5 arcminutes | - | - | Pelletier *et al.* (2016) |
| River network & stream length | 30 arcseconds | - | - | HydroSHEDS (Lehner *et al.*, 2008) |


**Table 2:** Model simulations, with changes to the basin average gross soil erosion rate (t ha$^{-1}$ y$^{-1}$), the basin average
sediment residence time Tau (years), and the enrichment factor, and the crop residue harvest intensity, RM (%).

| Default simulations | Gross soil erosion | Tau | Enrichment factor | RM |
|---|---|---|---|---|
| S0 | 0 | - | - | 0 |
| S1 | 3.94 | 94 | 1 | 0 |
| S2 | 3.94 | 94 | 1 | 0 |
| **Uncertainty simulations** | | | | |
| S1_min | 1.52 | 94 | 1 | 0 |
| S2_min | 1.52 | 94 | 1 | 0 |
| S1_max | 5.95 | 94 | 1 | 0 |

| | | | | |
|---|---:|---:|---:|---:|
| S2_max | 5.95 | 94 | 1 | 0 |
| **Sensitivity simulations** | | | | |
| S2_Tmin | 3.94 | 60 | 1 | 0 |
| S2_Tmax | 4.94 | 128 | 1 | 0 |
| S1_EF | 5.94 | 94 | 2 | 0 |
| S2_EF | 6.94 | 94 | 2 | 0 |
| S0_RM | 0 | - | - | 100 |
| S1_RM | 3.94 | 94 | 1 | 100 |
| S2_RM | 3.94 | 94 | 1 | 100 |


**Table 3:** Goodness-of-fit results of the comparison of the simulated gross and net erosion rates to those of other studies at
subbasin level, taking into account 13 sub-basins of the Rhine. RMSE is the root mean square error in $10^6$ tons year$^{-1}$. E
stands for soil erosion.

| | E Cerdan *et al.* (2010) | E Germany | E RUSLE2015 | E Borrelli *et al.* (2018) |
|---|---|---|---|---|
| *r-squared* | 0.72 | 0.97 | 0.94 | 0.24 |
| *RMSE* | 0.68 | 1.98 | 0.92 | 1.35 |


**Table 4:** Goodness-of-fit results of the comparison of the simulated gross and net C erosion rates to those of the study of
Lugato *et al.* (2018) in the enhanced and reduced scenario, taking into account 13 sub-basins of the Rhine. RMSE is the
root mean square error in tons year$^{-1}$. Ce stands for gross C erosion, while Cd stands for net C erosion.

| | Ce enhanced | Ce reduced | Cd enhanced | Cd reduced |
|---|---|---|---|---|
| *r-squared* | 0.95 | 0.95 | 0.98 | 0.98 |
| *RMSE* | 7977 | 13797 | 3450 | 9822 |


**Table 5:** This table shows the results of the linear regression between the simulated total SOC stocks (Tg of C per year)
and those of the Global Soil dataset for Earth System Modeling (GSDE) and from the LUCAS database. The regression is
done after aggregating the data at sub-basin level for the 13 sub-basins that were delineated in the Rhine catchment.
RMSE is the root mean square error given in Tg of C per year, while the r-value is the spatial correlation coefficient.

| Regression | r-value | p-value | RMSE |
|---|---|---|---|
| This study versus LUCAS | 0.96 | < 0.01 | 28.69 |

| | | | |
|---|---|---|---|
| This study versus GSDE | 0.95 | < 0.01 | 29.32 |

**Table 6**: This table presents the scaling exponent (b) of equation 20 for floodplains and hillslopes. The scaling exponent was derived for selected points in the Rhine catchment for which measurements on the SOC storage were taken by Hoffmann *et al.* (2013a), and at sub-basin level after the data on area and SOC stocks was aggregated for each of the 13 sub-basins of the Rhine.

| | Scaling exponent floodplains | Scaling exponent hillslopes |
|---|---|---|
| Hoffmann et al. (2013a) | 1.23 ± 0.06 | 1.08 ± 0.07 |
| This study (selected points where measurements were taken) | 1.14 | 0.83 |
| This study (based on the 13 sub-basins) | 1.06 | 1.00 |

**Table 7**: Sensitivity analysis. The impacts of enrichment, changes to the sediment residence time ($\tau min$, $\tau max$), and crop residue management (*RM*) on the cumulative gross C erosion ($C_e$), the cumulative change in the total SOC stock (*dSOC*), the net C sink and the cumulative particulate organic C export flux ($POC_{exp}$) of the Rhine catchment. Units: Tg C

| | $C_e$ | dSOC | C sink | $POC_{exp}$ |
|---|---|---|---|---|
| **Default** | 66 | 142 | 216 | 0.029 |
| **enrichment** | 106 | 198 | 271 | 0.032 |
| **τmin** | 66 | 130 | 204 | 0.026 |
| **τmax** | 66 | 100 | 173 | 0.036 |
| **RM** | 52 | 105 | 194 | 0.031 |


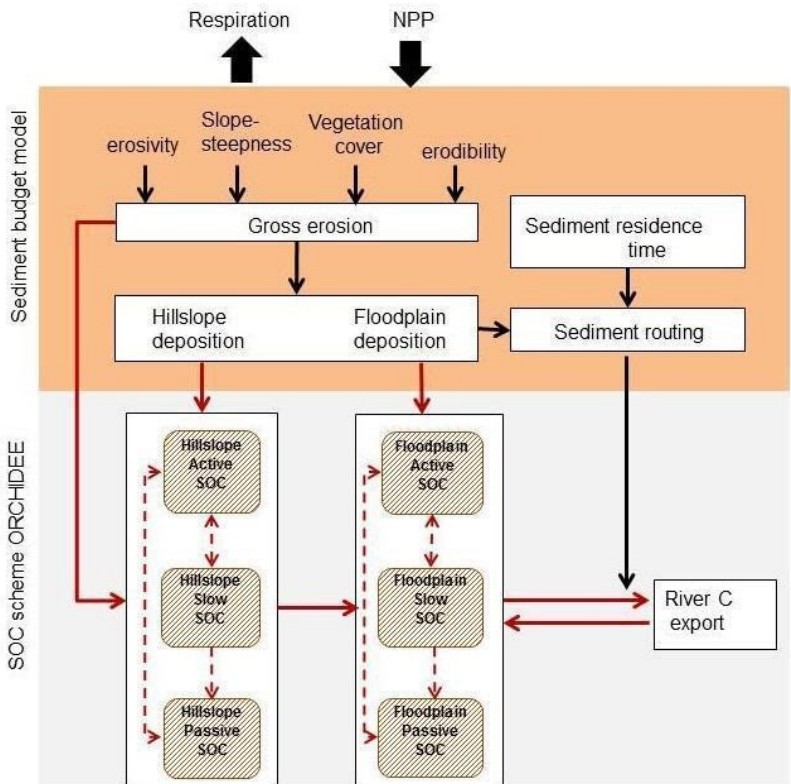

**Figure 1:** A conceptual diagram of CE-DYNAM. The red arrows represent the C fluxes between the C pools/reservoirs,
while the black arrows represent the link between the erosion processes (removal, deposition and transport).

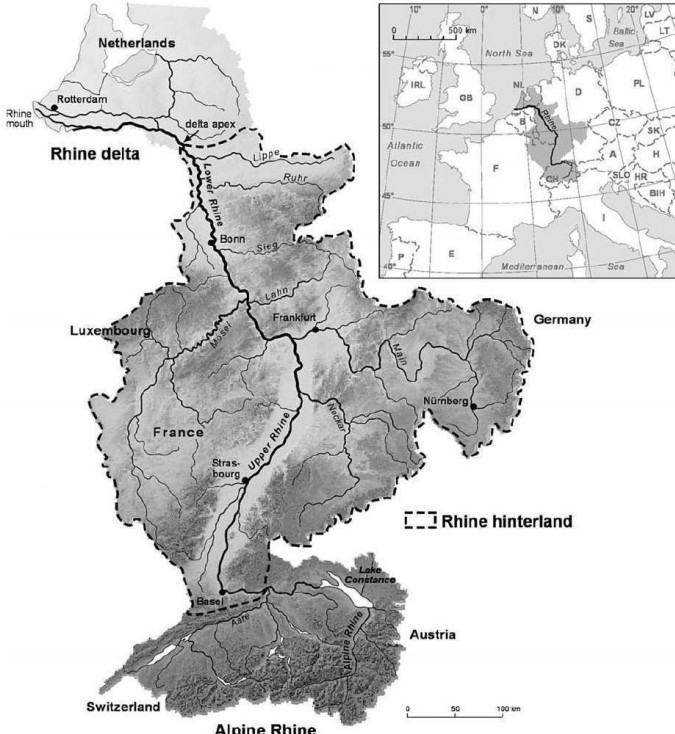

**Figure 2:** The Rhine catchment (Hoffmann et al., 2013), where the gray shades represent elevation and the continuous

black lines the main rivers.

**(A)**                                                                     **(B)**

**(C)**                                                                     **(D)**

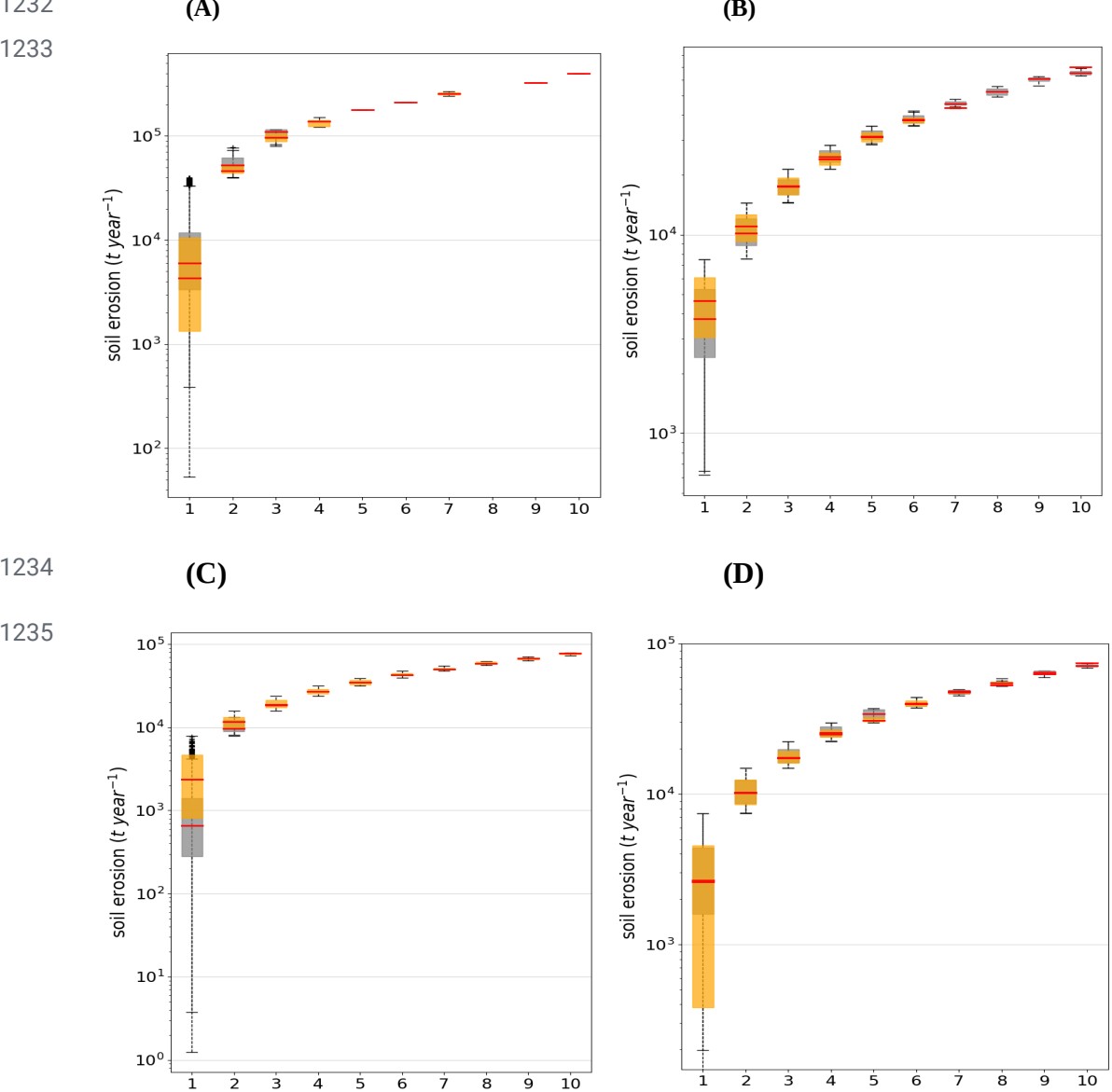

**Figure 3**: Quantile-whisker plot of simulated **gross** soil erosion rates (t/year) (grey whisker boxes), compared to (A) the

study of Cerdan et al. (2010), (B) the study of Panagos et al. (2015), and (C) the German potential erosion map by Bug et

al. (2014) (orange whisker boxes). (D) Quantile-whisker plot of simulated **net** soil erosion rates (t/year) (grey whisker

boxes), compared to the study of Borrelli et al. (2018) (orange whisker boxes). Medians are plotted as red horizontal lines.

The x-axis represents bins or evenly spaced ranges between the minimum and maximum total yearly soil erosion rates of

the Rhine derived from the data of (a) Cerdan et al. (2010), (b) Panagos et al. (2015), (c) Bug et al. (2014), and (d) Borrelli

et al. (2018).

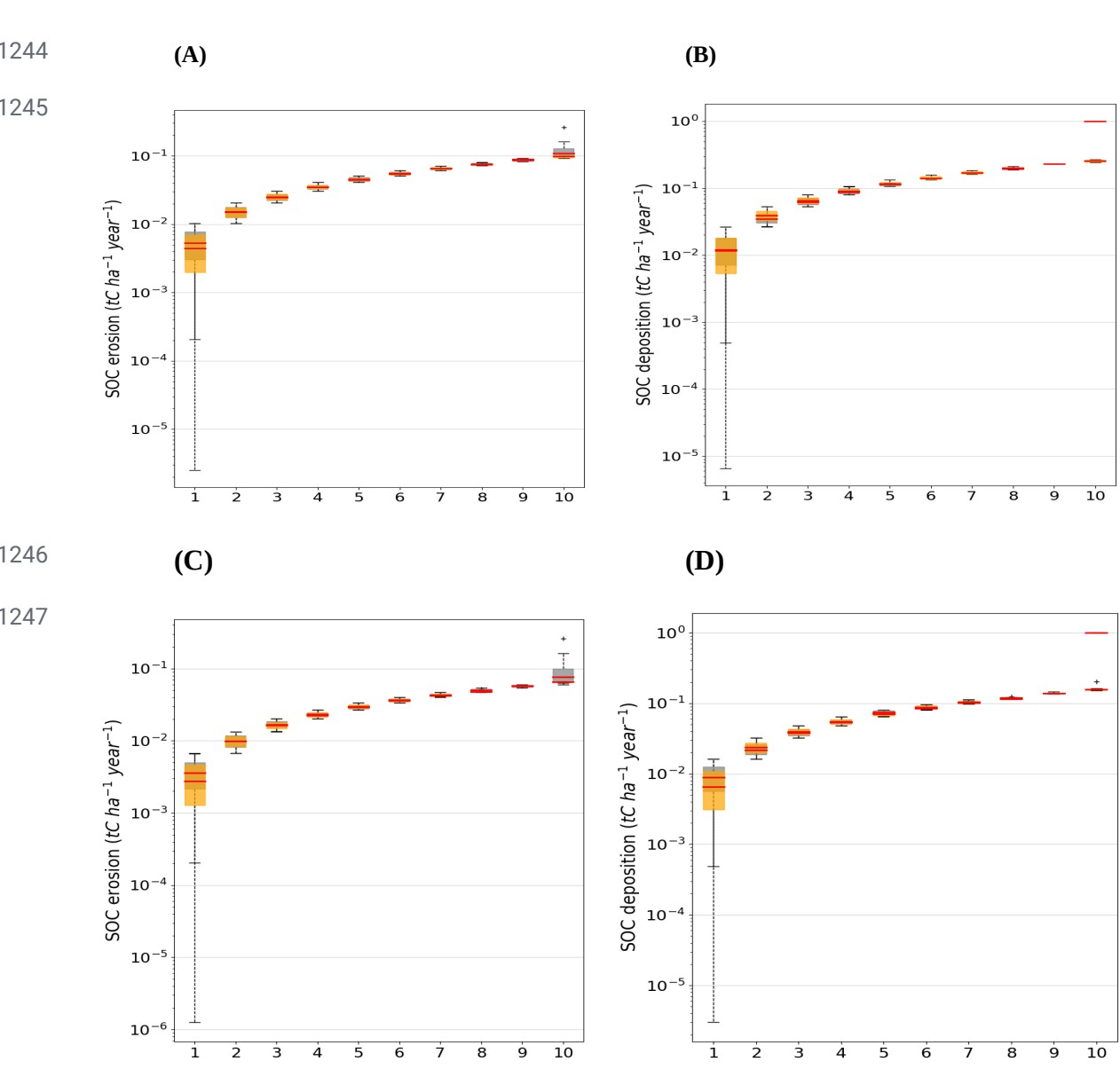

**Figure 4**: (A) Hillslope C erosion rates and, (B) C deposition rates, compared to the enhanced erosion scenario from Lugato et al. (2018). (C) Hillslope C erosion rates and, (D) C deposition rates, compared to the reduced erosion scenario from Lugato et al. (2018). The x-axis represents bins or evenly spaced ranges between the minimum and maximum total yearly soil erosion rates of the Rhine.


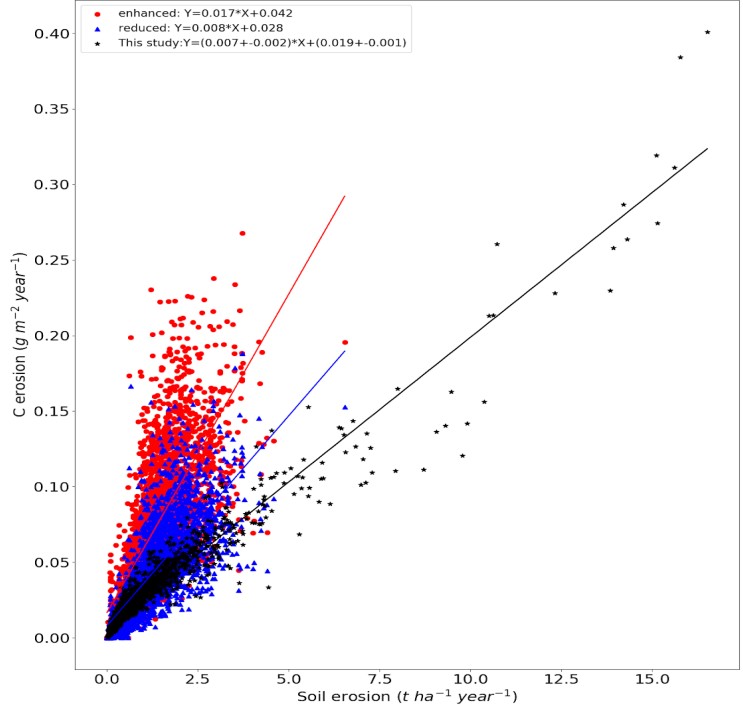

**Figure 5**: The relationship between soil erosion and C erosion of simulation S2 (blackstars) in comparison to the erosion
scenarios from the study of Lugato et al. (2018) with enhanced (red circles) and reduced erosion (blue triangles),
respectively. The straight lines are the trendlines of the linear regression between soil and C erosion.

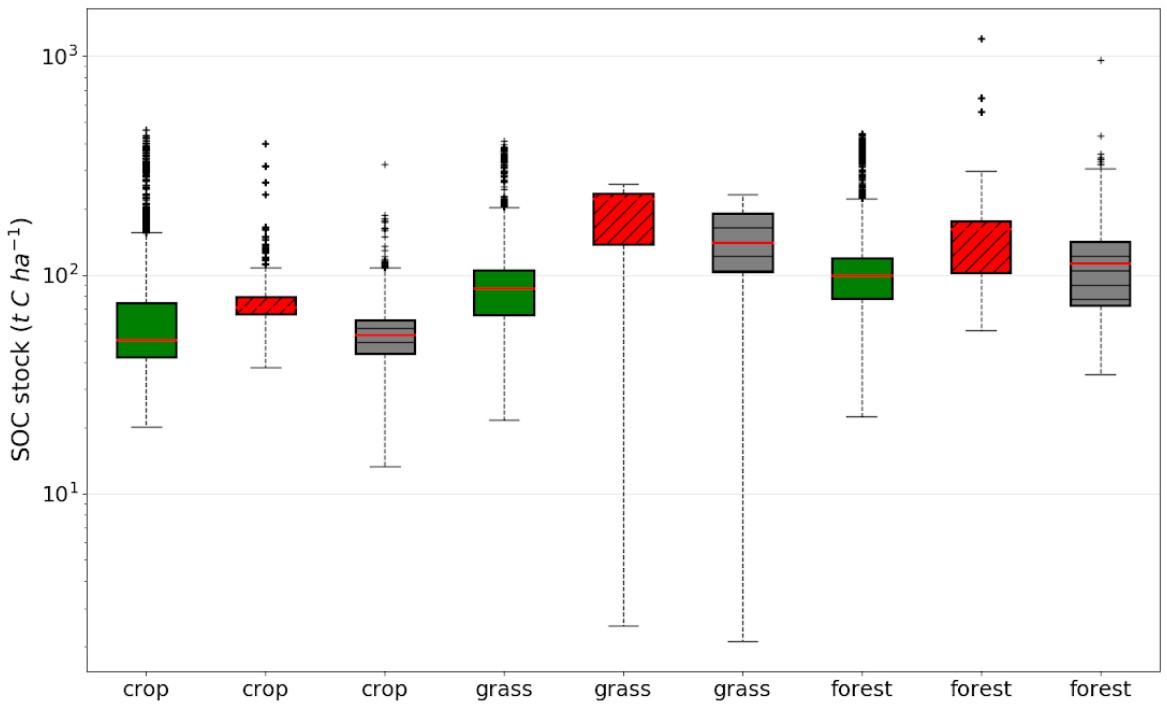

**Figure 6**: Comparison of the total SOC stocks per land cover type between the simulation without erosion (red boxes with
a '//' pattern), the simulation with erosion (black boxes with a '-' pattern) and the LUCAS data (green boxes without
pattern fill). The red horizontal lines are the medians, the dashed vertical lines represent the range between the minimum
and maximum, and the black dots are the outliers.
**(A)**                                                                         **(B)**

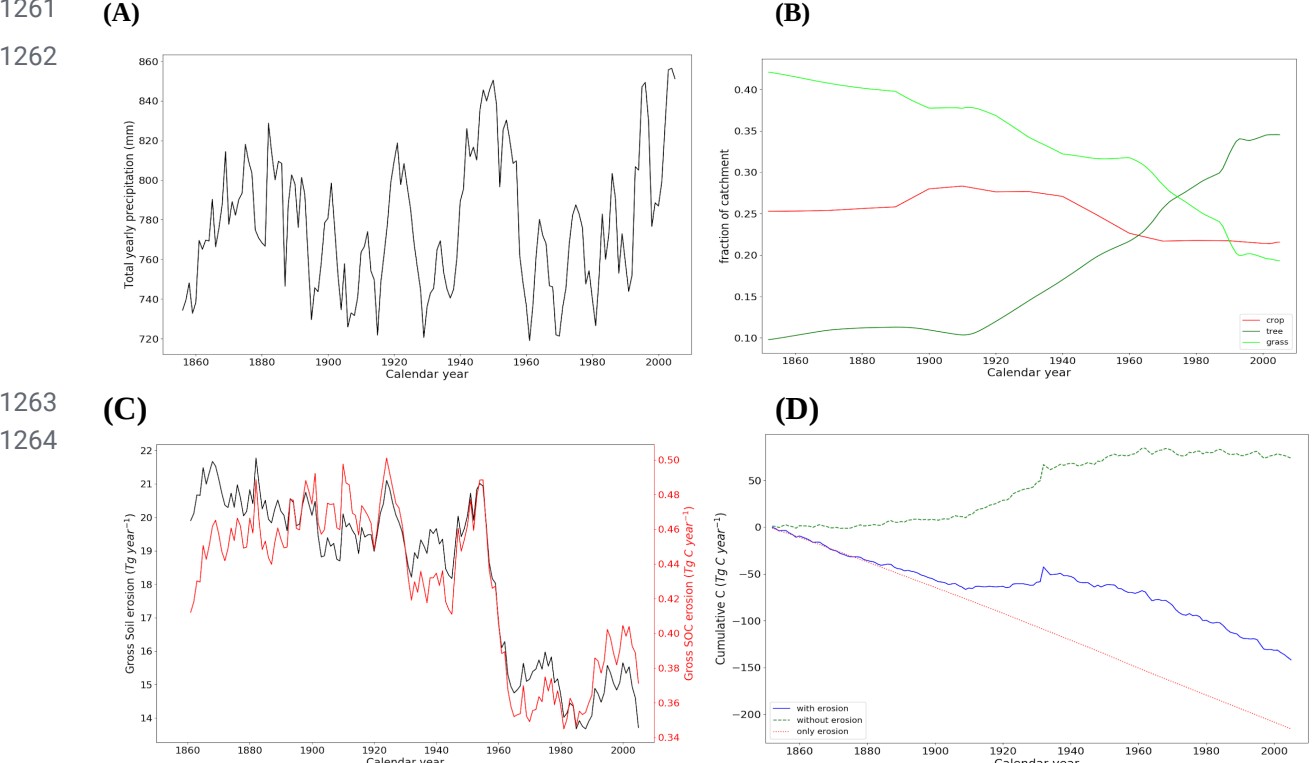

**(C)**                                                                         **(D)**

**Figure 7**: Timeseries of (A) the 5-year average yearly precipitation (mm), (B) changing land cover fractions, (C): 5-year
average total gross soil erosion (Pg year⁻¹) and total gross C erosion rates (Tg C year⁻¹), (D): Cumulative C emissions from
the soil to the atmosphere under land use change and climate change without soil erosion (green dashed line), with soil
erosion (blue straight line), due to additional respiration or stabilization of buried soil and photosynthetic replacement of C
under erosion (Ep, red dotted line). All graphs represent the non-Alpine region of the Rhine catchment.

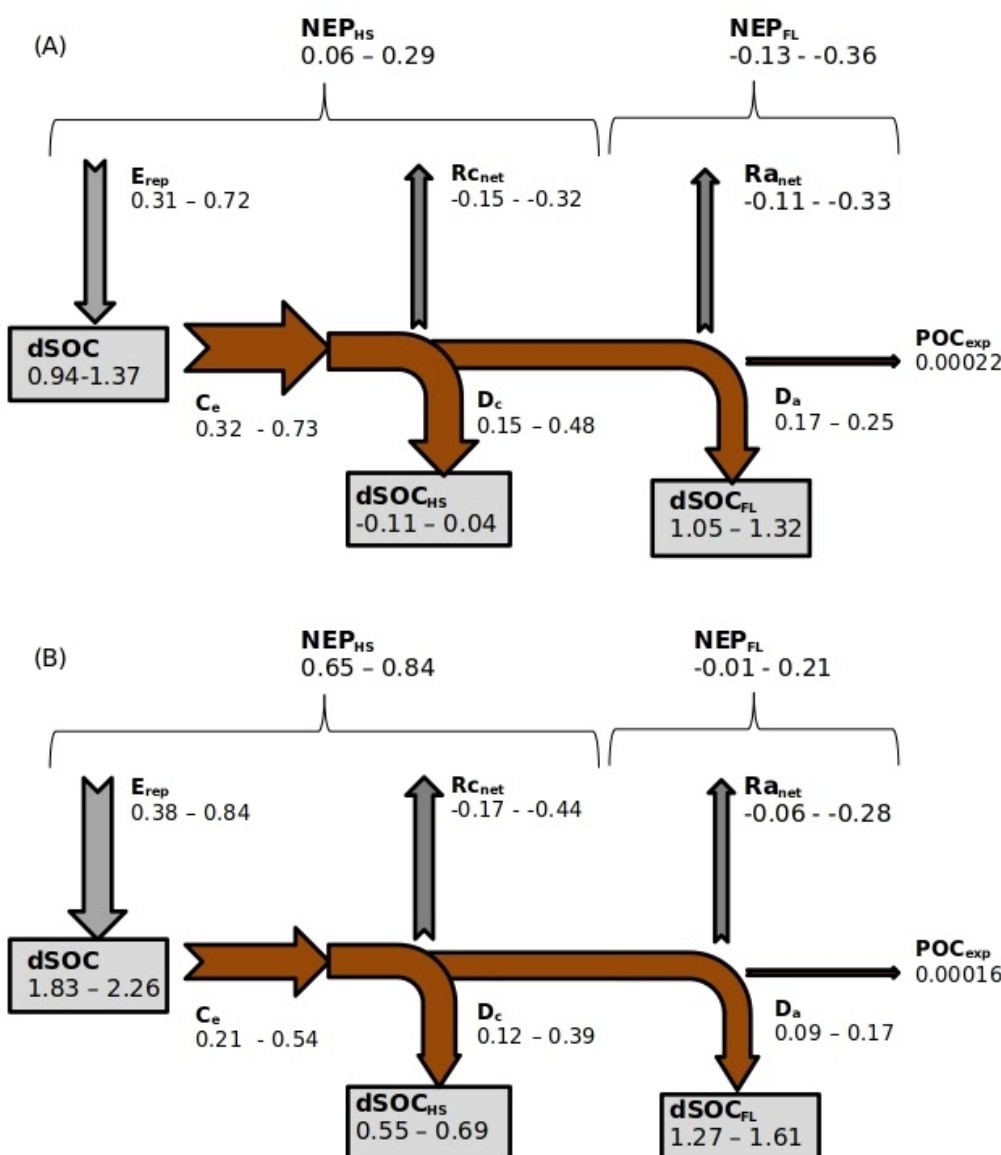

**Figure 8:** (A) C budget of the non-Alpine part of the Rhine for the period 1851-1861, and (B) for the period 1995-2005. The budget shows the net exchange of C (Tg C year$^{-1}$) between the soil and atmosphere as a result of accelerated soil erosion rates. Grey arrows are the erosion-induced yearly average **vertical** C fluxes, while the brown arrows are the erosion-induced yearly average **lateral** C fluxes. $C_e$: Gross C erosion from hillslopes; $D_c$: Deposition of C on hillslopes; $D_a$: Deposition of C in floodplains; $POC_{exp}$: net POC export flux; $E_p$: Erosion-induced C replacement on hillslopes (Eq. 21); $Ra_{net}$: Net respiration/burial of deposited C in floodplains (Eq. 23); $Rc_{net}$: Net respiration/burial of deposited C on hillslopes (Eq. 22); $NEP_{HS}$: Net ecosystem productivity of hillslopes; $NEP_{FL}$: Net ecosystem productivity of floodplains; The grey boxes represent yearly average changes in SOC stocks for the specific time period as a result of land use change, climate

change, erosion and deposition. *dSOC*: Yearly average change in the total SOC stock; *dSOC$_{HS}$*: Yearly average change in
the hillslope SOC stock; *dSOC$_{FL}$*: Yearly average change in the floodplain SOC stock.
**(A)**                                                    **(B)**
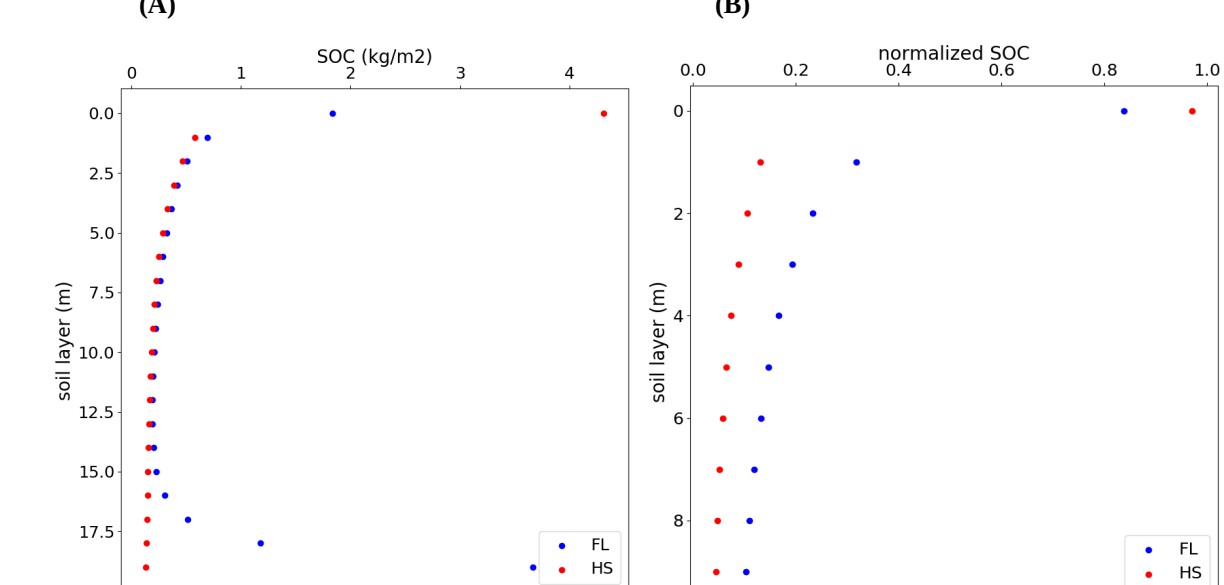
**Figure 9:** (A) Vertical distribution of hillslope (red) and floodplain (blue) SOC stocks (kg m$^{-2}$) with depth averaged over
the non-Alpine region of the Rhine catchment, and (B) the vertical distribution of normalized hillslope (red) and floodplain
(blue) SOC stocks (dimensionless) with depth.