# Peer review of "CE-DYNAM (v1), a spatially explicit, process-based carbon erosion"

_Geoscientific Model Development, 2019_

## Referee Comment (RC1) · Anonymous Referee #1 · 15 Jul 2019

In this study, Naipal and others described the development of a spatially explicit, process-based carbon erosion scheme (CE-DYNAM v1) for the use in ESMs. The leading author has already published several papers on the modeling of soil erosion and its impact on C cycling. Many components described in this manuscript have been well validated in Naipal et al. (2015, 2016 & 2018). Compared with those previous works, the major innovation in this new model is that it links the sediment and C dynamics on hillslopes with those on floodplains together and couples vertical C fluxes and pools from ORCHIDEE with lateral C fluxes from the soil erosion model. The authors designed three numerical experiments (S0: the baseline simulation or no-erosion simulation; S1: the erosion-only simulation; S2: the fully sediment dynamics simulation)

[Figure]

[Figure]
[Figure]

to quantify the effects of carbon erosion on the C cycle in the whole Rhine catchment. This study also performed a detailed model validation of soil erosion, C erosion, sediment storage and SOC stocks using either high-quality data or benchmark simulations in the Rhine catchment.

The manuscript is well written and fits the scope of GMD well. As a novel modeling work, it may advance the represent soil erosion, a process important for land C dynamics, in ESMs to better constrain the C climate feedbacks. The authors have given detailed descriptions of their methods and results in most parts of the manuscript.

My major concern is about the validity of one assumption in the model. I am not fully convinced and expect a better justification. Because the model does not represent the river routing process, it uses floodplain connectivity to simulate the transport of sediment along hydrological pathways. However, by doing so, it implicitly assumes all sediments as sand and gravels (non-cohesive sediment) and represents the transport of cohesive and non-cohesive sediment in the same way. But the cohesive sediments (loam and silt) can be transported by rivers efficiently and most of them would not be deposited. Further, loam and silt may be the major type of sediments that are generated from hillslope erosion (especially for interrill and rill erosion considered by RUSLE). As shown in the results, the current method can cause the severe underestimation of sediment and C that are transported to oceans.

Other Comments

L70-72: These two references are relevant to this sentence.

Galy, V., Peucker-Ehrenbrink, B., & Eglinton, T. (2015). Global carbon export from the terrestrial biosphere controlled by erosion. Nature, 521, 204–207. https://doi.org/10.1038/nature14400

Tan, Z., Leung, L. R., Li, H., Tesfa, T., Vanmaercke, M., Poesen, J., ... Hartmann, J. (2017). A Global data analysis for representing sediment and particulate organic

carbon yield in Earth System Models. Water Resources Research, 53, 10,674–10,700. https:// doi.org/10.1002/2017WR020806

L117: it should be noted that as discussed in Naipal et al. (2015), the formulation of R factor is related to climate type. So in the millennia time scale, one area may need different R factors due to the change of climate.

L170: Reference for Eq. 5? Also, I recommend to show the spatial variability of the f factor in the Rhine catchment.

L192: This may be true for sand and gravel sediment (the majority of floodplain sediment) that Hoffmann et al. (2008) studied. But for cohesive sediment (loam and silt), they can be transported through river channels to oceans without the large fraction of deposition (at least not as large as what is set in this model). They are also the major sediments of soil erosion.

L202: Similar above, this routing scheme may be fine for floodplain but whether it is appropriate for river sediment routing is questionable. And river sediment routing transports large amounts of sediment and POC from hillslopes to oceans.

L322-326: Could you make the meanings of each term in RHS of these equations more clearly? Especially, I do not very understand what the second term of RHS of Eq. 16 stand for. Also in Eq. 17, what is the difference between $1/(\tau*365)$ and kiout for SOCFLi(0,t) in the third term?

L431-432: Or as argued by Tan et al. (2018), rainfall erosivity itself tends to be less variable if using large scale rainfall data to calculate it.

L455: could the map of these 13 sub-basins be shown?

L471-473: if much more sediment was generated but sediment deposition may still follow the long-term level, where did this additional sediment go? I suspect that it mostly was transported to oceans, a process not or poorly represented in the current model.

L474: that only 0.2% of sediment is exported out of the catchment is too low to believe. Are there any data to support it?

Section 4.2: The model also does not represent the impact of water management (such as flooding control) on floodplain connection.

Figures: As discussed above, I recommend to add a few more figures (in either supplementary or appendix) to show the 13 sub-basins of the Rhine catchment and the spatial variability of the floodplain factor f and the sediment residence time $\tau$.

Figure 2: What does the gray level stand for? Elevation?

Figure 3: What does the x-axis mean? Why do not you do a cell-to-cell comparison instead?

Figure 4. Do you have another way to convey the message? It looks messy currently.

References: Generally good. I recommend to also acknowledge the progress in other groups to represent soil erosion at large scale numerical models, such as Pelletier (2012) and Tan et al. (2018).

Pelletier, J. D. (2012), A spatially distributed model for the long-term suspended sediment discharge and delivery ratio of drainage basins, J. Geophys. Res., 117, F02028, doi:10.1029/2011JF002129.

Tan, Z., Leung, L. R., Li, H.-Y., & Tesfa, T. (2018). Modeling sediment yield in land surface and Earth system models: Model comparison, development, and evaluation. Journal of Advances in Modeling Earth Systems, 10, 2192–2213. https://doi. Org/10.1029/2017MS001270.
* * *

---

## Referee Comment (RC2) · Anonymous Referee #2 · 23 Jul 2019

General comments:

The manuscript presents a very interesting approach to model the effects of soil erosion and deposition by water on the C balance of a mesoscale catchment. As it combines techniques from LSMs with an newly developed erosion/deposition model it prepares the ground for large scale modelling. Therefore, I think the paper fits perfectly into the scope of Geoscientific Model Development. However, especially for a model development and testing paper which is building the ground for future environmental / climate analysis I see some major point which need to be addressed. Hence, I suggest major revision for the following reasons:

[Figure]

1. First of all the paper lacks clear aims (or research questions). In Line 68 to 83 the authors give an overview of the contents of the paper, but I think the entire paper would improve substantially if clear aims would be given here. For example, (i) introduce a coupled soil erosion and C turnover model with an LSM model which is applicable on regional scales. (ii) Rigidly test the model for the Rhine Catchment against other modelling results and regionally available data. (iii) Analyze the sensitivity/uncertainty of the model results due to weak input data and a priori model assumptions. (regarding (iii) see comments below.

2. Taking the temporal and spatial scale into account which should be later on analyzed with the model I think the authors found a good balance between model complexity and simplicity. However, the model is full of a priory assumptions, which will fundamentally affect the modelling results, so I personally do not think any model results can be interpreted without some estimates of at least the sensitivity of the model against these assumptions. The most important assumptions which could be tested easily are: C input via plants especially crops depending on erosion status, C enrichment during erosion and depletion during deposition, reduced C turnover in alluvial soils due to wetter conditions, etc. Overall, it is one of the major shortcomings of the paper that the modeling results in section 3.2 are presented single values (e.g. for 159 Tg C for C removal by erosion) and also conclusions based on this single model results are presented. I strongly suggest performing a a sensitivity analyses (including as far as possible effects of a priory assumptions) and giving results with a reasonable range. I am fully aware that it would be hardly possible to do a full uncertainty analysis and even an sensitivity analysis might be quite ambitious given the catchment size and the complexity of the involved models. However, it is not enough just stating in the discussion some important processes are not taken into account.

3. From my understanding of the paper and accounting for the scope of the journal testing such new model against data is essential. The authors did try doing so but here a lot of improvement is easily possible: (i) Include a section under methods explain

which data are used to test the model and also explain in some detail how this is done. For example, the comparison with other models as given in Fig. 3 and 5 is not clear, as the following information is missing: (a) Were the data from the more high resolution models aggregated to the raster cell size of CE-DYNAM to do a raster-by-raster comparison? (b) If a CE-DYNAM raster cell consist of erosional and depositional sites, which are not resolved in the raster cell, how to compare with gross erosion of a high resolution model (e.g. Panagos et al. 2015) which might have different proportions of erosional and depositional raster cells in this large 8 x 8 km$^2$ raster cell. (c) It is not clear at all what is compared as all model results from literature do not focus on the time span from 1850 to 2005. These details are essential for the reader to understand your model validation. (ii) From the figures I have some doubts that the different models fit very well (why not giving statistical goodness-of-fit-parameters?). So the question is how good the other models are (please see e.g. the scientific debate regarding the Panagos et al. (2015) map. So, at least in the discussion this model to model comparison needs to be stressed. (iii) Generally, the erosion (partly deposition) validation of CE-DYNAM is mostly done against other models also using USLE technology (USLE factors might be even derived from same data sources), so an extended discussion if this is meaning full is needed.

Specific comments:

Line 68-83: see general comment.

Line 89: be more explicit regarding 'low number of parameters'

Line 118 ff: I do not agree that not taking the L factor into account is a reasonable decision. I agree that it is somewhat difficult to estimate (for the German part of the Rhine catchment there are some estimates) but if you are interested in land use change it is an essential factor if you kick it out the entire basis of the USLE is set into question. (The P factor is simpler as it is set to 1 in most studies).

Line 130: The statement "...has been calibrated and validated for the Rhine

catchment..." is confusing here? If calibration and validation was already done why doing it again? If the model has changed you need a new validation (but what about calibration? Are you using parameters in CE-DYNAM which were calibrated before it is necessary to indicate this in detail).

Line 113: Alluvial soils are indicated in German soil maps, so the statement is not correct for the largest part of the Rhine catchment.

Line 141 / Eq. 2: Generally I think it would be good being more precise with the equation. For example in case of Eq. 2, I would expect a reference to the different raster cells (Aft(i) = Lstream(i) x Wstream(i); whereas i is the raster cell.) as for other equations e.g. Eq. 4a it was not clear if this refers to the entire catchment is calculated for each raster cell.

Line 148: If alpha and b are constants it means that the upstream area necessary to result in a stream is always the same. I understand that in case of a large scale model simplifications are necessary but this assumptions is for sure not true for the Rhine catchment (see papers from hydrology of maps of the stream system (which by the way would be available for the entire Rhine catchment).

Line 159: '... at 8 km resolution... " I guess this means 8 km x 8 km raster cells. Should be changed throughout the text (also with other resolution given).

Line 163: I do not think the the assumptions of reduced hydrological and geomorphological connectivity in arable landscapes (compared to forest) is correct. From the recent studies dealing with flash floods it is obvious that it is a main problem that this landscapes have a very high connectivity as so many ditches, drainages etc. were built over the last century to get rid of any surplus of water on arable land. So, your assumptions for the range of the parameter f in different landscapes must be underlined by reasonable data.

Line 187: Does a multiple flow algorithm makes sense in case of a resolution of 8 km

x 8 km?

Line 230 ff: In general this is a reasonable assumption for the crop residues. However, studies in small catchment clearly indicate that residue management is a key factor of SOC, so this a priori assumption has potentially a huge effect on the produced results. So, its importance must be analyzed with the model!

Line 238-240: The given equation are a fundamental problem with modelling the effect of soil erosion on SOC turnover. For example, using standard SOC pool residence times for all landscape positions is of tremendous importance for the entire C balance effect of erosion. So, again it would be very important to know how sensitive the results are against this assumptions. At least give some estimates / measurements at different landscape positions in the discussion and comment of the potential effect in modelling results.

Line 265: "... The next soil layer contains less C and therefore at the following time-step les C will be eroded under the same erosion rate..." If this would be always true one would expect a continuous decline in SOC in soils. However, assuming a long-term forest use on a slope you will found the soil in an equilibrium between new C input via plants and small amount of erosion. So, in this case the eroded material will have a more or less constant C content.

Line 277: Calculating a daily erosion fraction is a reasonable approach. However, if taking the episodic nature of erosion and deposition into account the C balance will be different compared to a small continuous process (see literature). Might be also discussed.

Line 291 ff: The assumption that there is no C selectivity (enrichment in eroded material and depletion at erosional sites) is taken in many modelling approaches. However, if there would be no enrichment of fines in the sediments transported in river systems, one would find e.g. sand in suspended sediments of larger rivers. Which is e.g. in case of the lowland Rhine not the case. Discuss this in the context of the scale of your

paper. Also important regarding the loss of C to the ocean.

Line 341: Where do the data regarding afforestation during the last two decades come from. To my knowledge this is a process already started in the late 1959th (please give reference)

Line 379: (see also general comments). I wonder why you did not use other more specific and potentially profound national data. E.g. for Germany there are several maps for potential erosion which are much more elaborated than the map of Panagos et al. (2015). Moreover, I wonder why you did not use the sediment delivery data of the Rhine which are freely available - I guess since the 1950th - which would be a good and reliable additional data set for validation.

Line 397-401: I suggest omitting these sentences and Fig. 4, because I do not see any additional value of this here. It is obvious from the model structure of all USLE based models (and all other erosion models) that an increase of erosivity and slope directly leads to an increase in erosion. Moreover, there is a coincidence in the catchment that highest erosivity and highest slopes occur at the same alpine area, but this is not any proof for the model. Hence, hence I think this is weakening your validation more that it would strengthen it. By the way: Erosivity and slope might explain 70% erosion if very different rainfall regimes and slopes (mountain areas and lowlands) are compared, but with a catchment like the Rhine (where except for the alpine part) the differences in slope and erosivity are relatively small soil cover (C factor) is getting much more important (erosion rates between grassland and arable land vary by a factor of 10-20).

Line 402: As I modeler I expect a goodness-of-fit parameter with this statement. (See also general comments regarding model to model comparison of different USLE implementations).

Line 424 ff: The comparison with the data from Hoffman et al. (2013) underlines a deficit in all your comparisons. It is at no time clear what is compared exactly. Mean of 7500 years against 1850-2005?

Line 438: Does the outflux fit to measured data? Would be easy to test even if this is not essential as only a very small amount will be delivered into the sea. (could be tested at several subcatchment, as data are available).

Line 451-452: This is a clear contradiction to your statement that differences in erosivity are very important for spatial differences in erosion.

Line 453: The close link between C erosion and soil erosion is obvious from your modelling structure but not necessarily correct (C enrichment depending on event size?)

Line 434 ff: See also general comments

Line 446: I suggest not to over interpret the modeling results from the alpine area of the catchment as the modelling and the data are weakest there. (i) Increase in measured precipitation most uncertain; (ii) calculated R factor very uncertain in all USLE approaches; (iii) alpine USLE factors not very well underlined by data (compared to arable and grassland),

Line 497ff. See comments regarding connectivity above. Moreover, even if the connectivity is high under forest (which I doubt), forest will produce not a lot of sediment and hence are not so important for building up alluvial soils at all.

Line 493ff. I do not see from the results the $CO_2$ fertilization plays an important role for an increase in dynamic replacement. I guess that the increase in yields due to changes in management are much more important (as reduced yields are not taken into account at erosional sites) as they boost dynamic replacement of eroded soils.

Line 501ff: It is obvious from first order kinetics that colluvial soils must have higher $CO_2$ effluxes as they contain more C. So, this is not a very new finding.

Line 506-507: Question: Is the modelled increase in respiration from floodplains resulting from an temperature increase or from an increase in depositional material, which would also result in an increase of respiration? Comment: Under real conditions the increase in respiration from floodplains is also a result of decreasing groundwater levels,

diking etc. which are associated with river management. This is shown in local studies but to my knowledge not taken into account in any modelling.

Line 542-561: This is a nice collection of model deficits. However, for a modelling paper I would expect a bit more (see general comments).

Line 576-588: I think this conclusions are not fully supported by the results as the modelled C fluxes might be affected by a priori assumptions and model parameters which are not tested enough (see general comment regarding senistiviy analysis).

Table 1: I guess the spatial resolution is always given in raster cells, e.g. $0.25°$ x $0.25°$

Table 2: As the resolution of the data sets are different how to make sure that the comparison fits, e.g. the higher resolution data set might exclude the river network from the SOC calculation while the lower resolution data set might include this areas into the SOC stock calculation. Give somewhat more details.

Fig. 3 and 5: What are the 10 classes given on the X-Axis?

Fig. 4. Omit this figure (see comment above)

Fig. 5c/d: What does it mean if CE-DYNAM has erosion rates which are up to a factor smaller than the Lugato model and C deposition rates which are more or less the same? Is it a result of different areas affected by erosion and deposition? Should be explained / discussed.

Fig. 7. Just a comment of an handicapped. About 4-8% of the male population are to a certain extend color blind (especially red/green is problematic), so if you do not what to lose these proportion of your readers you should adapted your color in your figures. There are color blind friendly color ranges available in most software packages. If the dashed lines range between min and max the outliers cannot be above or below the lines. So, I guess the lines represent something else.
* * *
[Figure]

2019.

GMDD

Interactive
comment

---

## Author Comment (AC1) · 5 Sep 2019

**Response to Reviewer 1**

We would like to thank the anonymous reviewer for his or her constructive comments. In this response we provide an answer to all the comments and the indicate the changes that will be applied in the revised manuscript.

**Comment 1:** *My major concern is about the validity of one assumption in the model. I am not fully convinced and expect a better justification. Because the model does not represent the river routing process, it uses floodplain connectivity to simulate the transport of sediment along hydrological pathways. However, by doing so, it implicitly assumes all sediments as sand and gravels (non-cohesive sediment) and represents the transport of cohesive and non-cohesive sediment in the same way. But the cohesive sediments (loam and silt) can be transported by rivers efficiently and most of them would not be deposited. Further, loam and silt may be the major type of sediments that are generated from hillslope erosion (especially for interrill and rill erosion considered by RUSLE). As shown in the results, the current method can cause the severe underestimation of sediment and C that are transported to oceans.*

**Answer:** We understand the reviewer's concern regarding the absence of an explicit representation of rivers and river routing in CE-DYNAM. We agree that in this way we treat the transport of all sediments types (cohesive and non-cohesive) in the same way, which can lead to uncertain sediment and POC fluxes carried away by rivers. However, our model assumption does not imply that all sediment are in the form of coarse material, instead, the main assumption is that the majority of the eroded soil and transported sediment is fine sand, silt and loam. This assumption is supported by the fact that the sediment residence time is calculated based on observed floodplain deposit ages of the Rhine (Hoffmann et al. 2007, 2008, 2013). These studies show that most of the deposits in the floodplains are overbank deposits that consist of fine sediment such as sand, loam, silt and clay and organic material. The long residence time (up to 2000 years) that they measured for the floodplains based on the C14 signature of C associated with sediment samples show that the fine sediment can stay buried for a long time in the

floodplains. Although the model lacks explicit river process representations, it reproduces the spatial variability in floodplain sediment and C storage across the Rhine sub-basins as is shown by table 3 of this manuscript and by a previous study where we validated the global sediment budget model (Naipal et al., 2016, ESD). It should be noted that the model has been developed and calibrated to simulate long-term changes in sediment and carbon storage on land and not the short-term variations in sediment and POC fluxes carried by rivers.

Finally, the model produces a sediment export flux at the end of the year 2005 of $1.6 \times 10^7$ tonnes per year, which is a magnitude higher than the measured suspended sediment flux of about $3.15 \times 10^6$ tonnes per year (Asselman et al.,2003). The higher sediment flux is the result of absent riverine processes in CE-DYNAM such as sediment burial behind dams, and the fact that we assume an equilibrium state for the Rhine catchment based on the period 1850-1860 where agricultural soil erosion rates were already high. The simulated total cumulative sediment export of 2.5 Gt for the Rhine over the period 1850-2005 is about 36 % of the cumulative gross soil erosion flux of 6.8 Gt. This sediment flux leads to a cumulative POC export of about 0.14 Tg of C for the Rhine over the period 1850-2005 (based on a new simulation S2, see more details in the following paragraph below). This is 0.2 % of the cumulative C erosion flux. The yearly POC flux at the end of the year 2005 is 0.02 tC $km^2$ $year^{-1}$ (normalized over the total basin area), which is an order of magnitude lower compared to other studies who found 0.9 tC $km^2$ $year^{-1}$ (Beusen et al., 2009; Soribas et al., 2016).

This underestimation in POC in CE-DYNAM is most likely a result of the high sediment residence time of floodplains downstream of the Rhine and the absence of increased plant productivity of floodplains, leading to the decomposition of a large fraction of the deposited C. We tested the effect of the sediment residence time on the resulting lateral C fluxes of the model and find that they do not change the POC export of the Rhine significantly (see our detailed response to comment 2 of reviewer 2). Increased plant productivity of floodplains is shown to contribute significantly to the higher SOC stocks of floodplains compared to hillslopes, and to the export of DOC and POC to rivers (Van Oost et al., 2012; Hoffmann et al., 2013).

***Changes to the manuscript:*** We will address the model uncertainty related to the POC export and include the above mentioned findings in section 4.1 on the limitations of the model.

New transient simulation S2 based on an improved equilibrium state

We redid the simulation S2 for the Rhine catchment using a different model spin-up. In the old spin-up we let the model run continuously for 2000 years, whereas in the new spin-up we ran the model for 3000 years and calculated analytically the temporary equilibrium state of the floodplain SOC pools every 10 years. This new spin-up method resulted in the floodplain SOC pools being close to equilibrium at the end of the 3000 year spin-up period, where the yearly change in the floodplain SOC stocks was less than 0.001% of the total floodplain SOC stock. Therefore, it was not needed to subtract the additional increase in the SOC stocks resulting from the disequilibrium state from those of the transient simulation (see section 2.11). The new transient simulation S2 resulted in different absolute values for the C budget of the Rhine. However, the main conclusions did not change. We also performed an uncertainty analysis with a minimum and maximum soil erosion scenario, based on the uncertainty ranges in the rainfall erosivity and land cover factors of the Adjusted RUSLE model. The revised manuscript will contain the new figures and tables. In addition, section 3 will be modified to include the new results with uncertainty ranges.

**Specific comments**

**Comment S1:** *L70-72: These two references are relevant to this sentence.*

*Galy, V., Peucker-Ehrenbrink, B., & Eglinton, T. (2015). Global carbon export from the terrestrial biosphere controlled by erosion. Nature, 521, 204–207.https://doi.org/10.1038/nature14400*

*Tan, Z., Leung, L. R., Li, H., Tesfa, T., Vanmaercke, M., Poesen, J., ... Hartmann, J. (2017). A Global data analysis for representing sediment and particulate organic C carbon yield in Earth*

*System Models. Water Resources Research, 53, 10,674–10,700. https:// doi.org/10.1002/2017WR020806*

**Answer:** We will add these references in the revised manuscript

**Comment S2:** *L117: it should be noted that as discussed in Naipal et al. (2015), the formulation of R factor is related to climate type. So in the millennia time scale, one area may need different R factors due to the change of climate.*

**Answer:** This is right. In the paper of Naipal et al. (2016), where the global sediment budget model is applied for the last millennium, we take the change in climate in the calculation of the erosivity into account. For this study, we assume that the climate zones as defined by the Koeppen-Geiger climate classification have not changed drastically since 1850 AD.

***Changes to manuscript***: We will mention this assumption in the methods section

**Comment S3:** *L170: Reference for Eq. 5? Also, I recommend to show the spatial variability of the f factor in the Rhine catchment.*

**Answer:** This equation has been adopted from the study of Naipal et al. (2016), that presents the global sediment budget model for the Rhine. We will include the reference to this equation in the revised manuscript and add the spatial variability of the *f* factor in the supplementary document.

[Figure]

Supplementary figure: Spatial variability of the floodplain deposition factor (f)

**Comment S4:** *L192: This may be true for sand and gravel sediment (the majority of floodplain sediment) that Hoffmann et al. (2008) studied. But for cohesive sediment (loam and silt), they can be transported through river channels to oceans without the large fraction of deposition (at least not as large as what is set in this model). They are also the major sediments of soil erosion.*

**Answer:** See our answer to the previous comment, where we argue that most of the floodplain sediment studied by Hoffmann et al. (2008) consists mostly out of organic material (gyttja, peat) and fine sediments (fine sand, loam, silt) in overbank deposits (see table 2 in Hoffmann et al., 2008). These fine sediments are a result of long-term soil erosion on the hillslopes. Also a large part has been transported and deposited in the floodplains under major storms, such as the one in the 14[th] century (Bork et al., 2003). In this study (Table 3, section 3.1) and a previous study on the millennial sediment storage of the Rhine (Naipal et al., 2016) we show that by getting the scaling relationships as found by Hoffmann et al. (2013) right, the sediment residence time is realistic.

To show the potential effects of a different sediment residence time on the SOC storage and POC flux, we performed a sensitivity study where we changed the basin average sediment residence time to be 50% higher or 50% lower but keeping the maximum sediment residence time at 1500

years. We find that the low sediment residence scenario leads to a 43% higher cumulative C export flux of the Rhine catchment, while the high sediment residence scenario leads to a 15% lower export flux compared to default conditions. However, the impacts of a modified sediment residence time on the total SOC storage of the Rhine are non-linear. The results of this sensitivity study will be summarized in table 5 in the discussion section of the revised manuscript. See changes to the manuscript in our response to reviewer 2, where we describe in the model sensitivity analysis in more detail.

**Comment S5:** *L202: Similar above, this routing scheme may be fine for floodplain but whether it is appropriate for river sediment routing is questionable. And river sediment routing transports large amounts of sediment and POC from hillslopes to oceans.*

**Answer:** See our response to comments 1 and S1

**Comment S6:** *L322-326: Could you make the meanings of each term in RHS of these equations more clearly? Especially, I do not very understand what the second term of RHS of Eq. 16 stand for. Also in Eq. 17, what is the difference between $1/(\tau *365)$ and kiout for SOCFLi(0,t) in the third term?*

**Answer:** The second term at the RHS of Eq. 16 stands for the C flux flowing into soil layer z from the soil layer z+1 below, and is related to the C export flux of the floodplain part of a grid cell. When the topsoil layer loses C due to sediment routing, the C from the subsoil layer 'moves' upward as is also done for C loss due to soil erosion (section 2.7). In Eq. 17 $ki_{out}$ stands for the C import rate from the neighboring grid cells. We will provide a short explanation of each term in the equations 16 and 17 in the revised manuscript.

**Comment S7:** *L431-432: Or as argued by Tan et al. (2018), rainfall erosivity itself tends to be less variable if using large scale rainfall data to calculate it.*

*Answer:* We agree with this statement and will add a comment in the revised manuscript mentioning the effect of the spatial resolution.

**Comment S8:** *L455: could the map of these 13 sub-basins be shown?*

*Answer:* We will include a map of the sub-basins of the Rhine catchment in the supplementary information.

**Comment S9:** *L471-473: if much more sediment was generated but sediment deposition may still follow the long-term level, where did this additional sediment go? I suspect that it mostly was transported to oceans, a process not or poorly represented in the current model.*

**Answer:** We agree that a large part of the sediment is transported out of the catchment, more specifically 36% of the cumulative gross soil erosion rates over the entire period (see our response to the first comment). We aim to explicitly represent riverine processes in a future study on the further development of CE-DYNAM where we also plan to include the impact of dams on the sediment export. However, the focus of this study lies on the redistribution of soil and C on land and their effect on the land-atmosphere C exchange, rather than on the riverine export fluxes of sediment and C.

**Comment S10:** *L474: that only 0.2% of sediment is exported out of the catchment is too low to believe. Are there any data to support it?*

**Answer:** See our answer to comment 1

**Comment S11:** *Section 4.2: The model also does not represent the impact of water management (such as flooding control) on floodplain connection.*

*Answer:* This is correct. We assume a 'natural' state of the catchment where the main river channel is not managed and the floodplains are more or less dynamic. We will specify this in the revised manuscript.

***Changes to manuscript:***

**Section 4.1:** In CE-DYNAM we assume a 'natural' state of the catchment where the main river channel is not managed and the floodplains are more or less dynamic. This may affect the behaviour of the POC export and residence time of C in floodplains.

**Comment S12:** *Figures: As discussed above, I recommend to add a few more figures (in either supplementary or appendix) to show the 13 sub-basins of the Rhine catchment and the spatial variability of the floodplain factor f and the sediment residence time τ.*

**Answer:** We added these figures in the supplementary info

[Figure]
 Supplementary figure: Spatial variability of the floodplain sediment residence time (tau)

**Comment S13:** *Figure 2: What does the gray level stand for? Elevation?*

**Answer:** The gray level stands for elevation, where the darker colors represent higher elevations. We added this information in the figure caption of the revised manuscript.

**Comment S14:** *Figure 3: What does the x-axis mean? Why do not you do a cell-to-cell comparison instead?*

**Answer:** The x-axis represents bins or evenly spaced ranges between the minimum and maximum total yearly soil erosion rates of the Rhine. A cell-to-cell comparison does not show a clear result due to the large variability in erosion rates. We find a quantile plot like figure 3 more useful to see for which erosion ranges the rates differ significantly between the models. For example, from Fig. 3A we can see that for the higher soil erosion rates (bins 2-10) CE-DYNAM produces a lower average compared to the erosion data of Cerdan et al. (2010).

**Changes to manuscript:** We revised the figure caption to include the information on the bins.

**Comment S15:** *Figure 4. Do you have another way to convey the message? It looks messy currently.*

**Answer:** We agree that this figure does not convey the message properly, after reviewer 2 had a similar opinion. We also think that the figure is not very important. However, we will keep the text where we explain that the spatial variability of soil erosion rates are dominated by erosivity and slope as is shown by similar studies using a higher model resolution.

**Changes to manuscript:** We will remove this figure from the manuscript

**Comment S16:** *References: Generally good. I recommend to also acknowledge the progress in other groups to represent soil erosion at large scale numerical models, such as Pelletier (2012) and Tan et al. (2018).*

**Changes to manuscript:** We will acknowledge these studies in the introduction and will add the references in the revised manuscript.

---

## Author Comment (AC2) · 5 Sep 2019

**Response to Reviewer 2**

We would like to thank the anonymous reviewer for his or her constructive comments. In this response we provide an answer to all the comments and then indicate the changes that will be applied in the revised manuscript.

**Comment 1:** *First of all the paper lacks clear aims (or research questions). In Line 68 to 83 the authors give an overview of the contents of the paper, but I think the entire paper would improve substantially if clear aims would be given here. For example, (i) introduce a coupled soil erosion and C turnover model with an LSM model which is applicable on regional scales. (ii) Rigidly test the model for the Rhine Catchment against other modelling results and regionally available data. (iii) Analyze the sensitivity/uncertainty of the model results due to weak input data and a priori model assumptions. (regarding (iii) see comments below.*

**Answer:** We will clarify the aims of our study in the Introduction section of the revised manuscript. See potential changes below.

**Changes to manuscript**:

**L76-92:** To address these knowledge gaps, we present a parsimonious process-based Carbon Erosion Dynamics Model (CE-DYNAM), which integrates sediment dynamics resulting from water erosion with the SOC dynamics at the regional scale. The SOC dynamics are calculated consistently with drivers of land use change, $CO_2$ and climate change by a process-based land surface model, with a simplified reconstruction of the last century increase of crop productivity. This modelling approach consists of a global sediment budget model coupled to the SOC removal, input, and decomposition processes diagnosed from the ORCHIDEE global land surface model (LSM) in an offline setting (Naipal et al., 2018). The main aim of our study is to quantify the horizontal transport of sediment and C along the continuum of hillslopes, floodplains and rivers, and at the same time analyze its impacts on the land-atmosphere C exchange. We calibrate and validate the new model with regional observations and high-resolution modelling results of the Rhine catchment. It should be noted here that the

structure of CE-DYNAM is designed in a way that the model can be adapted easily to other large catchments and finally run globally. Finally, we also discuss the model uncertainties and the sensitivity of the model to changes in key parameters and assumptions made. In the next sections we give a detailed overview of CE-DYNAM model structure, the coupling of erosion, deposition and transport with the coarse-resolution SOC dynamics of ORCHIDEE, model application and validation for the Rhine catchment, and its potentials and limitations.

**L83-88** goes to the discussion section where we also discuss the potential of CE-DYNAM: to combine erosion, transport, and re-deposition of soil material, for which small scale differences in topography are of utter importance, with a state-of-the-art representation of large-scale SOC dynamics driven by land use, climate, and atmospheric $CO_2$ as simulated by the ORCHIDEE LSM. The flexible structure of CE-DYNAM makes the model adaptable to the SOC dynamics of any other LSM. By coupling soil erosion with the C dynamics of LSMs it is possible to study the main processes behind the linkages of soil erosion and the global C cycle.

**Comment 2:** *Taking the temporal and spatial scale into account which should be later on analyzed with the model I think the authors found a good balance between model complexity and simplicity. However, the model is full of a priori assumptions, which will fundamentally affect the modelling results, so I personally do not think any model results can be interpreted without some estimates of at least the sensitivity of the model against these assumptions. The most important assumptions which could be tested easily are: C input via plants especially crops depending on erosion status, C enrichment during erosion and depletion during deposition, reduced C turnover in alluvial soils due to wetter conditions, etc. Overall, it is one of the major shortcomings of the paper that the modeling results in section 3.2 are presented single values (e.g. for 159 Tg C for C removal by erosion) and also conclusions based on this single model results are presented. I strongly suggest performing a a sensitivity analyses (including as far as possible effects of a priory assumptions) and giving results with a reasonable range. I am fully aware that it would be hardly possible to do a full uncertainty analysis and even an sensitivity analysis might be quite ambitious given the catchment size and the complexity of the involved*

*models. However, it is not enough just stating in the discussion some important processes are not taken into account.*

**Answer:** We agree that an uncertainty analysis is important for a regional modelling study such as ours. Therefore, we performed additional simulations with a minimum and maximum soil erosion scenario, based on the uncertainty ranges in the rainfall erosivity and land cover factors of the Adjusted RUSLE model. Chapter 3 of the revised manuscript will be modified to include the new uncertainty results. We will also modify figure 9 to include the uncertainty ranges in the C budget components.

Regarding the sensitivity analysis of the model we tested the assumption of C enrichment during erosion as suggested by the reviewer. Here, we performed two additional simulations with an enrichment factor of two adapted from the study of Lugato et al. (2018): S1_EF (erosion only) and S2_EF (erosion with deposition and transport). We also tested the rate of C transport between floodplains by letting the basin average sediment residence time to vary between a 50% lower and 50% higher value compared to the default. For this purpose we did another two additional simulations (S2_Tmin and S2_Tmax). However, we abstained from testing the model performance to a changed C turnover in alluvial soils as a result of wetter conditions. Previous studies show that there are still large uncertainties related to the turnover of C in depositional environments, and more specifically of alluvial soils, as they represent complex soil profiles with a wide range in physical, chemical and biological parameters that affect the C turnover in interaction with climatic variables such as soil moisture. For example, the studies of Doetterl et al. (2018) and Rasmussen et al. (2018) show that the C turnover of alluvial soils is determined by C stabilization affected by the availability of minerals (such as Iron, Aluminium) and nutrients, mediated by soil microbes and by the formation of peat deposits on river banks. Yet, old alluvial soils can be far from water-saturated, in which case the C turnover would not be substantially decreased as a result of additional oxygen limitation. In our study we also include floodplains that do not get flooded regularly. Therefore, it is not clear if these alluvial soils are in general 'wetter' than the colluvial soils and would therefore have a significantly different C turnover. Also, our model does not include a good representation of groundwater dynamics and a soil

moisture function for alluvial soils. After performing an extensive literature study on C turnover in alluvial soils we could not find a way to easily but realistically modify the C turnover of alluvial soils, for example by using a simple turnover reduction factor derived from observations. We will introduce a new section 4.3 in chapter 4, where we will discuss the results from the sensitivity simulations, see changes to manuscript below.

*Changes to manuscript:*

**Section 4.3: Sensitivity analysis**

The carbon enrichment factor (EF) represents a higher C concentration in eroded soil compared to the original soil, due to the selectivity of erosion (Bertol et al., 2017). By increasing the EF from 1 to 2, we assume a strong enrichment of C during erosion and find that this enrichment results in a gross C erosion flux that is 1.5 times larger compared to the flux without enrichment. This leads also to a larger dynamic replacement of C on eroding sites in combination with a larger burial in depositional sites, which is in accordance with the study of Lugato et al. (2018). The resulting C sink from the enrichment simulation is 1.5 times larger than the sink under default conditions. The cumulative POC flux during enrichment is somewhat lower (Table 5).

To show the potential effects of a different sediment residence time on the C dynamics, we performed a sensitivity study where we changed the basin average sediment residence time to be 50% higher or 50% lower but keeping the maximum sediment residence time at 1500 years. By changing the average sediment residence time and keeping the maximum fixed, it will be the grid cells with the lowest residence times that will undergo the largest changes in residence time and consequently in the floodplain SOC storage and export. The higher the residence time, the longer the deposited soil C will reside in the floodplains, where it can either be respired or buried in deeper soil layers. Therefore, we find that the effects of the sediment residence time on the SOC dynamics are non-linear. Under default conditions we find the highest SOC storage. A 50% higher average sediment residence time leads to the lowest total SOC storage, with a decrease of 20% compared to default conditions. Here, the erosional C sink is reduced by 19% (Table 5). This could be explained by a higher C decomposition flux for floodplains due to the long residence time of C in deposition areas. Especially, in mountainous regions where the soil

erosion flux is large and removes a large part of the labile C, a higher sediment residence time will lead to higher C decomposition emissions in floodplains. The turnover seems to dominate over the C burial in deeper layers and export. A 50% lower average sediment residence time also leads to a decrease (of 9%) in the total SOC storage and a decrease of 8% in the erosional C sink compared to default conditions. Also here, the largest changes are found in the mountainous regions where a low sediment residence time leads to a large export of C, which is then deposited in lower lying, more extensive floodplains. Thus, increasing or decreasing the residence time leads to a smaller total SOC storage, resulting from different spatial distributions of this SOC storage (this will be shown in a Supplementary figure). The POC flux under the low sediment residence time scenario is substantially higher than under default conditions (Table 5).

|  | Ce | dSOC_tot | C sink/source | POC |
|---|---|---|---|---|
| Default | 181 | 152 | 163 | 0.138 |
| enrichment | 269 | 230 | 240 | 0.137 |
| тmin | 181 | 139 | 150 | 0.198 |
| тmax | 181 | 121 | 132 | 0.117 |

*Table 5: Sensitivity analysis. The impacts of enrichment and changes to the sediment residence time on the cumulative gross C erosion (Ce), the cumulative change in the total SOC stock (dSOC$_{tot}$), the net C sink and the cumulative particulate organic C export flux (POC) of the Rhine catchment. Units: Tg C*

**L297, Eq 15:** $k_E = \frac{f*(\frac{E}{365})}{BD*dz} * EF$

*Where EF is the enrichment factor, set to 1 by default.*

**Section 2.11:** We also performed 4 additional sensitivity simulations and 4 additional uncertainty simulations. Simulation S1_EF and S2_EF are performed to test the model assumption of a C enrichment during erosion. Here, we changed the enrichment factor EF to two, based on the study of Lugato et al. (2018). Simulations S2_Tmin and S2_Tmax are performed to test the rate of C transport between floodplains. Here we modified the average sediment residence time for

the Rhine catchment to a minimum of 60 years (50 % lower than the current value), and to a maximum of 128 years (50% higher than the current value), respectively. However, we kept the maximum sediment residence time at 1500 years.

For the uncertainty analysis we performed simulations S1_min and S2_min with a minimum soil erosion scenario, and S1_max and S2_max with a maximum soil erosion scenario. These soil erosion scenarios are based on the uncertainty ranges in the rainfall erosivity and land cover factors of the erosion model. See the supplementary material for information on how these uncertainty ranges are derived. All the model simulations are summarized in table 2.

| Default simulations | Soil erosion | Tau | Enrichment |
|---|---|---|---|
| S0 | 0 | | |
| S1 | 3.94 | 94 | 1 |
| S2 | 3.94 | 94 | 1 |
| Uncertainty simulations | | | |
| S1_min | 1.52 | 94 | 1 |
| S2_min | 1.52 | 94 | 1 |
| S1_max | 5.95 | 94 | 1 |
| S2_max | 5.95 | 94 | 1 |
| Sensitivity simulations | | | |
| S2_Tmin | 3.94 | 60 | 1 |
| S2_Tmax | 4.94 | 128 | 1 |
| S1_EF | 5.94 | 94 | 2 |
| S2_EF | 6.94 | 94 | 2 |

*Table 2: Model simulations, with changes to the basin average gross soil erosion rate (t ha$^{-1}$ y$^{-1}$), the basin average sediment residence time (years), and the enrichment factor.*

***L471-475:*** We find an average annual soil erosion rate of 4.66+- 2.22 t ha$^{-1}$ year$^{-1}$ over the period 1850-2005, which is about two times larger than the average erosion rate simulated for the last millennium (Naipal et al., 2016) and about four times larger than the average erosion rate of the Holocene (Hoffmann et al., 2013). This soil erosion flux mobilized around 181+-117 Tg of C over the period 1850-2005, of which 35+-9 % is deposited in colluvial reservoirs, 65+-9 % is deposited in alluvial reservoirs and 0.08 % is exported out of the catchment.

**L495-498:** We find that the cumulative C erosion removal flux of 181+-68 Tg of C leads to a cumulative net C sink of 163+- 31 Tg C for the whole Rhine region (Fig 8E). This is about 1.3 – 1.9 % of the cumulative NPP and about 35-51% of the cumulative land C sink of the Rhine without erosion.

**Comment 3:** *From my understanding of the paper and accounting for the scope of the journal testing such new model against data is essential. The authors did try doing so but here a lot of improvement is easily possible: (i) Include a section under methods explain which data are used to test the model and also explain in some detail how this is done. For example, the comparison with other models as given in Fig. 3 and 5 is not clear, as the following information is missing: (a) Were the data from the more high resolution models aggregated to the raster cell size of CE-DYNAM to do a raster-by-raster comparison? (b) If a CE-DYNAM raster cell consist of erosional and depositional sites, which are not resolved in the raster cell, how to compare with gross erosion of a high resolution model (e.g. Panagos et al. 2015) which might have different proportions of erosional and depositional raster cells in this large 8 x 8 km 2 raster cell. (c) It is not clear at all what is compared as all model results from literature do not focus on the time span from 1850 to 2005. These details are essential for the reader to understand your model validation. (ii) From the figures I have some doubts that the different models fit very well (why not giving statistical goodness-of-fit-parameters?). So the question is how good the other models are (please see e.g. the scientific debate regarding the Panagos et al. (2015) map. So, at least in the discussion this model to model comparison needs to be stressed. (iii) Generally, the erosion (partly deposition) validation of CE-DYNAM is mostly done against other models also using USLE technology (USLE factors might be even derived from same data sources), so an extended discussion if this is meaning full is needed.*

**Answer:** To better clarify the model validation and comparison against data and other models, we will include an additional section 2.12 in the revised manuscript where we discuss the data used, and how the validation is done in more detail. In this new section we will also discuss

reasons why we do this model to model comparison, where we will give more background information on the various models. Finally, we provide a statistical goodness-of-fit summary by comparing the total soil erosion rates of sub-basins of the Rhine . See changes to manuscript below.

***Changes to manuscript***:

[revised manuscript text omitted]

***Section 3.1:*** We also compared the total gross soil erosion rates of 17 sub-basins of the non-Alpine region of the Rhine. We find that the spatial distribution of the gross soil erosion rates compare well to the findings of other studies. A summary of the statistical goodness of fit between the soil erosion rates is given in the following table.

| | E Cerdan et al. | E Auerswald et al. | E RUSLE2015 |
|---|---|---|---|

| r-squared | 0.72 | 0.97 | 0.94 |
|---|---|---|---|
| RMSE | 0.68 | 1.98 | 0.92 |

Table 3: A goodness-of-fit summary of gross soil erosion rates (E) from CE-DYNAM and other studies. The RMSE is the root mean square error give in tons*1E-6

**Specific comments:**

**Comment S1:** *Line 68-83: see general comment.*

*Answer:* See our response to general comment 1

**Comment S2:** *Line 89: be more explicit regarding 'low number of parameters'*

***Changes to manuscript:*** We will modify this part of the sentence to: "… 3) the low number of parameters compared to other carbon erosion models that operate at a high spatial resolution (Lugato et al., 2018; Billings et al., 2019), which allows running the model at large spatial scales…."

**Comment S3:** *Line 118 ff: I do not agree that not taking the L factor into account is a reasonable decision. I agree that it is somewhat difficult to estimate (for the German part of the Rhine catchment there are some estimates) but if you are interested in land use change it is an essential factor if you kick it out the entire basis of the USLE is set into question. (The P factor is simpler as it is set to 1 in most studies).*

*Answer:* We agree that leaving both the L and P factors out of the equation will induce some bias in the results, especially for agricultural land. In our next study we aim to make CE-DYNAM better applicable for agricultural land, where these factors play an important role. For this purpose we will focus on the development of new methods that can quantify the L and P factors reliably at the global scale, and will need to re-calibrate the erosion module of CE-DYNAM, the Adj.RUSLE. Our decision of leaving out the L and P factors from the erosion equation in our study is based on the global study of Doetterl et al. (2012), which showed that the S, R, C and K factors explain approximately 78% of the total erosion rates on cropland in the USA. This

indicates that on cropland the L and P factors, which are related to agriculture and land management, contribute only for 22 % to the overall erosion rates. This percentage is comparable to the uncertainty range in the estimation of the S, R, C and K factors at the regional scale from coarse resolution data. Renard and Ferreira (1993) also mention that the soil loss estimates are less sensitive to slope length than to most other factors.

Furthermore, various studies argue that the estimation of the L factor for large areas is complicated and thus can induce significant uncertainty in soil erosion rates calculated based on coarse resolution data  (Foster et al., RUSLE2 user guide; Kinnell, 2007). Especially, for natural landscapes, such as forest, the estimation of the L factor is not straightforward as these natural landscapes usually include steep slopes (Elliot, 2004). In order to stay consistent with the estimation of potential soil erosion for all land cover types, we remove the L factor from the equation. The Adj.RUSLE has been already successfully validated at the regional scale, without the L and P factors where the spatial variability of soil erosion rates compares well to other high resolution modeling studies and observational data and the absolute values fall within the uncertainty ranges of those validation data (Naipal et al., 2015; Naipal et al., 2016; Naipal et al., 2018; and this study). Finally, the aim of this study was to develop and validate a carbon erosion module for applications at the global scale, where the estimations of the L and P factors is even more limited. By showing that the erosion rates from the Adj.RUSLE and CE-DYNAM are within the uncertainty of other data and modelling studies, we can assume that it will be applicable for other large catchments in the temperate region.

***Changes to manuscript:*** In section 4.2 'Model limitations' of the revised manuscript, we will address the lack of the L and P factors in more detail in the same way as described above.

**Comment S4:** *Line 130: The statement ". . .has been calibrated and validated for the Rhine catchment. . ." is confusing here? If calibration and validation was already done why doing it again? If the model has changed you need a new validation (but what about calibration? Are you using parameters in CE-DYNAM which were calibrated before it is necessary to indicate this in detail).*

*Answer:* In our study we use the model parameters of the global sediment budget model as defined and calibrated by Naipal et al. (2016), such as the sediment residence time or the floodplain deposition factor. We did not perform an additional calibration of the sediment dynamics part of CE-DYNAM, only a validation, because of the use of different input datasets.

*Changes to manuscript:* We will add a sentence in the revised manuscript specifying why we redid the validation.

**Comment S5:** *Line 113: Alluvial soils are indicated in German soil maps, so the statement is not correct for the largest part of the Rhine catchment.*

*Changes to manuscript:* We will modify the sentence to: 'It should be noted that global soil databases do not identify floodplain soil as a separate soil class, although national soil databases might. However, the aim of this study is to present a carbon erosion model that should be also applicable for other catchments and eventually, globally. Therefore, we followed a 2-step methodology to derive floodplains in the Rhine catchment using hydrological parameters and existing data on hillslopes and valleys.'

**Comment S6:** *Line 141 / Eq. 2: Generally I think it would be good being more precise with the equation. For example in case of Eq. 2, I would expect a reference to the different raster cells (Aft(i) = Lstream(i) x Wstream(i); whereas i is the raster cell.) as for other equations e.g. Eq. 4a it was not clear if this refers to the entire catchment is calculated for each raster cell.*

*Changes to manuscript:* We will modify the equations accordingly in the revised manuscript.

**Comment S7:** *Line 148: If alpha and b are constants it means that the upstream area necessary to result in a stream is always the same. I understand that in case of a large scale model simplifications are necessary but this assumptions is for sure not true for the Rhine catchment (see papers from hydrology of maps of the stream system (which by the way would be available for the entire Rhine catchment).*

*Answer:* We agree, that they might not be the same for the entire catchment, and variations exist. But these constants have been derived from 467 cross-sections of the Rhine catchment combining 1:25 000 geological maps and catchment area extracted from the SRTM 3arcsec digital elevation model (Hoffmann et al., 2007).

**Comment S8:** *Line 159: '. . . at 8 km resolution. . . " I guess this means 8 km x 8 km raster cells. Should be changed throughout the text (also with other resolution given).*

*Answer:* We will modify this accordingly in the revised manuscript

**Comment S9:** *Line 163: I do not think the the assumptions of reduced hydrological and geomorphological connectivity in arable landscapes (compared to forest) is correct. From the recent studies dealing with flash floods it is obvious that it is a main problem that this landscapes have a very high connectivity as so many ditches, drainages etc. were built over the last century to get rid of any surplus of water on arable land. So, your assumptions for the range of the parameter f in different landscapes must be underlined by reasonable data.*

*Answer:* This assumption is underlied by several studies (Hoffmann et al., 2013; de Moor and Verstraeten, 2008; Gumiere et al., 2011;Wang et al., 2014) on the effect of erosion on sediment yield, where is shown that man-made activities on agricultural landscapes result in a trapping of eroded soil in colluvial deposition sites, reducing the sediment transport from hillslopes to the floodplains. The model parameter *f* has been calibrated for the Rhine catchment before in Naipal et al. (2016), where this range is found to produce a ratio between hillslope and floodplain sediment storage that was comparable to observations. The studies of Wang et al. (2010; 2014) identify a range for the hillslope sediment delivery to be between 50 and 80 %, which is similar to the range in the (1-f) factor of our model.

**Changes to manuscript:** We will better underline the choice for the *f* parameter in the revised manuscript as described above.

**Comment S10:** *Line 187: Does a multiple flow algorithm makes sense in case of a resolution of 8 km*

**Answer:** The multiple flow algorithm is especially effective in hilly regions, and we expect it to work better than the single flow algorithm for these regions. In such steep landscapes the river courses are meandering a lot. So the downstream part of a river can easily cut through the boundary of two downstream lying adjacent 8 km x 8 km cells. With the coarse resolution of 8 km, a single direction algorithm would lead to an extreme straightening of the river network, and it would underestimate the number of cells which have a proportion of floodplain area. However, we agree that the coarse resolution grid size will affect the results of both algorithms.

**Comment S11:** *Line 230 ff: In general this is a reasonable assumption for the crop residues. However, studies in small catchment clearly indicate that residue management is a key factor of SOC, so this a priori assumption has potentially a huge effect on the produced results. So, its importance must be analyzed with the model!*

**Answer:** We agree that the harvest and crop residues left on the field would have a large effect on the SOC dynamics of agricultural landscapes. It should be noted that we implemented an increase in the harvest index during the period 1850-present-day based on the study of Hay (1995), which already partly accounts for crop residue management. To explicitly quantify the potential impacts of crop residue management we performed an additional sensitivity simulation where we assumed that all above-ground crop residues are harvested. After running CE-DYNAM to equilibrium we find that the total crop litter C stock is about 70% smaller under the extreme crop residue management scenario. This leads to a total SOC stock (under steady state conditions) that is 60% smaller under no erosion (S0), and 35% smaller under erosion (S2). The transient sensitivity simulations considering crop residue management are still running. However, the preliminary results confirm that soil management practices such as residue management have a substantial effect on the SOC dynamics. We will present these findings in section 4.3 of the revised manuscript, sensitivity analysis.

**Changes to manuscript:** Table 2 will be modified to include the new sensitivity simulations and the new section 4.3 will include the discussion on the results of the effects of crop residue management.

**Section 2.11: Input data and model simulations**

We performed 3 additional sensitivity simulations (S0_RM, S1_RM, S2_RM) where we test the model assumption on crop residue management. In these simulations we assume that all the above-ground crop residues are continuously harvested.

**Section 4.3: Sensitivity analysis**

Here, we investigate the effect of reduced C input into the soil by an crop residue harvesting. After performing sensitivity simulations where all the above-ground crop residues have been harvested, we find that under steady state conditions the total crop litter C stock is about 70% smaller compared to the default case. This leads to a total SOC stock that is 60% smaller under no erosion (S0), and 35% smaller under erosion (S2). Our findings confirm that soil management practices such as residue management have a substantial effect on the SOC dynamics.

*Comment S12:* *Line 238-240: The given equation are a fundamental problem with modelling the effect of soil erosion on SOC turnover. For example, using standard SOC pool residence times for all landscape positions is of tremendous importance for the entire C balance effect of erosion. So, again it would be very important to know how sensitive the results are against this assumptions. At least give some estimates / measurements at different landscape positions in the discussion and comment of the potential effect in modelling results.*

*Answer:* The SOC pool residence times at different landscape positions (hillslope, depositional sites etc), is interrelated with weathering, soil erosion and sediment transport processes (Berhe et al., 2008). To be able to have different residence times for each SOC pool as a function of the landscape position, soil erosion effects on for example the aggregation of soil particles and transport of minerals and nutrients has to be included. This can currently not be done in global land surface models. However, there are efforts to change the current SOC dynamics scheme of

LSMs by introducing measurable SOC pools (Abramoff et al., 2018). In this case it might be possible to calibrate the residence time of each pool based on the landscape position.

**Changes to manuscript:**

**Section 4.2: Model limitations**

The current SOC scheme of CE-DYNAM does not account for different residence times of SOC as a function of landscape position along a hillslope. The SOC decomposition rates can vary significantly along a hillslope due to changes in soil moisture, temperature, aggregation, and the transport of minerals and nutrients (Burke et al., 1995; Doetterl et al., 2016). Currently, these processes are not resolved in coarse resolution LSMs, contributing to the uncertainty in the large-scale linkage between soil erosion and SOC dynamics.

*Comment S13: Line 265: ". . . The next soil layer contains less C and therefore at the following time-step les C will be eroded under the same erosion rate. . ." If this would be always true one would expect a continuous decline in SOC in soils. However, assuming a long- term forest use on a slope you will found the soil in an equilibrium between new C input via plants and small amount of erosion. So, in this case the eroded material will have a more or less constant C content.*

*Answer:* We understand that this sentence might be unclear. What we mean is that the reduction of C erosion at each timestep due to less C being available for erosion, and the existence of a compensatory C sink due to the erosional removal of C, will ultimately lead to an equilibrium state. Which is also reached in our model.

**Changes to manuscript:** We add the following sentences: " The removal of C by erosion also triggers a compensatory C sink due to the reduction in SOC respiration on eroding land. This compensatory C sink and reduced C erosion over time will ultimately lead to an equilibrium state."

*Comment S14: Line 277: Calculating a daily erosion fraction is a reasonable approach. However, if taking the episodic nature of erosion and deposition into account the C balance will be different compared to a small continuous process (see literature). Might be also discussed.*

*Answer:* We will mention this aspect in the discussion chapter of the revised manuscript.

*Comment S15:* *Line 291 ff: The assumption that there is no C selectivity (enrichment in eroded material and depletion at erosional sites) is taken in many modelling approaches. However, if there would be no enrichment of fines in the sediments transported in river systems, one would find e.g. sand in suspended sediments of larger rivers. Which is e.g. in case of the lowland Rhine not the case. Discuss this in the context of the scale of your paper. Also important regarding the loss of C to the ocean.*

*Answer:* We agree and performed a sensitivity analysis of the model where we changed the C enrichment factor to 2 , hereby, partly accounting for the selectivity of erosion. We find that although the POC export to the ocean is not significantly affected, the SOC storage and resulting C sink is increased. See our detailed response to general comment 2.

*Comment S16:* *Line 341: Where do the data regarding afforestation during the last two decades come from. To my knowledge this is a process already started in the late 1959[th] (please give reference)*

*Answer:* We use the land use change data from the study of Peng et al. (2017), who bases their estimates of forest cover on Houghton (2003,2008) for the period 1850-1990 and satellite data for the recent decades. The historical national forest area used by Houghton et al. since 1850 are from national surveys and they are arguably the best data available, although uncertainty arises when downscaling historical forest area change on a grid, using (uncertain) gridded reconstruction of agricultural land (HYDE) and land use transitions rules (see details for the method used by Peng et al. 2017).

*Comment S17:* *Line 379: (see also general comments). I wonder why you did not use other more specific and potentially profound national data. E.g. for Germany there are several maps for potential erosion which are much more elaborated than the map of Panagos et al. (2015). Moreover, I wonder why you did not use the sediment delivery data of the Rhine which are freely*

*available - I guess since the 1950th - which would be a good and reliable additional data set for validation.*

*Answer:* We did not use the sediment delivery data of the Rhine, because the comparison to our simulated coarse resolution model results will most likely be not entirely justified. In our model we do not take into account daily changes in precipitation and runoff and how that affects the erosion rates and sediment transport. Instead we use yearly totals. We also do not take into account dams and other man-made structures that would affect the river transport of sediment. Further, we focus only on the rill and interrill erosion and do not account for other soil erosion processes and flash floods that might have a larger effect on the sediment delivery. Finally, our model has not been developed to simulate the river transport of sediment and C, but instead is focused on the redistribution of soil on land and the resulting sediment and SOC storage. See also our response to the general comment of reviewer 1. For the future development of CE-DYNAM we aim to better represent the river transport processes of sediment and C.

Regarding the validation of soil erosion with observations, we used the database of Cerdan et al. (2010), which already includes German national estimates on soil erosion. We also performed a comparison of our agricultural soil erosion model estimates to the agricultural soil erosion map of Auerswald et al. (2009), available at 250 m resolution, in the revised manuscript (see figure below).

***Changes to manuscript:***

**Section 2.11:** For the validation of gross soil erosion rates we also used the German national estimates on agricultural water erosion from Auerswald et al. (2009). This national data was available at a resolution of 250 m and is based on standardized erosion measurements from 27 studies and 1067 plot years.

**Section 3.1:** Inclusion of an additional figure (fig 3d) to show the comparison of our simulated agricultural soil erosion rates to the German map on agricultural soil erosion  (Auerswald et al., 2009)

[Figure]

figure 3d: Quantile-whisker plot of simulated gross agricultural soil erosion rates (t/year) (grey whisker boxes), compared to (A) national estimated of agricultural soil erosion in Germany (Auerswald et al., 2009)

***Comment S18:*** *Line 397-401: I suggest omitting these sentences and Fig. 4, because I do not see any additional value of this here. It is obvious from the model structure of all USLE based models (and all other erosion models) that an increase of erosivity and slope directly leads to an increase in erosion. Moreover, there is a coincidence in the catchment that highest erosivity and highest slopes occur at the same alpine area, but this is not any proof for the model. Hence, hence I think this is weakening your validation more that it would strengthen it. By the way: Erosivity and slope might explain 70% erosion if very different rainfall regimes and slopes (mountain areas and lowlands) are compared, but with a catchment like the Rhine (where except for the alpine part) the differences in slope and erosivity are relatively small soil cover (C factor) is getting much more important (erosion rates between grassland and arable land vary by a factor of 10-20).*

***Answer:*** We agree and will omit figure 4 in the revised manuscript.

***Comment S19:*** *Line 402: As I modeler I expect a goodness-of-fit parameter with this statement. (See also general comments regarding model to model comparison of different USLE implementations).*

***Answer:*** We will include a goodness-of-fit summary related to the soil erosion result comparison, see our response to comment 3.

***Comment S20:*** *Line 424 ff: The comparison with the data from Hoffman et al. (2013) underlines a deficit in all your comparisons. It is at no time clear what is compared exactly. Mean of 7500 years against 1850-2005?*

***Answer:*** See the last paragraph in our response to general comment 3 and modifications to the manuscript.

***Comment S21:*** *Line 438: Does the outflux fit to measured data? Would be easy to test even if this is not essential as only a very small amount will be delivered into the sea. (could be tested at several subcatchment, as data are available).*

**Answer:** See our response to general comment 1 of reviewer 1

***Comment S22:*** *Line 451-452: This is a clear contradiction to your statement that differences in erosivity are very important for spatial differences in erosion.*

***Answer:*** We agree and will remove this sentence in the revised manuscript.

***Comment S23:*** *Line 453: The close link between C erosion and soil erosion is obvious from your modelling structure but not necessarily correct (C enrichment depending on event size?)*

***Changes to manuscript:*** We will add the following sentence to this statement: "It should be noted that the correlation between soil and C erosion might be affected by processes not properly captured by the model such as the selectivity of erosion, which may also include the enrichment of C in eroded material."

**Comment S24:** *Line 434 ff: See also general comments*

**Answer:** We address this in our response to the general comments

**Comment S25:** *Line 446: I suggest not to over interpret the modeling results from the alpine area of the catchment as the modelling and the data are weakest there. (i) Increase in measured precipitation most uncertain; (ii) calculated R factor very uncertain in all USLE approaches; (iii) alpine USLE factors not very well underlined by data (compared to arable and grassland),*

**Answer:** We are aware of this bias and, therefore, presented model results without the Alpine region (see for example figures 8 and 9).

**Changes to manuscript:** We will add a sentence in the revised manuscript in the same section mentioning these biases for the Alpine region.

**Comment S26:** *Line 497ff. See comments regarding connectivity above. Moreover, even if the connectivity is high under forest (which I doubt), forest will produce not a lot of sediment and hence are not so important for building up alluvial soils at all.*

**Answer:** We agree that soil erosion in forests is minimal but forests also appear often in hilly landscapes that will contribute to the sediment production. When analyzing soil erosion rates over timescales longer than a few decades, extensive forest areas will contribute significantly to the overall removal of soil. Forests contain also a lot of SOC, and so minimal rates of soil erosion might be still significant for the SOC dynamics, also in depositional areas, as is demonstrated in the recent study of Billings et al. (2019).

**Comment S27:** *Line 493ff. I do not see from the results the CO2 fertilization plays an important role for an increase in dynamic replacement. I guess that the increase in yields due to changes in management are much more important (as reduced yields are not taken into account at erosional sites) as they boost dynamic replacement of eroded soils.*

**Answer:** We agree that increased yields due to management boost the dynamic replacement of eroded soil but the largest effect comes from the $CO_2$ fertilization due to increased atm. $CO_2$

concentrations. See below figure 1 representing the actual C replacement under land use change and climate change, and figure 2 representing the potential C replacement under a fixed climate (no change in temperature, precipitation, CO2 atm. concentrations, but with a changing land use and management).

[Figure]

*Fig 1 & 2: C replacement on eroding soils*

**Comment S28:** *Line 501ff: It is obvious from first order kinetics that colluvial soils must have higher CO2 effluxes as they contain more C. So, this is not a very new finding.*

*Answer:* We agree that this might be an obvious finding, but this has not been quantified for such a large catchment before. With this finding we also indicate that the model reproduces process knowledge from field work.

**Comment S29:** *Line 506-507: Question: Is the modelled increase in respiration from floodplains resulting from an temperature increase or from an increase in depositional material, which would also result in an increase of respiration? Comment: Under real conditions the increase in respiration from floodplains is also a result of decreasing groundwater levels,*

*Answer:* In this case the increase in respiration of floodplains is mainly due to the additional deposition of material. In our model we have not a specific representation of ground water for floodplains.

**Changes to manuscript:** We will clarify this in the revised manuscript.

***Comment S30:*** *Line 542-561: This is a nice collection of model deficits. However, for a modelling paper I would expect a bit more (see general comments).*

***Answer:*** See our response to the general comments and resulting improvements made to the manuscript.

***Comment S31:*** *Line 576-588: I think this conclusions are not fully supported by the results as the modelled C fluxes might be affected by a priori assumptions and model parameters which are not tested enough (see general comment regarding sensitivity analysis).*

***Answer:*** We will add a 4th key finding in the conclusions related to the results of the sensitivity analysis.

***Changes to manuscript:*** After performing a sensitivity analysis on key model parameters we find that the C enrichment by erosion, crop residue management and a different spatial variability of the residence time of floodplain sediment can substantially change the overall values of C fluxes and SOC storages. However, the main findings, such as soil erosion being a net C sink for the Rhine catchment, are not changed.

***Comment S32:*** *Table 1: I guess the spatial resolution is always given in raster cells, e.g. 0.25 ◦ x 0.25 ◦.*

***Answer:*** We will change this in the revised manuscript

***Comment S33:*** *Table 2: As the resolution of the data sets are different how to make sure that the comparison fits, e.g. the higher resolution data set might exclude the river network from the SOC calculation while the lower resolution data set might include this areas into the SOC stock calculation. Give somewhat more details.*

***Answer:*** We give more details on the resolution and comparison in the new section 2.12 of the revised manuscript. See our response to comment 3.

***Comment S34:*** *Fig. 3 and 5: What are the 10 classes given on the X-Axis?*

***Answer:*** The x-axis represents bins or evenly spaced ranges between the minimum and maximum erosion or deposition rates. We will change the figure caption to include this information in the revised manuscript.

***Comment S35:*** *Fig. 4. Omit this figure (see comment above)*

***Answer:*** Will be removed

***Comment S36:*** *Fig. 5c/d: What does it mean if CE-DYNAM has erosion rates which are up to a factor smaller than the Lugato model and C deposition rates which are more or less the same? Is it a result of different areas affected by erosion and deposition? Should be explained / discussed.*

***Answer:*** This was a mistake as we used different bins on the x-axis for the different datasets. after changing the ranges for the bins we find that the simulated rates and those of Lugato et al. are similar. Thank you for pointing this out.

[Figure]

*Figure 5c & d*

***Comment S37:*** *Fig. 7. Just a comment of an handicapped. About 4-8% of the male population are to a certain extend color blind (especially red/green is problematic), so if you do not what to*

*lose these proportion of your readers you should adapted your color in your figures. There are color blind friendly color ranges available in most software packages. If the dashed lines range between min and max the outliers cannot be above or below the lines. So, I guess the lines represent something else.*

*Answer:* We apologize for this oversight and will use different colors for the figures that are better adapted for color-blind readers. The dashed lines do not represent the min and max but the outer extremes. The outliers are defined as values that are larger than the 3th quantile by at least 1.5 times the interquartile range (IQR), or smaller than 1st quantile by at least 1.5 times the IQR.

*Changes to manuscript:* We will adapt the colorscheme of the figures in the revised manuscript

---

## Author Response (AR1)

**Response to Reviewer 1**

We would like to thank the anonymous reviewer for his or her constructive comments. In this response we provide an answer to all the comments and then indicate the changes that are applied in the revised manuscript. All line numbers we will refer to are based on the revised manuscript (not the marked-up manuscript).

**Comment 1:** *My major concern is about the validity of one assumption in the model. I am not fully convinced and expect a better justification. Because the model does not represent the river routing process, it uses floodplain connectivity to simulate the transport of sediment along hydrological pathways. However, by doing so, it implicitly assumes all sediments as sand and gravels (non-cohesive sediment) and represents the transport of cohesive and non-cohesive sediment in the same way. But the cohesive sediments (loam and silt) can be transported by rivers efficiently and most of them would not be deposited. Further, loam and silt may be the major type of sediments that are generated from hillslope erosion (especially for interrill and rill erosion considered by RUSLE). As shown in the results, the current method can cause the severe underestimation of sediment and C that are transported to oceans.*

**Answer:** We understand the reviewer's concern regarding the absence of an explicit representation of rivers and river routing in CE-DYNAM. We agree that in this way we treat the transport of all sediments types (cohesive and non-cohesive) in the same way, which can lead to uncertain sediment and POC fluxes carried away by rivers. However, our model assumption does not imply that all sediments are in the form of coarse material, instead, the main assumption is that the majority of the eroded soil and transported sediment is fine sand, silt and loam. This assumption is supported by the fact that the sediment residence time is calculated based on observed floodplain deposit ages of the Rhine (Hoffmann et al. 2007, 2008, 2013). These studies show that most of the deposits in the floodplains are overbank deposits that consist of fine sediment such as sand, loam, silt and clay and organic material. The long residence time (up to 2000 years) that they measured for the floodplains based on the C14 signature of C associated with sediment samples show that the fine sediment can stay buried for a long time in the floodplains. Although the model lacks explicit river process representations, it reproduces the spatial variability in floodplain sediment and C storage across the Rhine sub-basins as is shown by table 3 of this manuscript and by a previous study where we validated the global sediment budget model (Naipal et al., 2016, ESD). It should be noted that the model has been developed and calibrated to simulate long-term changes in sediment and carbon storage on land and not the short-term variations in sediment and POC fluxes carried by rivers.

Finally, the model produces a sediment export flux at the end of the year 2005 of $1.6 \times 10^7$ tonnes per year, which is a magnitude higher than the measured suspended sediment flux of about $3.15 \times 10^6$ tonnes per year (Asselman et al.,2003). The higher sediment flux is the result of absent riverine processes in CE-DYNAM such as sediment burial behind dams, and the fact that we assume an equilibrium state for the Rhine catchment based on the period 1850-1860 where agricultural soil erosion rates were already high. The simulated total cumulative sediment export of 2.5 Gt for the Rhine over the period 1850-2005 is about 36 % of the cumulative gross soil erosion flux of 6.8 Gt. This sediment flux leads to a cumulative POC export of about 0.14 Tg of C for the Rhine over the period 1850-2005 (based on a new simulation S2, see more details in the following paragraph below). This is 0.2 % of the cumulative C erosion flux. The yearly POC flux at the end of the year 2005 is 0.02 tC $km^2$ $year^{-1}$ (normalized over the total basin area), which is an order of magnitude lower compared to other studies who found 0.9 tC $km^2$ $year^{-1}$ (Beusen et al., 2009; Soribas et al., 2016).

This underestimation in POC in CE-DYNAM is most likely a result of the high sediment residence time of floodplains downstream of the Rhine and the absence of increased plant productivity of floodplains, leading to the decomposition of a large fraction of the deposited C. We tested the effect of the sediment residence time on the resulting lateral C fluxes of the model and find that they do not change the POC export of the Rhine significantly (see our detailed response to comment 2 of reviewer 2). Increased plant productivity of floodplains is shown to contribute significantly to the higher SOC stocks of floodplains compared to hillslopes, and to the export of DOC and POC to rivers (Van Oost et al., 2012; Hoffmann et al., 2013).

**Changes to the manuscript:** See lines 703-724 of section 4.2 of the revised manuscript.

New transient simulation S2 based on an improved equilibrium state

We redid the simulation S2 for the Rhine catchment using a different model spin-up. In the old spin-up we let the model run continuously for 2000 years, whereas in the new spin-up we ran the model for 3000 years and calculated analytically the temporary equilibrium state of the floodplain SOC pools every 10 years. This new spin-up method resulted in the floodplain SOC pools being close to equilibrium at the end of the 3000 year spin-up period, where the yearly change in the floodplain SOC stocks was less than 0.001% of the total floodplain SOC stock. Therefore, it was not needed to subtract the additional increase in the SOC stocks resulting from the disequilibrium state from those of the transient simulation (see section 2.11). The new transient simulation S2 resulted in different absolute values for the C budget of the Rhine. However, the main conclusions did not change. We also performed an uncertainty analysis with a minimum and maximum soil erosion scenario, based on the uncertainty ranges in the rainfall erosivity and land cover factors of the Adjusted RUSLE model. The revised manuscript will contain the adapted figures and tables. In addition, section 3 is modified to include the new results with uncertainty ranges.

**Changes to the manuscript:** See lines 436-439 of section 2.11 of the revised manuscript.

**Specific comments**

**Comment S1:** *L70-72: These two references are relevant to this sentence.*

*Galy, V., Peucker-Ehrenbrink, B., & Eglinton, T. (2015). Global carbon export from the terrestrial biosphere controlled by erosion. Nature, 521, 204–207.https://doi.org/10.1038/nature14400*

*Tan, Z., Leung, L. R., Li, H., Tesfa, T., Vanmaercke, M., Poesen, J., ... Hartmann, J. (2017). A Global data analysis for representing sediment and particulate organic C carbon yield in Earth*

*System Models. Water Resources Research, 53, 10,674–10,700. https://doi.org/10.1002/2017WR020806*

**Answer:** We added these references in the revised manuscript

**Changes to the manuscript:** See lines 72-73 of the introduction of the revised manuscript.

**Comment S2:** *L117: it should be noted that as discussed in Naipal et al. (2015), the formulation of R factor is related to climate type. So in the millennia time scale, one area may need different R factors due to the change of climate.*

**Answer:** This is right. In the paper of Naipal et al. (2016), where the global sediment budget model is applied for the last millennium, we take the change in climate in the calculation of the erosivity into account. For this study, we assume that the climate zones as defined by the Koeppen-Geiger climate classification have not changed drastically since 1850 AD.

**Changes to manuscript:** See lines 142-144 of section 2.2 of the revised manuscript.

**Comment S3:** *L170: Reference for Eq. 5? Also, I recommend to show the spatial variability of the f factor in the Rhine catchment.*

**Answer:** This equation has been adopted from the study of Naipal et al. (2016), that presents the global sediment budget model for the Rhine. We included the reference to this equation in the revised manuscript and added the spatial variability of the *f* factor in the supplementary document.

**Changes to manuscript:** See line 172 of section 2.3 in the revised manuscript and section S1 and S2 of the supplementary document.

**Comment S4:** *L192: This may be true for sand and gravel sediment (the majority of floodplain sediment) that Hoffmann et al. (2008) studied. But for cohesive sediment (loam and silt), they can be transported through river channels to oceans without the large fraction of deposition (at least not as large as what is set in this model). They are also the major sediments of soil erosion.*

**Answer:** See our answer to the previous comment, where we argue that most of the floodplain sediment studied by Hoffmann et al. (2008) consists mostly out of organic material (gyttja, peat) and fine sediments (fine sand, loam, silt) in overbank deposits (see table 2 in Hoffmann et al., 2008). These fine sediments are a result of long-term soil erosion on the hillslopes. Also a large part has been transported and deposited in the floodplains under major storms, such as the one in the 14th century (Bork et al., 2003). In this study (Table 6, and section 3.1 of the revised manuscript) and a previous study on the millennial sediment storage of the Rhine (Naipal et al., 2016) we show that by getting the scaling relationships as found by Hoffmann et al. (2013) right, the sediment residence time is realistic.

To show the potential effects of a different sediment residence time on the SOC storage and POC flux, we performed a sensitivity study where we changed the basin average sediment residence time to be 50% higher or 50% lower but keeping the maximum sediment residence time at 1500 years. We find that the POC flux under the low sediment residence time scenario is substantially higher than under default conditions compared to default conditions. However, the impacts of a modified sediment residence time on the total SOC storage of the Rhine are non-linear. The results of this sensitivity study is summarized in the new table 7 and in the discussion section of the revised manuscript. See changes to the manuscript and our response to reviewer 2, where we describe in the model sensitivity analysis in more detail.

**Changes to manuscript:** See lines 447-450 of section 2.11, lines 411-414 of section 2.10, and lines 790-807 of the new section 4.3 in the revised manuscript. See also the new table 7.

**Comment S5:** *L202: Similar above, this routing scheme may be fine for floodplain but whether it is appropriate for river sediment routing is questionable. And river sediment routing transports large amounts of sediment and POC from hillslopes to oceans.*

**Answer:** See our response to comments 1 and S1

**Comment S6:** *L322-326: Could you make the meanings of each term in RHS of these equations*

*more clearly? Especially, I do not very understand what the second term of RHS of Eq. 16 stand for. Also in Eq. 17, what is the difference between $1/(\tau *365)$ and kiout for SOCFLi(0,t) in the third term?*

**Answer:** The second term at the RHS of Eq. 16 stands for the C flux flowing into soil layer z from the soil layer z+1 below, and is related to the C export flux of the floodplain part of a grid cell. When the topsoil layer loses C due to sediment routing, the C from the subsoil layer 'moves' upward as is also done for C loss due to soil erosion (section 2.7). In Eq. 17 $ki_{out}$ stands for the C import rate from the neighboring grid cells. We provided a short explanation of each term in the equations 16 and 17 in the revised manuscript.

**Changes to manuscript:** See lines 366-376 of section 2.8 in the revised manuscript.

**Comment S7:** *L431-432: Or as argued by Tan et al. (2018), rainfall erosivity itself tends to be less variable if using large scale rainfall data to calculate it.*

**Answer:** We agree with this statement, however, we removed figure 4 and its explanation in the revised manuscript as we think that it does not show any new results and is thus not needed.

**Comment S8:** *L455: could the map of these 13 sub-basins be shown?*

**Answer:** We included a map of the sub-basins of the Rhine catchment in the supplementary information.

**Changes to manuscript:** See section S3 of the supplementary document

**Comment S9:** *L471-473: if much more sediment was generated but sediment deposition may still follow the long-term level, where did this additional sediment go? I suspect that it mostly was transported to oceans, a process not or poorly represented in the current model.*

**Answer:** We agree that a large part of the sediment is transported out of the catchment, more specifically 36% of the cumulative gross soil erosion rates over the entire period (see our response to the first comment).In contrast, the largest part of the eroded C in either buried in deposition areas or respired. We aim to explicitly represent riverine processes in a future study on the further development of CE-DYNAM where we also plan to include the impact of dams on the sediment export. However, the focus of this study lies on the redistribution of soil and C on land and their effect on the land-atmosphere C exchange, rather than on the riverine export fluxes of sediment and C.

**Comment S10:** *L474: that only 0.2% of sediment is exported out of the catchment is too low to believe. Are there any data to support it?*
**Answer:** See our answer to comment 1

**Comment S11:** *Section 4.2: The model also does not represent the impact of water management (such as flooding control) on floodplain connection.*
*Answer:* This is correct. We assume a 'natural' state of the catchment where the main river channel is not managed and the floodplains are more or less dynamic. We will specify this in the revised manuscript.
**Changes to manuscript:** See lines 722-723 of section 4.2 of the revised manuscript

**Comment S12:** *Figures: As discussed above, I recommend to add a few more figures (in either supplementary or appendix) to show the 13 sub-basins of the Rhine catchment and the spatial variability of the floodplain factor f and the sediment residence time $\tau$.*
**Answer:** We added these figures in the supplementary info, see sections S1,S2 and S3 in the supplementary document

**Comment S13:** *Figure 2: What does the gray level stand for? Elevation?*
**Answer:** The gray level stands for elevation, where the darker colors represent higher elevations.
**Changes to manuscript:** We added this information in the figure caption of the revised manuscript.

**Comment S14:** *Figure 3: What does the x-axis mean? Why do not you do a cell-to-cell comparison instead?*

**Answer:** The x-axis represents bins or evenly spaced ranges between the minimum and maximum total yearly soil erosion rates of the Rhine. A cell-to-cell comparison does not show a clear result due to the large variability in erosion rates. We find a quantile plot like figure 3 more useful to see for which erosion ranges the rates differ significantly between the models.

**Changes to manuscript:** We adapted the figure captions of figure 3 and 4 to include the information on the bins.

**Comment S15:** *Figure 4. Do you have another way to convey the message? It looks messy currently.*

**Answer:** We agree that this figure does not convey the message properly, after reviewer 2 had a similar opinion. We also think that the figure is not very important.

**Changes to manuscript:** We removed this figure from the manuscript

**Comment S16:** *References: Generally good. I recommend to also acknowledge the progress in other groups to represent soil erosion at large scale numerical models, such as Pelletier (2012) and Tan et al. (2018).*

**Changes to manuscript:** We acknowledged these studies in the introduction.

**Response to Reviewer 2**

We would like to thank the anonymous reviewer for his or her constructive comments. In this response we provide an answer to all the comments and then indicate the changes that are applied in the revised manuscript. All line numbers we will refer to are based on the revised manuscript (not the marked-up manuscript).

**Comment 1:** *First of all the paper lacks clear aims (or research questions). In Line 68 to 83 the authors give an overview of the contents of the paper, but I think the entire paper would improve substantially if clear aims would be given here. For example, (i) introduce a coupled soil erosion and C turnover model with an LSM model which is applicable on regional scales. (ii) Rigidly test the model for the Rhine Catchment against other modelling results and regionally available data. (iii) Analyze the sensitivity/uncertainty of the model results due to weak input data and a priori model assumptions. (regarding (iii) see comments below.*

**Answer:** We clarified the aims of our study in the Introduction section of the revised manuscript. See changes below.

**Changes to manuscript**: Lines 78-91 of the revised manuscript

**Comment 2:** *Taking the temporal and spatial scale into account which should be later on analyzed with the model I think the authors found a good balance between model complexity and simplicity. However, the model is full of a priori assumptions, which will fundamentally affect the modelling results, so I personally do not think any model results can be interpreted without some estimates of at least the sensitivity of the model against these assumptions. The most important assumptions which could be tested easily are: C input via plants especially crops depending on erosion status, C enrichment during erosion and depletion during deposition, reduced C turnover in alluvial soils due to wetter conditions, etc. Overall, it is one of the major shortcomings of the paper that the modeling results in section 3.2 are presented single values (e.g. for 159 Tg C for C removal by erosion) and also conclusions based on this single model*

*results are presented. I strongly suggest performing a a sensitivity analyses (including as far as possible effects of a priory assumptions) and giving results with a reasonable range. I am fully aware that it would be hardly possible to do a full uncertainty analysis and even an sensitivity analysis might be quite ambitious given the catchment size and the complexity of the involved models. However, it is not enough just stating in the discussion some important processes are not taken into account.*

**Answer:** We agree that an uncertainty analysis is important for a regional modelling study such as ours. Therefore, we performed additional simulations with a minimum and maximum soil erosion scenario, based on the uncertainty ranges in the rainfall erosivity and land cover factors of the Adjusted RUSLE model. Chapter 3 of the revised manuscript is modified to include the new uncertainty results. We also modified figure 9, which is figure 8 in the revised manuscript, to include the uncertainty ranges in the C budget components.

Regarding the sensitivity analysis of the model we tested the assumption of C enrichment during erosion as suggested by the reviewer. Here, we performed two additional simulations with an enrichment factor of two adapted from the study of Lugato et al. (2018): S1_EF (erosion only) and S2_EF (erosion with deposition and transport). We also tested the rate of C transport between floodplains by letting the basin average sediment residence time to vary between a 50% lower and 50% higher value compared to the default. For this purpose we did another two additional simulations (S2_Tmin and S2_Tmax). We also tested the model sensitivity to the crop residue management (S0_RM, S1_RM, S2_RM) as suggested by reviewer 1. Here we assumed an extreme scenario where all above-ground crop litter is harvested.

However, we abstained from testing the model performance to a changed C turnover in alluvial soils as a result of wetter conditions. Previous studies show that there are still large uncertainties related to the turnover of C in depositional environments, and more specifically of alluvial soils, as they represent complex soil profiles with a wide range in physical, chemical and biological parameters that affect the C turnover in interaction with climatic variables such as soil moisture. For example, the studies of Doetterl et al. (2018) and Rasmussen et al. (2018) show that the C turnover of alluvial soils is determined by C stabilization affected by the availability of minerals (such as Iron, Aluminium) and nutrients, mediated by soil microbes and by the formation of peat deposits on river banks. Yet, old alluvial soils can be far from water-saturated, in which case the C turnover would not be substantially decreased as a result of additional oxygen limitation. In our study we also include floodplains that do not get flooded regularly. Therefore, it is not clear if these alluvial soils are in general 'wetter' than the colluvial soils and would therefore have a significantly different C turnover. Also, our model does not include a good representation of groundwater dynamics and a soil moisture function for alluvial soils. After performing an extensive literature study on C turnover in alluvial soils we could not find a way to easily but realistically modify the C turnover of alluvial soils, for example by using a simple turnover reduction factor derived from observations. We introduced a new section 4.3 in chapter 4, where we discuss the results from the sensitivity simulations. See changes to manuscript below.

**Changes to manuscript**: Section 2.11 lines 445-457, equation 15, and section 4.3 of the revised manuscript. See also the new tables 2 and 7, and adjusted figure 8 of the revised manuscript.

**Comment 3:** *From my understanding of the paper and accounting for the scope of the journal testing such new model against data is essential. The authors did try doing so but here a lot of improvement is easily possible: (i) Include a section under methods explain which data are used to test the model and also explain in some detail how this is done. For example, the comparison with other models as given in Fig. 3 and 5 is not clear, as the following information is missing: (a) Were the data from the more high resolution models aggregated to the raster cell size of CE-DYNAM to do a raster-by-raster comparison? (b) If a CE-DYNAM raster cell consist of erosional and depositional sites, which are not resolved in the raster cell, how to compare with gross erosion of a high resolution model (e.g. Panagos et al. 2015) which might have different proportions of erosional and depositional raster cells in this large 8 x 8 km 2 raster cell. (c) It is not clear at all what is compared as all model results from literature do not focus on the time span from 1850 to 2005. These details are essential for the reader to understand your model validation. (ii) From the figures I have some doubts that the different models fit very well (why not giving statistical goodness-of-fit-parameters?). So the question is how good the other models*

*are (please see e.g. the scientific debate regarding the Panagos et al. (2015) map. So, at least in the discussion this model to model comparison needs to be stressed. (iii) Generally, the erosion (partly deposition) validation of CE-DYNAM is mostly done against other models also using USLE technology (USLE factors might be even derived from same data sources), so an extended discussion if this is meaning full is needed.*

*Answer:* To better clarify the model validation and comparison against data and other models, included an additional section 2.12 in the revised manuscript where we discuss the validation data used, and how the validation is done in more detail. In this new section we also mention the reasons why we do this model to model comparison, where we provide more background information on the various models. Finally, we provide a statistical goodness-of-fit summary by comparing the total soil erosion and carbon erosion rates at sub-basin level to those of the other studies (see tables 3 and 4 of the revised manuscript).

**Changes to manuscript**: New section 2.12 on validation data and methods added to the revised manuscript. See also the new tables 3 and 4 of the revised manuscript.

**Specific comments:**

**Comment S1:** *Line 68-83: see general comment.*

*Answer:* See our response to general comment 1

**Comment S2:** *Line 89: be more explicit regarding 'low number of parameters'*

***Changes to manuscript:*** See line 100-101 of the revised manuscript.

**Comment S3:** *Line 118 ff: I do not agree that not taking the L factor into account is a reasonable decision. I agree that it is somewhat difficult to estimate (for the German part of the Rhine catchment there are some estimates) but if you are interested in land use change it is an essential factor if you kick it out the entire basis of the USLE is set into question. (The P factor is simpler as it is set to 1 in most studies).*

*Answer:* We agree that leaving both the L and P factors out of the equation will induce some bias in the results, especially for agricultural land. In our next study we aim to make CE-DYNAM better applicable for agricultural land, where these factors play an important role. For this purpose we will focus on the development of new methods that can quantify the L and P factors reliably at the global scale, and will need to re-calibrate the erosion module of CE-DYNAM, the Adj.RUSLE. Our decision of leaving out the L and P factors from the erosion equation in our study is based on the global study of Doetterl et al. (2012), which showed that the S, R, C and K factors explain approximately 78% of the total erosion rates on cropland in the USA. This indicates that on cropland the L and P factors, which are related to agriculture and land management, contribute only for 22 % to the overall erosion rates. This percentage is comparable to the uncertainty range in the estimation of the S, R, C and K factors at the regional scale from coarse resolution data. Renard and Ferreira (1993) also mention that the soil loss estimates are less sensitive to slope length than to most other factors.

Furthermore, various studies argue that the estimation of the L factor for large areas is complicated and thus can induce significant uncertainty in soil erosion rates calculated based on coarse resolution data  (Foster et al., RUSLE2 user guide; Kinnell, 2007). Especially, for natural landscapes, such as forest, the estimation of the L factor is not straightforward as these natural landscapes usually include steep slopes (Elliot, 2004). In order to stay consistent with the estimation of potential soil erosion for all land cover types, we remove the L factor from the equation. The Adj.RUSLE has been already successfully validated at the regional scale, without the L and P factors where the spatial variability of soil erosion rates compares well to other high resolution modeling studies and observational data and the absolute values fall within the uncertainty ranges of those validation data (Naipal et al., 2015; Naipal et al., 2016; Naipal et al., 2018; and this study). Finally, the aim of this study was to develop and validate a carbon erosion module for applications at the global scale, where the estimations of the L and P factors is even more limited. By showing that the erosion rates from the Adj.RUSLE and CE-DYNAM are within the uncertainty of other data and modelling studies, we can assume that it will be applicable for other large catchments in the temperate region.

**Changes to manuscript:** In section 4.2 'Model limitations' of the revised manuscript, we addressed the lack of the L and P factors in more detail in the same way as described above (see lines 750-771).

**Comment S4:** *Line 130: The statement ". . .has been calibrated and validated for the Rhine catchment. . ." is confusing here? If calibration and validation was already done why doing it again? If the model has changed you need a new validation (but what about calibration? Are you using parameters in CE-DYNAM which were calibrated before it is necessary to indicate this in detail).*

*Answer:* In our study we use the model parameters of the global sediment budget model as defined and calibrated by Naipal et al. (2016), such as the sediment residence time or the floodplain deposition factor. We did not perform an additional calibration of the sediment dynamics part of CE-DYNAM, only a validation, because of the use of different input datasets.

**Changes to manuscript:** We added a sentence specifying why we redid the validation, see line 467-468 of section 2.12 in the revised manuscript.

**Comment S5:** *Line 113: Alluvial soils are indicated in German soil maps, so the statement is not correct for the largest part of the Rhine catchment.*

*Changes to manuscript:* We will modify the sentence to: 'It should be noted that global soil databases do not identify floodplain soil as a separate soil class, although national soil databases might. However, the aim of this study is to present a carbon erosion model that should be also applicable for other catchments and eventually, globally. Therefore, we followed a 2-step methodology to derive floodplains in the Rhine catchment using hydrological parameters and existing data on hillslopes and valleys.' See lines 151-155 of section 2.3 of the revised manuscript.

**Comment S6:** *Line 141 / Eq. 2: Generally I think it would be good being more precise with the equation. For example in case of Eq. 2, I would expect a reference to the different raster cells*

*(Aft(i) = Lstream(i) x Wstream(i); whereas i is the raster cell.) as for other equations e.g. Eq. 4a it was not clear if this refers to the entire catchment is calculated for each raster cell.*

**Changes to manuscript:** We modified the equations accordingly in the revised manuscript.

**Comment S7:** *Line 148: If alpha and b are constants it means that the upstream area necessary to result in a stream is always the same. I understand that in case of a large scale model simplifications are necessary but this assumptions is for sure not true for the Rhine catchment (see papers from hydrology of maps of the stream system (which by the way would be available for the entire Rhine catchment).*

*Answer:* We agree, that they might not be the same for the entire catchment, and variations exist. But these constants have been derived from 467 cross-sections of the Rhine catchment combining 1:25 000 geological maps and catchment area extracted from the SRTM 3 arcsec digital elevation model (Hoffmann et al., 2007).

**Comment S8:** *Line 159: '... at 8 km resolution... " I guess this means 8 km x 8 km raster cells. Should be changed throughout the text (also with other resolution given).*

*Answer:* Yes, this means indeed a 8 km x 8 km raster. We added a sentence in section 2.3 explaining this.

**Changes to manuscript:** See line 138-140 of section 2.2

**Comment S9:** *Line 163: I do not think the the assumptions of reduced hydrological and geomorphological connectivity in arable landscapes (compared to forest) is correct. From the recent studies dealing with flash floods it is obvious that it is a main problem that this landscapes have a very high connectivity as so many ditches, drainages etc. were built over the last century to get rid of any surplus of water on arable land. So, your assumptions for the range of the parameter f in different landscapes must be underlined by reasonable data.*

*Answer:* This assumption is underlied by several studies (Hoffmann et al., 2013; de Moor and Verstraeten, 2008; Gumiere et al., 2011;Wang et al., 2015) on the effect of erosion on sediment yield, where is shown that man-made activities on agricultural landscapes result in a trapping of eroded soil in colluvial deposition sites, reducing the sediment transport from hillslopes to the floodplains. The model parameter *f* has been calibrated for the Rhine catchment before in Naipal et al. (2016), where this range is found to produce a ratio between hillslope and floodplain sediment storage that was comparable to observations. The studies of Wang et al. (2010; 2015) identify a range for the hillslope sediment delivery to be between 50 and 80 %, which is similar to the range in the (1-f) factor of our model.

**Changes to manuscript:** We included these arguments on the choice for the *f* parameter in section 2.3 lines 188-196 of the revised manuscript.

*Comment S10:* *Line 187: Does a multiple flow algorithm makes sense in case of a resolution of 8 km*

*Answer:* The multiple flow algorithm is especially effective in hilly regions, and we expect it to work better than the single flow algorithm for these regions. In such steep landscapes the river courses are meandering a lot. So the downstream part of a river can easily cut through the boundary of two downstream lying adjacent 8 km x 8 km cells. With the coarse resolution of 8 km, a single direction algorithm would lead to an extreme straightening of the river network, and it would underestimate the number of cells which have a proportion of floodplain area. However, we agree that the coarse resolution grid size will affect the results of both algorithms.

*Comment S11:* *Line 230 ff: In general this is a reasonable assumption for the crop residues. However, studies in small catchment clearly indicate that residue management is a key factor of SOC, so this a priori assumption has potentially a huge effect on the produced results. So, its importance must be analyzed with the model!*

*Answer:* We agree that the harvest and crop residues left on the field would have a large effect on the SOC dynamics of agricultural landscapes. It should be noted that we implemented an increase in the harvest index during the period 1850-present-day based on the study of Hay (1995), which already partly accounts for crop residue management. To explicitly quantify the potential impacts of crop residue management we performed additional sensitivity simulations where we assumed that all above-ground crop residues are harvested. After running CE-DYNAM with crop residue management we find that total litter C stock is about 15% smaller compared to the default case by the end of the year 2005. This leads to a total change in the transient SOC stocks that is 20% smaller under no erosion (S0), and 26% smaller under erosion (S2). Our findings confirm that soil management practices such as residue management have a substantial effect on the SOC dynamics.

**Changes to manuscript:** Section 2.11 and table 2 of the revised manuscript describes the setup of the sensitivity simulations with respect to crop residue management. The new section 4.3 and table 7 present the results of crop residue management.

*Comment S12:* *Line 238-240: The given equation are a fundamental problem with modelling the effect of soil erosion on SOC turnover. For example, using standard SOC pool residence times for all landscape positions is of tremendous importance for the entire C balance effect of erosion. So, again it would be very important to know how sensitive the results are against this assumptions. At least give some estimates / measurements at different landscape positions in the discussion and comment of the potential effect in modelling results.*

*Answer:* The SOC pool residence times at different landscape positions (hillslope, depositional sites etc), is interrelated with weathering, soil erosion and sediment transport processes (Berhe et al., 2008). To be able to have different residence times for each SOC pool as a function of the landscape position, soil erosion effects on for example the aggregation of soil particles and transport of minerals and nutrients has to be included. This can currently not be done in global land surface models. However, there are efforts to change the current SOC dynamics scheme of LSMs by introducing measurable SOC pools (Abramoff et al., 2018). In this case it might be possible to calibrate the residence time of each pool based on the landscape position.

**Changes to manuscript:** See lines 739-743 in section 4.2 of the revised manuscript.

**Comment S13:** *Line 265: ". . . The next soil layer contains less C and therefore at the following time-step les C will be eroded under the same erosion rate. . ." If this would be always true one would expect a continuous decline in SOC in soils. However, assuming a long- term forest use on a slope you will found the soil in an equilibrium between new C input via plants and small amount of erosion. So, in this case the eroded material will have a more or less constant C content.*

**Answer:** We understand that this sentence might be unclear. What we mean is that the reduction of C erosion at each timestep due to less C being available for erosion, and the existence of a compensatory C sink due to the erosional removal of C, will ultimately lead to an equilibrium state. Which is also reached in our model.

**Changes to manuscript:** We add the following sentences to the revised manuscript (lines 308-310 of section 2.7): " The removal of C by erosion also triggers a compensatory C sink due to the reduction in SOC respiration on eroding land. This compensatory C sink and reduced C erosion over time will ultimately lead to an equilibrium state."

**Comment S14:** *Line 277: Calculating a daily erosion fraction is a reasonable approach. However, if taking the episodic nature of erosion and deposition into account the C balance will be different compared to a small continuous process (see literature). Might be also discussed.*

**Answer:** We will mention this aspect in the discussion chapter of the revised manuscript.

**Changes to manuscript:** See lines 773-776 of section 4.3 of the revised manuscript.

**Comment S15:** *Line 291 ff: The assumption that there is no C selectivity (enrichment in eroded material and depletion at erosional sites) is taken in many modelling approaches. However, if there would be no enrichment of fines in the sediments transported in river systems, one would find e.g. sand in suspended sediments of larger rivers. Which is e.g. in case of the lowland Rhine not the case. Discuss this in the context of the scale of your paper. Also important regarding the loss of C to the ocean.*

*Answer:* We agree and performed a sensitivity analysis of the model where we changed the C enrichment factor to 2 , hereby, partly accounting for the selectivity of erosion. We find that although the POC export to the ocean is not significantly affected, the SOC storage and resulting C sink is increased. See our detailed response to general comment 2.

*Comment S16:* *Line 341: Where do the data regarding afforestation during the last two decades come from. To my knowledge this is a process already started in the late 1959[th] (please give reference)*

*Answer:* We use the land use change data from the study of Peng et al. (2017), who bases their estimates of forest cover on Houghton (2003,2008) for the period 1850-1990 and satellite data for the recent decades. The historical national forest area used by Houghton et al. since 1850 are from national surveys and they are arguably the best data available, although uncertainty arises when downscaling historical forest area change on a grid, using (uncertain) gridded reconstruction of agricultural land (HYDE) and land use transitions rules (see details for the method used by Peng et al. 2017).

*Comment S17:* *Line 379: (see also general comments). I wonder why you did not use other more specific and potentially profound national data. E.g. for Germany there are several maps for potential erosion which are much more elaborated than the map of Panagos et al. (2015). Moreover, I wonder why you did not use the sediment delivery data of the Rhine which are freely available - I guess since the 1950th - which would be a good and reliable additional data set for validation.*

*Answer:* We did not use the sediment delivery data of the Rhine, because the comparison to our simulated coarse resolution model results will most likely be not entirely justified. In our model we do not take into account daily changes in precipitation and runoff and how that affects the erosion rates and sediment transport. Instead we use yearly totals. We also do not take into account dams and other man-made structures that would affect the river transport of sediment. Further, we focus only on the rill and interrill erosion and do not account for other soil erosion processes and flash floods that might have a larger effect on the sediment delivery. Finally, our model has not been developed to simulate the river transport of sediment and C, but instead is focused on the redistribution of soil on land and the resulting sediment and SOC storage. See also our response to the general comment of reviewer 1. For the future development of CE-DYNAM we aim to better represent the river transport processes of sediment and C.

Regarding the validation of soil erosion with observations, we used the database of Cerdan et al. (2010), which already includes German national estimates on soil erosion. We also performed a comparison of our agricultural soil erosion model estimates to the agricultural soil erosion potential map of the Federal Institute for Geosciences and Natural Resources of Germany, available at 250 m resolution, in the revised manuscript.

**Changes to manuscript:** Figure 3C is added and discussed in section 3.1 of the revised manuscript. Section 2.12 describes the German soil erosion map used for validation.

***Comment S18:*** *Line 397-401: I suggest omitting these sentences and Fig. 4, because I do not see any additional value of this here. It is obvious from the model structure of all USLE based models (and all other erosion models) that an increase of erosivity and slope directly leads to an increase in erosion. Moreover, there is a coincidence in the catchment that highest erosivity and highest slopes occur at the same alpine area, but this is not any proof for the model. Hence, hence I think this is weakening your validation more that it would strengthen it. By the way: Erosivity and slope might explain 70% erosion if very different rainfall regimes and slopes (mountain areas and lowlands) are compared, but with a catchment like the Rhine (where except for the alpine part) the differences in slope and erosivity are relatively small soil cover (C factor) is getting much more important (erosion rates between grassland and arable land vary by a factor of 10-20).*

***Answer:*** We agree and omitted figure 4 in the revised manuscript.

**Comment S19:** *Line 402: As I modeler I expect a goodness-of-fit parameter with this statement. (See also general comments regarding model to model comparison of different USLE implementations).*

*Answer:* We included a goodness-of-fit summary related to the soil erosion result comparison at sub-basin level, see our response to comment 3.

**Changes to manuscript:** See new table 3 and 4 of the revised manuscript and their description in section 3.1

**Comment S20:** *Line 424 ff: The comparison with the data from Hoffman et al. (2013) underlines a deficit in all your comparisons. It is at no time clear what is compared exactly. Mean of 7500 years against 1850-2005?*

*Answer:* See the last paragraph in our response to general comment 3 and modifications to the manuscript.

**Comment S21:** *Line 438: Does the outflux fit to measured data? Would be easy to test even if this is not essential as only a very small amount will be delivered into the sea. (could be tested at several subcatchment, as data are available).*

**Answer:** See our response to general comment 1 of reviewer 1

**Changes to manuscript:** See lines 703-724 of section 4.2 of the revised manuscript.

**Comment S22:** *Line 451-452: This is a clear contradiction to your statement that differences in erosivity are very important for spatial differences in erosion.*

*Answer:* We agree and removed this sentence in the revised manuscript.

**Comment S23:** *Line 453: The close link between C erosion and soil erosion is obvious from your modelling structure but not necessarily correct (C enrichment depending on event size?)*

**Changes to manuscript:** See lines 592-594 of section 3.2 of revised manuscript

***Comment S24:*** *Line 434 ff: See also general comments*

***Answer:*** We address this in our response to the general comments

***Comment S25:*** *Line 446: I suggest not to over interpret the modeling results from the alpine area of the catchment as the modelling and the data are weakest there. (i) Increase in measured precipitation most uncertain; (ii) calculated R factor very uncertain in all USLE approaches; (iii) alpine USLE factors not very well underlined by data (compared to arable and grassland),*

***Answer:*** We agree and are aware of this bias and therefore, present all model and validation results for the non-alpine region only.

**Changes to manuscript:** See line 534-535 at the beginning of chapter 3 of the revised manuscript.

***Comment S26:*** *Line 497ff. See comments regarding connectivity above. Moreover, even if the connectivity is high under forest (which I doubt), forest will produce not a lot of sediment and hence are not so important for building up alluvial soils at all.*

***Answer:*** We agree that soil erosion in forests is minimal but forests also appear often in hilly landscapes that will contribute to the sediment production. When analyzing soil erosion rates over timescales longer than a few decades, extensive forest areas will contribute significantly to the overall removal of soil. Forests contain also a lot of SOC, and so minimal rates of soil erosion might be still significant for the SOC dynamics, also in depositional areas, as is demonstrated in the recent study of Billings et al. (2019).

***Comment S27:*** *Line 493ff. I do not see from the results the CO2 fertilization plays an important role for an increase in dynamic replacement. I guess that the increase in yields due to changes in management are much more important (as reduced yields are not taken into account at erosional sites) as they boost dynamic replacement of eroded soils.*

***Answer:*** We agree that increased yields due to management boost the dynamic replacement of eroded soil but the largest effect comes from the $CO_2$ fertilization due to increased atm. $CO_2$

concentrations. See below figure 1 representing the actual C replacement under land use change and climate change, and figure 2 representing the potential C replacement under a fixed climate (no change in temperature, precipitation, CO2 atm. concentrations, but with a changing land use and management).

[Figure]

*Fig 1 & 2: C replacement on eroding soils*

**Changes to manuscript:** We include these figures in the supplementary material section S4 A&B

***Comment S28:*** *Line 501ff: It is obvious from first order kinetics that colluvial soils must have higher CO2 effluxes as they contain more C. So, this is not a very new finding.*

***Answer:*** We agree that this might be an obvious finding, but this has not been quantified for such a large catchment before. With this finding we also indicate that the model reproduces process knowledge from field work.

***Comment S29:*** *Line 506-507: Question: Is the modelled increase in respiration from floodplains resulting from an temperature increase or from an increase in depositional material, which would also result in an increase of respiration? Comment: Under real conditions the increase in respiration from floodplains is also a result of decreasing groundwater levels,*

***Answer:*** After redoing the simulations using a better spinup method (see section 2.11, lines 436-439) we find that the respiration from floodplains is rather variable and shows a decreasing trend. This is mainly a result of a decreased C deposition flux. In our model we do not have a specific representation of ground water for floodplains.

**Changes to manuscript:** See lines 644-653 of section 3.2 of the revised manuscript

*Comment S30:* *Line 542-561: This is a nice collection of model deficits. However, for a modelling paper I would expect a bit more (see general comments).*

*Answer:* See our response to the general comments and resulting improvements made to the manuscript.

*Comment S31:* *Line 576-588: I think this conclusions are not fully supported by the results as the modelled C fluxes might be affected by a priori assumptions and model parameters which are not tested enough (see general comment regarding sensitivity analysis).*

*Answer:* We adjusted the findings in the conclusions

*Changes to manuscript:* See lines 836-839 of chapter 5 of the revised manuscript and lines 845-846.

*Comment S32:* *Table 1: I guess the spatial resolution is always given in raster cells, e.g. 0.25 ∘ x 0.25 ∘.*

*Answer:* See our response to comment S10 and respective changes in the revised manuscript.

*Comment S33:* *Table 2: As the resolution of the data sets are different how to make sure that the comparison fits, e.g. the higher resolution data set might exclude the river network from the SOC calculation while the lower resolution data set might include this areas into the SOC stock calculation. Give somewhat more details.*

*Answer:* We give more details on the resolution and comparison in the new section 2.12 of the revised manuscript. See our response to comment 3.

*Comment S34:* *Fig. 3 and 5: What are the 10 classes given on the X-Axis?*

*Answer:* The x-axis represents bins or evenly spaced ranges between the minimum and maximum erosion or deposition rates. We adjusted the figure caption to include this information in the revised manuscript.

*Comment S35:* *Fig. 4. Omit this figure (see comment above)*
*Answer:* Is removed

*Comment S36:* *Fig. 5c/d: What does it mean if CE-DYNAM has erosion rates which are up to a factor smaller than the Lugato model and C deposition rates which are more or less the same? Is it a result of different areas affected by erosion and deposition? Should be explained / discussed.*
*Answer:* This was a mistake as we used different bins on the x-axis for the different datasets. after changing the ranges for the bins we find that the simulated rates and those of Lugato et al. are similar. Thank you for pointing this out.
**Changes to manuscript:** See the new figure 4c/d of the revised manuscript

*Comment S37:* *Fig. 7. Just a comment of an handicapped. About 4-8% of the male population are to a certain extend color blind (especially red/green is problematic), so if you do not what to lose these proportion of your readers you should adapted your color in your figures. There are color blind friendly color ranges available in most software packages. If the dashed lines range between min and max the outliers cannot be above or below the lines. So, I guess the lines represent something else.*
*Answer:* We apologize for this oversight and used patterns and different line styles to adapt figures 5, 6 and 7 of the revised manuscript, where we expect difficulties for color-blind readers. The dashed lines in figure 7 do not represent the min and max but the outer extremes. The outliers are defined as values that are larger than the 3th quantile by at least 1.5 times the interquartile range (IQR), or smaller than 1st quantile by at least 1.5 times the IQR.
**Changes to manuscript:** See the adapted new figure 5,6 and 7

[revised manuscript text omitted]

---

## Author Response (AR2)

**Report reviewer1**

Thanks for the detailed response. It indeed clarifies many things.

If I understand correctly, because the long sediment residence time in downstream grid cells near to the river mouth (Fig. S2), the sediment and C export from the river to the sea should be mainly controlled by the sediment and C budget in those downstream grid cells. Using the value of sediment export ($1.6\times10^7$ tons $yr^{-1}$) and the value of C export ($0.0095\times10^{12}$ g C $yr^{-1}$), I calculated that the C density in those grid cells is only $6\times10^{-5}$ g C $g^{-1}$. This is extremely low. I suspect that the simulated C density in other grid cells of large sediment residence times is also low.

Because the authors have explained that their focus is on the impact of soil erosion and deposition on land C dynamics in the revision, I actually agree that it is not important to make the simulated export of sediment and C to the sea match the observations. However, as said above, the problem of simulating C dynamics over floodplain cannot be ignored. I am not sure whether the problem is on the CE-DYNAM side or the ORCHIDEE side. Could you compare the surface layer C density in grid cells of long residence time between S0 and S2 to identify the cause? If it is on the CE-DYNAM side, further model improvement is needed.

**Response:** We thank the reviewer for his or her constructive comments. First, we would like to correct the POC and sediment export values of the Rhine. We mistakenly calculated the sediment bound POC export using the SOC concentration averaged over the whole soil profile, while we should have used the SOC concentration averaged over the topsoil layer (top 10cm) only as it is done in the in the routing of C in CE-DYNAM. After accounting for this oversight, we find that the new POC export at the Rhine outlet is around $2\times10^8$ g C $yr^{-1}$ instead of the $9.5\times10^8$ g C $yr^{-1}$ as written before. This new value is two orders of magnitude lower than the $2.6\times10^{10}$ g C $yr^{-1}$ given by the GlobalNEWS2 model (Mayorga et al., 2010), and is a result of the low simulated sediment export rate. The simulated sediment export value of the Rhine is 6472 tons $yr^{-1}$ instead of the $1.6\times10^7$ tons $yr^{-1}$ as written before. This value is also two orders of magnitude smaller than the $3.15\times10^6$ tons $yr^{-1}$ that is found by Asselmann et al. (2003) or the $0.75\times10^6$ tons $yr^{-1}$ simulated by Li et al. (2020). This is due to the fact that CE-DYNAM does not explicitly simulate rivers and streams. As a result, CE-DYNAM does not differentiate between eroded hillslope soil that reaches the water network in a relative short time, and the sediment that is first retained in the floodplains before it reaches the water network due to fluvial erosion. Instead the sediment and POC export rates in the model are only controlled by the sediment residence time of floodplain (alluvial) soil, which ranges between 0 and 1500 years, while the residence time of fine suspended sediment in rivers is usually much shorter (Wallbrink et al., 2002).

To account for the direct sediment flux from hillslopes to the water network, we would need to modify the floodplain sediment deposition (f*E, see equation 4a), and differentiate between the sediment residence time of suspended sediment in rivers, and the residence time of alluvial deposits. However, the focus of this study was to model the full erosion-C loop, and not the sediment delivery to the water network. In a future study we aim to improve the sediment and POC export of the model by focussing on the suspended sediment fraction in the water network.

Furthermore, CE-DYNAM does not simulate fluvial erosion as a complex function of the channel geometry, riverbank erodibility and sheer stress (Droege et al., 1992), due to the lack of data on these parameters at the regional scale, and to keep a balance between model complexity and its computational ability. Also, our model does not resolve erosion of the deposited river sediment by flooding events. This simplified model concept for fluvial erosion contributes to the underestimation of sediment export in floodplains. Finally, with the current model setup we do not account for large soil erosion events before 1850 or extreme precipitation events that may have a long-term effect on the sediment export rate of the Rhine. For the above-mentioned reasons it is hard to compare the sediment and POC export rates of our model with observations.

However, we find that the C density in the topsoil layers of floodplain soils located downstream of the Rhine and the C concentration of the POC flux are realistic. We find a C concentration of ~3.3% in the exported fine sediments. Abril et al. (2005) found a 5.5% POC mass fraction in suspended sediments for the Rhine. The C density of the topsoil layer of the floodplains in the downstream grid cells in the S2 simulations (S2, S2_min, S2_max) is on average 4.47 kg C m$^{-2}$, which falls within the range of the average C density of 5.13±1.3 kg C m$^{-2}$ measured by Hoffmann et al. (2013a) for floodplain overbank deposits. By comparison, the average C density of the topsoil layers of downstream grid cells in the S0 simulation is 12.78 kg C m$^{-2}$, which is an overestimation.

**Changes to the manuscript:** We updated the discussion about the sediment and POC export rates in section 4.2 (Model advantages and limitations) between lines 703 and 724 in the revised manuscript. We also updated the POC export rates in figure 8 and table 7.

Other comments
1) There are Hoffmann et al. (2013a & 2013b) in the references but only Hoffmann et al. (2013) is cited. Please clarify.

**Response:** Thank you for pointing this out. We cite both papers of Hoffmann as indicated in the references of the revised manuscript.

2) When checked Naipal et al. (2016), I noticed that simulated sediment storage can be an order of magnitude lower than the observations, even they show the similar scaling factors. How is the absolute value of simulated C storage compared with Hoffmann et al. (2013a)?

**Response:** The absolute value of C storage from the S2 simulations of the non-Alpine region of the Rhine for the year 2005 is in the range of 2.74-2.99 Pg of C, which is larger than the 1.7±0.6 Pg of C that Hoffmann et al. (2013a) measured. It should be noted that the ORCHIDEE model (S0 simulation) already overestimates the total SOC stock of the Rhine (2.43 Pg of C), when the initial conditions of the period 1850-1860 are used. Although this is in contrast to the findings of Hoffmann et al. (2013a), the difference in SOC stocks between floodplains and hillslopes from the S2 simulations is significantly better than the difference derived from the S0 simulation. We find that floodplains store 1.28-1.72 and hillslopes 1.7-2 Pg of C when erosion and deposition processes are taken into account, compared to 0.69 Pg of C for floodplains and 2.29 Pg of C for hillslopes when these processes are lacking.To be closer to the observational difference between floodplains and hillslopes we would need to consider the period before 1850, consider extreme climate events, and a higher plant productivity in floodplains. Furthermore, it should be noted that the sediment dynamics play a less important role in the total C storage compared to the NPP and soil respiration. Especially, if our study only considers the period 1850-2005 and not the entire Holocene like the study of Hoffmann et al. (2013a).

**Changes to the manuscript:** We include the comparison of the total SOC stocks in the discussion section 4.1 of the revised manuscript.

3) Eq. 6: Is this residence time equation universal in the world since the authors emphasized that the model would be applied globally?

**Response:** Yes, the equation should be applicable for large river basins worldwide such as the Rhine. However, the parameters of this equation are calibrated based on data from the Rhine. To be able to apply it on other river basins, it would be necessary to re-calibrate the parameters based on the specific sediment storage conditions of those basins.

**Changes to manuscript:** We include the following sentence in the revised manuscript after line 213 (section 2.3): 'These constants will need to be calibrated based on local data of sediment ages before CE-DYNAM can be applied to other catchments globally. '

4) L436: Should be "lose" not "loose".

**Response:** Thank you for pointing this out. We will correct this in the revised manuscript.

5) Could you explain what is net soil erosion? Is this the eroded sediment deposited to the floodplain fraction?

**Response:** The net soil erosion is the eroded sediment that leaves the hillslopes and is deposited to the floodplains.

**Changes to manuscript:** We will adjust line 310 in the revised manuscript (section 2.7) as following: 'The change in C content due to net erosion (the eroded sediment/C that leaves the hillslopes after deposition) of the PFT-specific pools….'

6) Please define NPP and NEP before using abbreviations.

**Response:** NPP is the Net Primary Productivity and NEP is the Net Ecosystem Productivity. We will include these descriptions in the revised manuscript.

7) L548–449: Table 3 shows that the result of simulated net soil erosion is not good as described. Can be possibly related to my first comment?

**Response:** The estimation of the net soil erosion between our study and the study of Borrelli et al. (2018) is done in different ways, which may explain the difference in the results. In our study the deposition of sediment in hillslopes is explicitly calculated as a function of the slope, and vegetation type/cover. Borrelli et al. (2018) used the transport capacity concept (Van Rompaey et al., 2001). Both methods have their uncertainties when applied at large spatial scales. The method in our study has been designed and calibrated to be used at a large spatial scale, and at coarse resolution, while the method of Borrelli et al. (2018) was originally designed to be applied at spatial scales <100m, where the transport capacity coefficient needs to be calibrated with local data.

**Changes to manuscript:** We will include this explanation in the revised manuscript in section 3.1, after line 549.

8) L555-557: Sorry, I do not understand the mechanism. How can erosion-control and management practice affect this relationship?

**Response:** Indeed, erosion-control (EC) practices do not fully explain the difference in the relationships between soil and C erosion rates of the different studies. Thank you for pointing this out. EC practices reduce the overall erosion rates. Our study does not include them, and so our simulated erosion rates may be substantially larger in regions with EC. Figure 5 shows that our simulated erosion rates are in general larger than the erosion rates from Lugato et al. (2018), which may be explained by this mechanism. However, at the same time the C erosion rates of our study are lower than those of Lugato et al. (2018). This is likely due to the coarse spatial resolution of our underlying C-cycle -scheme, which is derived from the ORCHIDEE LSM.

**Changes to manuscript:** We will change lines 555-557 as following: 'On the one hand, our study does not include erosion-control (EC) practices, leading to substantially larger simulated soil erosion rates in regions with EC. Figure 5 shows that our simulated erosion rates are in general larger than the erosion rates from Lugato et al. (2018), which may be explained by this mechanism. On the other hand, the C erosion rates of our study are lower than those of Lugato et al. (2018), due to the coarse spatial resolution of our underlying C-cycle -scheme derived from the ORCHIDEE LSM.'

9) Figure 3A: Why is the simulation missed in some bins?

**Response:** This is because the bins in figure 3a are based on the min-max erosion rates of Cerdan et al. (2010). For bin 8 where data is missing, there were no erosion rates from Cerdan et al. recorded. In our study the simulated maximum soil erosion rates were lower compared to Cerdan et al., likely due to the coarse spatial resolution of our model, therefore there is no data for bins with a high number.

**Changes to manuscript:** We will change line 1166 of revised manuscript as following: ' The x-axis represents bins or evenly spaced ranges between the minimum and maximum total yearly soil erosion rates of the Rhine derived from the data of (a) Cerdan et al. (2010), (b) Panagos et al. (2015), (c) Bug et al. (2014), and (d) Borrelli et al. (2018).'
We will also modify line 542 as following: 'We find that the quantile distribution of the simulated gross soil erosion rates compares well to the distributions of other observational and high-resolution modelling studies (Cerdan et al., 2010, Panagos et al., 2015, Bug et al., 2014), although CE-DYNAM usually underestimates the very large soil erosion rates such as is found by Cerdan et al. (2010) (Fig 3A, B, C). This is due to the coarser spatial and temporal resolution of CE-DYNAM.'

**Report reviewer2:**

The manuscript presents a consistent and interesting study concerning carbon and erosion modelling. Authors presented the model CE-DYNAM which integrates sediment snd C dynamics (structure, concepts, limitations and evaluation) applied at Rhine catchment scale during the period 1850-2005 AD. The study contributes to model soil organic carbon stock changes using a processsbased model and to the evaluation of strategies to mitigate climate change at large scale.

Before publishing some minor revision should be addressed:

Line 61 "Most studies modeling soil erosion and its net effect on SOC dynamics at global scale" Please include references. Similarly in line 65

**Response:** We thank the reviewer for his or her constructive comments. We will include the following references to line 61 and 65 in the revised manuscript: 'Borrelli et al., 2018; Doetterl et al., 2012; Chappell et al., 2016; Lugato et al., 2018; Van Oost et al., 2007; Wang et al., 2017'

Line 107 Define Gross Primary Production

**Response:** We will define GPP in the revised manuscript.

Lines 132- 136 slope-length (describing the influence of topography on soil erosion risk.)

**Changes to manuscript:** We will include the following sentence to the revised manuscript: Note that the original Revised Universal Soil Loss Equation (RUSLE) (Renard et al., 1997) further includes a slope-length factor ( L ), which gives the length of a field in the direction of steepest descent, and a support practice factor ( P ), which accounts for management practices to mitigate soil erosion. These two factors have been excluded here, because their quantification still includes many uncertainties and is not practical for applications at regional to global scales.

The following sentence refers to slope-length? Revise: "These factors are a function of local manmade structures and management practices"

**Response:** This sentence refers to both L and P factors. The slope-length in agricultural steep landscapes is also largely controlled by the field size for example.

**Changes to manuscript:** We revise the sentence as: "These erosion factors are largely affected by local manmade structures (such as field size) and management practices"

Please clarify "we focus in this study on potential soil erosion and do not consider erosion-control practices"

**Changes to manuscript:** We modify this sentence as: 'In addition, we focus in this study on the potential effect of soil erosion on the C budget without erosion control practices. '

Line 153 similar catchments or not?
 recommend using past tense lines 190, 194, 210

**Response:** Similar catchments is meant in line 153. We will include these corrections in the revised manuscript.

Line 191 "reducing the sediment transport from hillslopes to floodplains" this cannot be alid for some other regions then line 153 could be revised as follows: "should be also applicable for other similar catchments

**Response:** We will revise line 153 accordingly

Line 236 revise: "discretization scheme SOC scheme"

Line 397 contrasting

Line 468 Please specify to highlight the new contribution of the study

**Changes to manuscript:** We add the following sentence after line 468 in the revised manuscript: 'In addition, the validation includes soil erosion data from new global soil erosion studies such as Borrelli et al. (2018) and Panagos et al. (2015).'

Line 479 Furthermore,

Line 518 avoid repetition "such as in our study. In our study"

Lines 519, 152, I suggest to include also in line 84

**Response:** We merge these lines in the last paragraph of the introduction of the revised manuscript.

Line 544 how about slope-length?

**Response/Changes to manuscript:** Thank you for pointing this out. We will add the following sentence in the revised manuscript: 'This is due to the coarser spatial and temporal resolution of CE-DYNAM, and because we did not include the effects of the slope-length factor (L), while Cerdan et al. (2010) assumed a constant slope length of a 100m.'

Line 585 "started around 1910 AD" Include a reference/source

**Changes to manuscript:** We will add to line 585: '....that started around 1910 AD according to the data on landcover and land use (Peng et al., 2017; Fig 7B).'

Line 604 Figs

Line 659 chapter? Section

Line 671-674 Move to introduction

**Response:** We will revise this accordingly.

Line 679 I recommend using past tense as in line 689

Line 750 due to this limitation I suggest to revise the sentences in lines 87-88 and line 153 (globally); 689-690

**Changes to manuscript:** We will revise sentence 87-88 as following: 'It should be noted here that the structure of CE-DYNAM is designed in a way that the model can be adapted easily to other large catchments after calibrating the model parameters to the specific environmental conditions in those catchments.'

We will revise line 153 as following: However, the aim of this study is to present a carbon erosion model that should be also applicable for other large catchments.

Line 821 showed as in line 817. Please revise in the whole manuscript and keep the same tense

**Response:** We will revise this accordingly

[revised manuscript text omitted]